# Toward stable replication of genomic information in pools of RNA molecules

**Ludwig Burger, Ulrich Gerland***

Physics of Complex Biosystems, Department of Bioscience, School of Natural Sciences, Technical University of Munich, Garching, Germany

## eLife Assessment

This **important** theoretical study examines the possibility of encoding genomic information in a collective of short overlapping strands (e.g., the Virtual Circular Genome (VCG) model). The study presents **convincing** theoretical arguments, simulations and comparisons to experimental data to point at potential features and limitations of such distributed collective encoding of information. The work should be of relevance to colleagues interested in molecular information processing and to those interested in pre-Central Dogma or prebiotic models of self-replication.

**Abstract** The transition from prebiotic chemistry to living systems requires the emergence of a scheme for enzyme-free genetic replication. Here, we analyze a recently proposed prebiotic replication scenario, the so-called Virtual Circular Genome (VCG) [Zhou et al., RNA 27, 1-11 (2021)]: Replication takes place in a pool of oligomers, where each oligomer contains a subsequence of a circular genome, such that the oligomers encode the full genome collectively. While the sequence of the circular genome may be reconstructed based on long oligomers, monomers and short oligomers merely act as replication feedstock. We observe a competition between the predominantly error-free ligation of a feedstock molecule to a long oligomer and the predominantly erroneous ligation of two long oligomers. Increasing the length of long oligomers and reducing their concentration decreases the fraction of erroneous ligations, enabling high-fidelity replication in the VCG. Alternatively, the formation of erroneous products can be suppressed if each ligation involves at least one monomer, while ligations between two long oligomers are effectively prevented. This kinetic discrimination (favoring monomer incorporation over oligomer–oligomer ligation) may be an intrinsic property of the activation chemistry, or can be externally imposed by selectively activating only monomers in the pool. Surprisingly, under these conditions, shorter oligomers are extended by monomers more quickly than long oligomers, a phenomenon that has already been observed experimentally [Ding et al., JACS 145, 7504-7515 (2023)]. Our work provides a theoretical explanation for this behavior and predicts its dependence on system parameters such as the concentration of long oligomers. Taken together, the VCG constitutes a promising scenario of prebiotic information replication: It could mitigate challenges in non-enzymatic copying via template-directed polymerization, such as short lengths of copied products and high error rates.

**\*For correspondence:**
gerland@tum.de

**Competing interest:** The authors declare that no competing interests exist.

## Introduction

In order to delineate possible pathways toward the emergence of life, it is necessary to understand how a chemical reaction network capable of storing and replicating genetic information might arise from prebiotic chemistry. RNA is commonly assumed to play a central role on this path, as it can store information in its sequence and catalyze its own replication (*Higgs and Lehman, 2015*; *Joyce, 1989*; *Robertson and Joyce, 2012*). While ribozymes capable of replicating strands of their own length have

been demonstrated in the laboratory (*Attwater et al., 2013*)*,* it remains elusive how enzyme-free self-replication might have worked before the emergence of such complex ribozymes.

One possible mechanism is template-directed primer extension (*Ding et al., 2022*; *Kervio et al., 2016*; *Leveau et al., 2022*; *Walton and Szostak, 2016*; *Welsch et al., 2023*). In this process, a primer hybridizes to a template and is extended by short oligonucleotides, thereby forming a (complementary) copy of the template strand. Considerable progress has been made in optimizing template-directed primer extension, but challenges remain: (i) The produced copies are likely to be incomplete. So far, at most 12 nt have been successfully added to an existing primer (*Leveau et al., 2022*). Moreover, as the pool of primer strands needs to emerge via random polymerization, the primer is likely to hybridize to the template at various positions, and not only to its 3'-end, leaving part of the 3'-end of the template uncopied (*Szostak, 2011*). (ii) Errors in enzyme-free copying are frequent due to the limited thermodynamic discrimination between correct Watson-Crick pairing and mismatches (*Kervio et al., 2010*; *Leu et al., 2013*; *Leu et al., 2011*). While some activation chemistries (relying on bridged dinucleotides) have been shown to exhibit improved fidelity (*Duzdevich et al., 2021*), the error probability still constrains the length of the genome that can be reliably replicated.

The issue of insufficient thermodynamic discrimination can, in principle, be mitigated by making use of kinetic stalling after the incorporation of a mismatch (*Leu et al., 2013*; *Rajamani et al., 2010*). By introducing a competition between the reduced polymerization rate and a characteristic timescale of the non-equilibrium environment, it is possible to filter correct sequences from incorrect ones (*Göppel et al., 2021*). To address the challenge of incomplete copies, Zhou et al. propose a new scenario of replication, the so-called Virtual Circular Genome (VCG) (*Zhou et al., 2021*). In this scenario, genetic information is stored in a pool of oligomers that are shorter than the circular genome they collectively encode: Each oligomer bears a subsequence of the circular genome, such that the collection of all oligomers encodes the full circular genome virtually. Within the pool, each oligomer can act as a template or primer (*Zhou et al., 2021*). The oligomers hybridize to each other and form complexes that allow for templated ligation of two oligomers, or for the extension of an oligomer by a monomer. Because the sequences of the ligated strands and the template are part of the genome, most of the products should also retain the sequence of the genome. That way, long oligomers encoding the circular genome can be produced at the expense of short oligomers (*Zhou et al., 2021*). The long strands, in turn, can assemble into catalytically active ribozymes. With a continuous influx of short oligomers, the VCG might allow for continuous replication of the virtually encoded circular genome. Importantly, replication in the VCG is expected to avoid the issue of incomplete copies. Since the genome is circular, it does not matter which part of the genome an oligomer encodes, as long as the sequence is compatible with the genome sequence. An additional feature of the VCG scenario is that replication should be achievable without the need of adding many nucleotides to a primer: Provided the concentration of oligomers decreases exponentially with their length, the concentration of each oligomer in the pool can be doubled by extending each oligomer only by a few nucleotides (*Zhou et al., 2021*). The extension of an oligomer by a few nucleotides in a VCG pool has already been demonstrated experimentally (*Ding et al., 2023*).

A recent computational study points out that the VCG scenario is prone to loss of genetic information via 'sequence scrambling' (*Chamanian and Higgs, 2022*). If the genome contains identical sequence motifs at multiple different loci, replication in the VCG will mix the sequences of these loci, thus destroying the initially defined genome. It is currently unclear which conditions could prevent this genome instability of VCGs, such that their genetic information is retained. Length distribution, sequence composition, oligonucleotide concentration, and environmental conditions, such as temperature, all affect the stability of complexes and thus the replication dynamics of the VCG pool. Here, we characterize the replication fidelity and yield of VCG pools using a kinetic model, which explicitly incorporates association and dissociation of RNA strands as well as templated ligation. We study a broad spectrum of prebiotically plausible and experimentally accessible oligomer pools, from pools containing only monomers and long oligomers of a single length to pools including a range of long oligomers with uniform or exponential concentration profile. The length of the included oligomers as well as their concentration is a free parameter of our model.

We find that, regardless of the pool composition, three competing types of template-directed ligation reactions emerge: (i) ligations between two short oligomers (or monomers), producing products too short to specify a unique genomic locus, (ii) ligations between a short and a long oligomer, typically

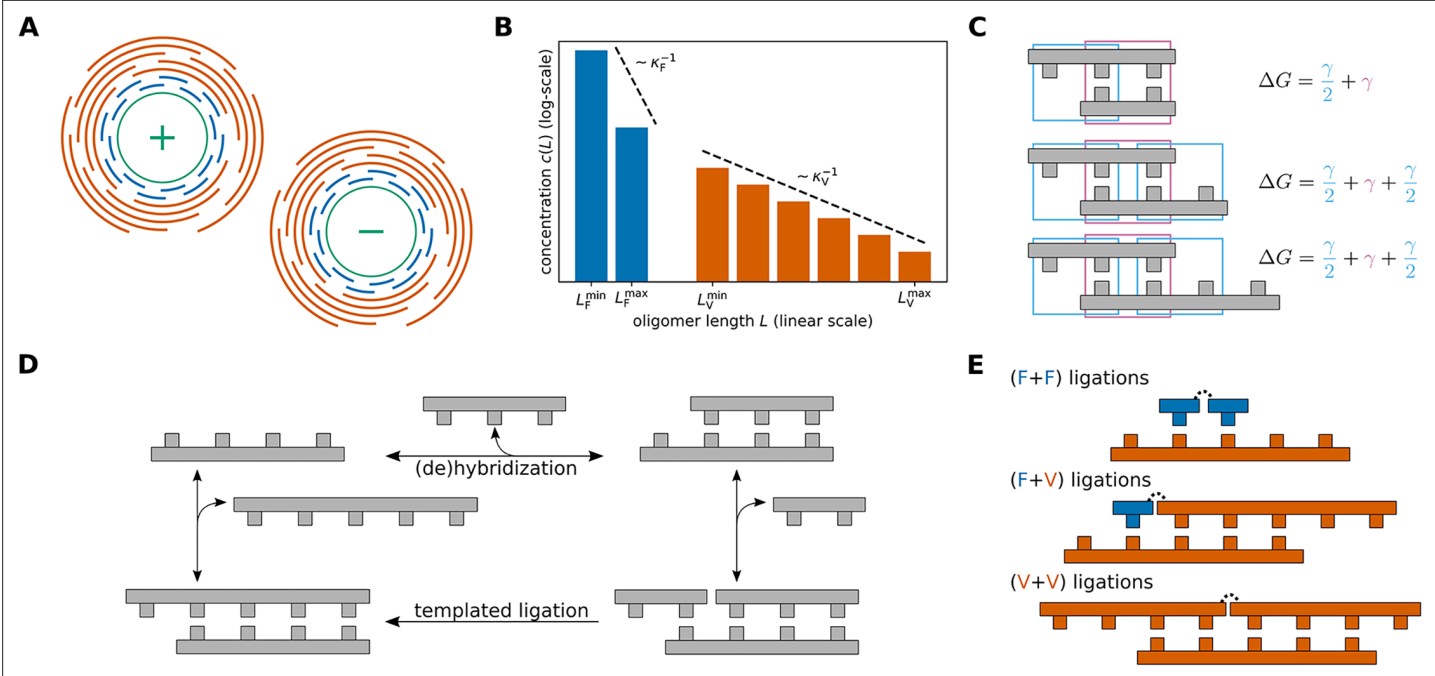

**Figure 1.** Model. (**A**) In the Virtual Circular Genome (VCG) scenario, a circular genome (depicted in green) as well as its sequence complement are encoded in a pool of oligomers (depicted in blue and orange). Collectively, the pool of oligomers encodes the whole sequence of the circular genome. Depending on their length, two types of oligomers can be distinguished: Long VCG oligomers specify a unique locus on the genome, while feedstock molecules (monomers or short oligomers) are too short to do so. (**B**) The length distribution of oligomers included in the VCG pool is assumed to be exponential. The concentration of feedstock and VCG oligomers as well as their respective length scales of exponential decay $\kappa_F^{-1}$ and $\kappa_V^{-1}$ can be varied independently. The set of included oligomer lengths can be restricted via $L_F^{min}$, $L_F^{max}$ and $L_V^{min}$, $L_V^{max}$. (**C**) The hybridization energy of complexes is computed using a simplified nearest-neighbor model: Each full block comprised of two base pairs (depicted in pink) contributes $\gamma$, while dangling end blocks (depicted in blue) contribute $\gamma/2$. (**D**) Oligomers form complexes via hybridization reactions, or dehybridize from an existing complex. The ratio of hybridization and dehybridization rate is governed by the hybridization energy (*Equation 1*). If two oligomers are adjacent to each other in a complex, they can undergo templated ligation. (**E**) Based on the length of the reacting oligomers, we distinguish three types of templated ligation: Ligation of two feedstock molecules (F+F), ligation of a feedstock molecule to a VCG oligomer (F+V) and ligation of two VCG oligomers (V+V).

generating longer products compatible with the genome sequence, and (iii) ligations between two long oligomers, which often yield sequences incompatible with the genome. These erroneous ligations of type (iii) are a key driver of sequence scrambling, as they covalently link oligomers originating from non-adjacent genomic loci, effectively 'mixing' distant regions of the genome. Fidelity is primarily determined by the competition between the correct extension of a long oligomer and the erroneous ligation of two long oligomers. The likelihood of the latter can be reduced by decreasing the relative abundance of long oligomers, even though this increases the frequency of unproductive ligations between short oligomers. As a result, fidelity can be improved at the cost of reduced yield. The efficiency, meaning the yield attainable at a fixed high fidelity, thus depends on the length distribution of the oligomers in the pool.

Alternatively, the issue of erroneous ligations is mitigated if ligations of long oligomers are kinetically suppressed, such that each ligation incorporates only one monomer at a time, as in the experimental study by *Ding et al., 2023*. In this case, the VCG concentration can be chosen arbitrarily large without compromising fidelity. Interestingly, our model predicts an unexpected feature: In the limit of high VCG concentrations, short oligomers are more likely to be extended than long oligomers, even though, intuitively, complexes containing longer oligomers are expected to be more stable and thus more productive. The same behavior was indeed observed experimentally (*Ding et al., 2023*). We provide an explanation for this feature and discuss its dependence on system parameters such as the length and the concentration of long oligomers in the pool.

# Results

## Modelling approach

In the VCG scenario, a circular genome is stored in a pool of oligomers, with each oligomer shorter than the genome it helps encode. Each oligomer bears a subsequence of the circular genome, such that, collectively, the oligomers encode the full genome (*Figure 1A*). As the spontaneous emergence of such VCG pools is expected to be rare (*Chamanian and Higgs, 2022*), our study focuses on the conditions under which an existing VCG pool can reliably replicate. We therefore begin with a known genome and an associated VCG pool, without addressing the question of origin. To set up our model of VCG dynamics, we specify (i) the circular genome used, (ii) the procedure by which the genome is mapped to a set of oligomers, and (iii) the chemical reactions governing the system's evolution.

### Circular genomes

For a given genome length, $L_G$, there are $4^{L_G}/2L_G$ distinct circular genomes (the factor $1/2L_G$ accounts for the freedom to select the starting position and to choose the Watson or the Crick strand as reference sequence). A key property of the genome is its unique motif length, $L_U$, defined as the shortest length such that all possible subsequences of length $L \geq L_U$ appear at most once in the genome. In other words, all subsequences of length $L \geq L_U$ specify a unique locus on the genome. In addition, each circular genome has another length scale, corresponding to the maximal motif length, up to which all possible motifs are contained in the genome. We refer to this length as the exhaustive coverage length, $L_E$. We typically analyze unbiased genomes in which all possible subsequences of length $L \leq L_E$ are contained at equal frequency.

### Construction of VCG pools

To specify a VCG pool that encodes a genomic sequence, one must select which subsequences are included in the pool at which concentrations. We consider unbiased pools, where the concentration of subsequences, $c(L)$, depends only on their length, $L$, that is all subsequences of a given length are included at equal concentration. We refer to the length-dependent concentration profile as the length distribution of the pool. Depending on their length, oligomers fall into two categories (*Figure 1B*): (i) short feedstock molecules (monomers and oligomers) and (ii) long VCG oligomers. Feedstock oligomers are oligomers that are shorter than the unique motif length $L_U$. Since their sequence appears multiple times on the genome, they do not encode a specific position on the genome. Thus, they serve as feedstock for the elongation of VCG oligomers rather than as information storage. Conversely, VCG oligomers, which are longer than the unique motif length $L_U$, have a unique locus on the circular genome. Collectively, the VCG oligomers enable the reconstruction of the full genome. The full-length distribution, $c(L)$, can be decomposed into the contributions of feedstock and VCG oligomers,

$$c(L) = c_F(L) + c_V(L).$$

We assume that both $c_F$ and $c_V$ follow an exponential length distribution. In our model, the concentration of VCG oligomers can be varied independently of the concentration of feedstock, and the length scales for the exponential decay ($\kappa_F^{-1}$ vs. $\kappa_V^{-1}$) may differ between feedstock and VCG oligomers. Additionally, we can restrict the set of oligomer lengths included in the pool by setting minimal and maximal lengths for feedstock and VCG oligomers individually,

$$c_F(L) = \hat{c}_F \exp\left(-\kappa_F L\right) \quad \text{if } L_F^{\min} \leq L \leq L_F^{\max},$$
$$c_V(L) = \hat{c}_V \exp\left(-\kappa_V L\right) \quad \text{if } L_V^{\min} \leq L \leq L_V^{\max}.$$

For any other oligomer length, the concentrations equal zero. This parameterization includes uniform length distributions as a special case ($\kappa_F = 0$ and $\kappa_V = 0$), and also allows for concentration profiles that are peaked. Peaked length distributions can emerge from the interplay of templated ligation, (de)hybridization, and outflux in open systems (*Rosenberger et al., 2021*). We define the total concentration of feedstock, $c_F^{\text{tot}} = \sum_L c_F(L)$, as well as the total concentration of VCG oligomers, $c_V^{\text{tot}} = \sum_L c_V(L)$. Their ratio will turn out to be an important determinant of the VCG dynamics.

## (De)hybridization kinetics

Oligomers can hybridize to each other to form double-stranded complexes, or dehybridize from an existing complex. For simplicity, we do not include self-folding within a strand, which is a reasonable assumption for short oligomers. The stability of a complex is determined by its hybridization energy, with lower hybridization energy indicating greater stability. We use a simplified nearest-neighbor energy model to compute the hybridization energy (*Göppel et al., 2022*; *Laurent et al., 2024*; *Rosenberger et al., 2021*): The total energy equals the sum of the energy contributions of all nearest-neighbor blocks in a given complex (*Figure 1C*). The energy contribution associated with a block of two Watson-Crick base pairs (matches) is denoted $\gamma < 0$, and dangling end blocks involving one Watson-Crick pair and one free base contribute $\gamma/2$. Nearest-neighbor blocks with mismatches increase the hybridization energy by $\gamma_{\mathrm{MM}} > 0$ per block, thus reducing the stability of the complex. The rate constants of hybridization and dehybridization are related via

$$\frac{k_{\mathrm{off}}}{k_{\mathrm{on}}} = c^{\circ} \exp\left(\beta \Delta G\right), \tag{1}$$

where $c^{\circ} = 1\mathrm{M}$ is the standard concentration, $\beta = (k_{\mathrm{B}}T)^{-1}$ is the Boltzmann factor, and $\Delta G$ is the free energy of hybridization. The association rate constant $k_{\mathrm{on}}$ is proportional to the encounter rate constant, $k_{\mathrm{enc}} = 1/(c^{\circ}t_0)$. The encounter timescale $t_0$ serves as the elementary time unit of the kinetic model, with all reaction timescales measured relative to it.

## Templated ligation

Two oligomers A and B that are hybridized adjacently to each other on a third oligomer C can produce a new oligomer A-B via templated ligation (*Figure 1D*). Depending on the length of A and B, we distinguish three types of ligation reactions (*Figure 1E*): (i) F+F ligations, in which two feedstock molecules ligate, (ii) F+V ligations, where a VCG oligomer is extended by a feedstock molecule, and (iii) V+V ligations involving two VCG oligomers. The formation of a covalent bond via templated ligation is not spontaneous but requires the presence of an activation reaction. Usually, these reactions add a leaving group to the 5′-end of the nucleotide, which is cleaved during bond formation (*Kervio et al., 2016*; *Walton and Szostak, 2016*). We assume that the concentration of activating agent is sufficiently high for the activation to be far quicker than the formation of the covalent bond, such that activation and covalent bond formation can be treated as a single effective reaction. When not otherwise stated, we assume that all possible templated ligation reactions occur with the same rate constant $k_{\mathrm{lig}}$.

## Observables

Templated ligation in the pool forms longer oligomers at the expense of shorter oligomers and monomers. While the product of an F+V ligation (or V+V ligation) is always a VCG oligomer, F+F ligations can produce feedstock or VCG oligomers. In both cases, a produced VCG oligomer can be correct (compatible with the genome) or incorrect (incompatible). We quantify these processes by measuring extension fluxes in units of nucleotides ligated to an existing strand (counting the length of the shorter ligated strand as the number of incorporated bases). In particular, we define the fidelity $f$ as the extension flux resulting in correct VCG oligomers relative to the flux resulting in any VCG oligomer,

$$f = \frac{\#\,\text{nucleotides incorporated in correct VCG oligomers}}{\#\,\text{nucleotides incorporated in VCG oligomers}}.$$

In addition, we introduce the yield $y$ as the proportion of total extension flux that produces VCG oligomers,

$$y = \frac{\#\,\text{nucleotides incorporated in VCG oligomers}}{\#\,\text{incorporated nucleotides}}.$$

Efficient replication of the VCG requires both high fidelity and high yield. Hence, we introduce the efficiency of replication $\eta$ as the product of fidelity and yield,

$$\eta = f \cdot y = \frac{\#\,\text{nucleotides incorporated in correct VCG oligomers}}{\#\,\text{incorporated nucleotides}}.$$

Moreover, we define the ligation share $s$ of a ligation type, which allows us to discern the contributions of different types of templated ligations (F+F, F+V, V+V) to fidelity, yield, and efficiency,

$$s(\text{type}) = \frac{\#\text{ nucleotides incorporated via ligation type}}{\#\text{ incorporated nucleotides}}.$$

## Replication efficiency reaches a maximum at intermediate concentrations of VCG oligomers

We begin our analysis of the dynamics of VCG pools with an exemplary genome of length $L_G = 16$ nt,

5 '-AAAGAGGACACGGCAU-3'
3 '-UUUCUCCUGUGCCGUA-5'.

This genome contains all possible monomers and dimers with equal frequency, ensuring that all motifs up to $L_E = 2$ nt are represented. Identifying a unique address on this genome requires at least three nucleotides. Therefore, the unique motif length is $L_U = 3$ nt, and VCG oligomers need to be at least 3 nt long. Further below, we also explore genomes of different lengths $L_G$, as well as varying characteristic sequence length scales $L_E$ and $L_U$ (genome construction detailed in the Methods section).

Based on the genome, we construct the initial oligomer pool. As a first step, we focus on a simple scenario in which the pool contains only monomers (serving as feedstock) and VCG oligomers of a single, defined length. The VCG pools are evolved in time using a stochastic simulation based on the Gillespie algorithm (*Göppel et al., 2022*; *Laurent et al., 2024*; *Rosenberger et al., 2021*). Since the Gillespie algorithm operates on the level of counts of molecules instead of concentrations, we must assign a volume to each system (in the range $1\,\mu m^3$ to $10000\,\mu m^3$, see Methods). Besides the volume, we also need to choose the reaction rate constants appropriately: (i) The association time $t_0$ is the fundamental time unit in our kinetic model, and all other times are expressed relative to $t_0$. Experimentally determined association rate constants are typically around $10^6 - 10^7\,M^{-1}s^{-1}$(*Ashwood et al., 2023*; *Braunlin and Bloomfield, 1991*; *Todisco et al., 2024a*; *Wetmur and Davidson, 1968*). For the purpose of estimating absolute timescales, we therefore assume a constant association timescale of $t_0 = (k_{on}c^\circ)^{-1} \approx 1\mu s$ in the following. (ii) The timescale of dehybridization is computed via *Equation 1* using the energy contribution $\gamma = -2.5\,k_B T$ for a matching nearest-neighbor block and $\gamma_{MM} = 25.0\,k_B T$ in case of mismatches. The high energy penalty of nearest-neighbor blocks involving mismatches, $\gamma_{MM}$, is chosen to suppress the formation of mismatches, while the value of $\gamma$ roughly matches the average energy of all matching nearest-neighbor blocks given by the Turner energy model of RNA hybridization (*Mathews et al., 2004*). (iii) For templated ligation, we select a reaction rate constant of $k_{lig} = 10^{-12}\,t_0^{-1}$. This choice of $k_{lig}$ is consistent with ligation rates measured in enzyme-free template-directed primer extension experiments, which range from around $10^{-6}s^{-1}$ ($10^{-12}\,t_0^{-1}$) (*Leveau et al., 2022*; *Sosson et al., 2019*) to roughly $10^{-4}s^{-1}$ ($10^{-10}\,t_0^{-1}$) (*Walton and Szostak, 2017*), depending on the underlying activation chemistry. This indicates that, for sufficiently short oligomers (up to about 11 nt long), hybridization and dehybridization occur much faster than ligation, so binding equilibrates before ligation takes place.

Based on the ligation events observed in the simulation, we compute the observables introduced above. Due to the small ligation rate constant, it is computationally unfeasible to simulate the time evolution for more than a few ligation time units. Consequently, ligation events are scarce, which leads to high variances in the computed observables. We mitigate this issue by calculating the observables based on the concentration of complexes that are in a productive configuration, even if they do not undergo templated ligation within the time window of the simulation (Methods).

The observable $y$ (yield) quantifies the fraction of this total flux directed to producing VCG oligomers. *Figure 2B* shows how the yield depends on the concentration of VCG oligomers, $c_V^{tot}$, at a fixed total monomer concentration, $c_F^{tot} = 0.1$mM. Here, the data points (with error bars) represent the simulation results with the different colors corresponding to different choices of initial VCG oligomer length, $L_V$. We observe that the yield increases monotonically with the concentration of VCG oligomers and approaches 100% for high $c_V^{tot}$. The concentration at which the pool reaches a yield of 50% depends on the oligomer length $L_V$: Shorter VCG oligomers require higher concentrations for high yield. The concentration dependence of the yield can be rationalized by the types of templated ligations that are involved. For low VCG concentration, most templated ligation reactions are dimerizations (1+1) with the VCG oligomers merely acting as templates (*Figure 2C*). As dimers are shorter than

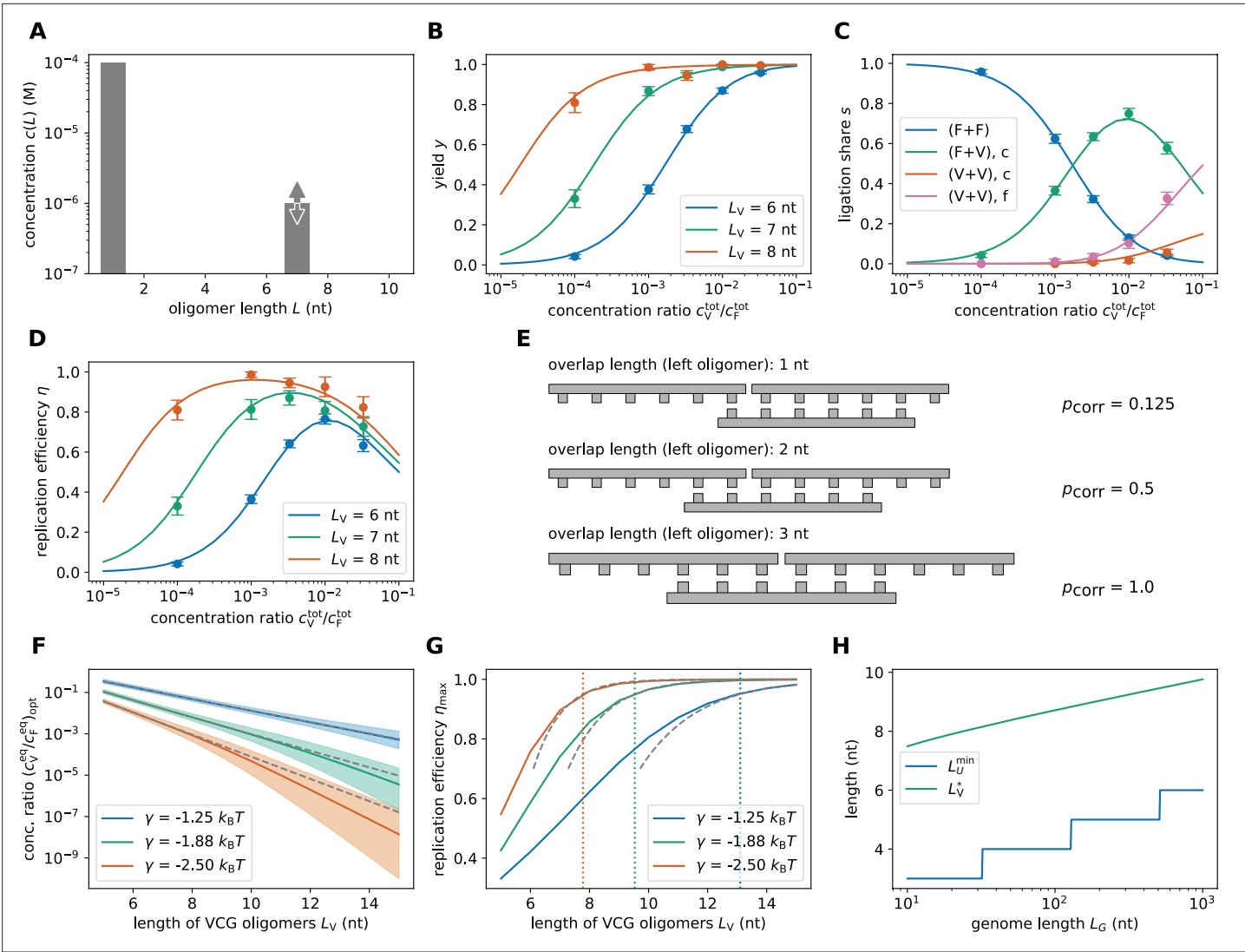

**Figure 2.** Replication performance of VCG pools containing VCG oligomers of a single length (single-length VCG pools). (**A**) The pool contains a fixed concentration of monomers, $c_F^{tot} = 0.1\,mM$, as well as VCG oligomers of a single length, $L_V$, at variable concentration $c_V^{tot}$ (the VCG oligomers cover all possible subsequences of the genome and its complement at equal concentration). (**B**) The yield increases as a function of $c_V^{tot}/c_F^{tot}$, because dimerizations become increasingly unlikely for high VCG concentrations. (**C**) The ligation share of different ligation types depends on the total VCG concentration: In the low concentration limit, dimerization (F+F) dominates; for intermediate concentrations, F+V ligations reach their maximum, while, for high concentrations, a substantial fraction of reactions are V+V ligations. The panel depicts the behavior for $L_V = 6\,nt$. (**D**) Replication efficiency is limited by the small yield for small $c_V^{tot}/c_F^{tot}$. In the limit of high $c_V^{tot}/c_F^{tot}$, replication efficiency decreases due to the growing number of error-prone V+V ligations. Maximal replication efficiency is reached at intermediate VCG concentration. (**E**) V+V ligations are prone to the formation of incorrect products due to the short overlap between educt strand and template. In general, the probability of correct product formation, $p_{corr}$, depends on the choice of circular genome and as well as its mapping to the VCG pool. The probabilities listed here refer to a VCG pool with $L_G = 16\,nt$, $L_E = 2\,nt$ and $L_U = 3\,nt$. (**F**) The optimal equilibrium concentration ratio of free VCG strands to free feedstock strands, which maximizes replication efficiency, decays as a function of length (continuous line). The analytical scaling law (dashed line, *Equation 2*) captures this behavior. The window of close-to-optimal replication, within which efficiency deviates no more than 1% from its optimum (shaded areas), increases with $L_V$ facilitating reliable replication without fine-tuning to match the optimal concentration ratio. (**G**) Maximal replication efficiency, which is attained at the optimal VCG concentration depicted in panel E, increases as a function of $L_V$ and approaches a plateau of 100%. For high efficiency, *Equation 3* provides a good approximation of the length-dependence of $\eta_{max}$ (dashed lines). The oligomer length at which replication efficiency equals 95% is determined using *Equation 3* (vertical dotted lines). (**H**) The unique motif length, $L_U^{min}$ increases logarithmically with the length of the genome, $L_G$. The length of VCG oligomers, $L_V$ at which the optimal replication efficiency reaches 95% (computed using *Equation 3*) exhibits the same logarithmic dependence on $L_G$.

the unique motif length $L_U$, their formation does not contribute to the yield, which explains the low yield in the limit of small $c_V^{tot}$. Conversely, for high VCG concentration, most of the templated ligations are F+V or V+V ligations, which produce oligomers of length $L \geq L_U$, implying high yield.

*Figure 2C* also shows that the relative contribution of V+V ligations increases with increasing $c_V^{tot}$, with a large fraction of them producing incorrect products (denoted V+V,f). This reduces the fidelity of replication, $f$, and leads to a trade-off between fidelity and yield, which causes replication efficiency, $\eta = f \cdot y$, to reach a maximum at intermediate VCG concentrations (*Figure 2D*). Using hexamers as an example, *Figure 2E* illustrates why V+V ligations are prone to forming incorrect oligomers: In order to ensure that two oligomers only ligate if their sequences are adjacent to each other in the true circular genome, both oligomers need to share an overlap of at least $L_U$ nucleotides with the template oligomer. Otherwise, the hybridization region is too short to identify the locus of the oligomer uniquely, and two oligomers from non-adjacent loci might ligate. The probability of forming incorrect products is a consequence of the combinatorics of possible subsequences in the VCG. Specifically, for a genome with $L_G = 16$ nt, there are 32 different hexamers. For example, if the left educt hexamer only has one nucleotide of overlap with the template, there are $1/4 \cdot 32 = 8$ possible educt oligomers. However, only one out of those eight hexamers is the correct partner for the right educt hexamer, implying an error probability of $1 - 1/8 = 7/8$ (first example in *Figure 2E*).

Characterizing replication efficiency via full simulations is computationally expensive. Depending on parameters, obtaining a single data point in *Figure 2B-D* can require hundreds of simulations, each lasting several days. To explore a broad parameter space more easily, we introduce an approximate adiabatic method that (i) assumes ligation is much slower than any hybridization or dehybridization event, and (ii) relies on a coarse-grained sequence-independent representation of oligomers. Details are provided in the Methods section. In brief, because ligation is rare, we first compute the equilibrium distribution of free and bound oligomers. Oligomers of the same length share a common concentration, and complex concentrations are determined via the mass action law using length-dependent dissociation constants. Combining the mass action law with a mass conservation constraint for each oligomer length allows us to compute the equilibrium concentrations of free VCG and feedstock strands, $c_V^{eq}$ and $c_F^{eq}$, given total concentrations $c_V^{tot}$ and $c_F^{tot}$. With these equilibrium values, we determine the concentrations of productive complexes and thus obtain the desired ligation-based observables without running full stochastic simulations.

The results of the adiabatic approach agree well with the simulation data (*Figure 2B-D*), supporting that the replication efficiency depends non-monotonously on the concentration of VCG oligomers, with a maximum at intermediate concentration. While the available simulation data only allows for a qualitative characterization of this trend, the adiabatic approach enables a quantitative analysis. For instance, we use the adiabatic approach to determine the equilibrium concentration ratio at which replication efficiency is maximal as a function of the VCG oligomer length $L_V$ (solid lines in *Figure 2F*). As expected from the qualitative trend observed in the simulation, pools containing longer oligomers reach their maximum for lower concentration of VCG oligomers. The shaded area indicates the range of VCG concentrations within which the pool's efficiency deviates by no more than one percent from its optimum. We observe that this range of close-to-optimal VCG concentrations increases with $L_V$. Thus, pools containing longer oligomers require less fine-tuning of the VCG concentration for replication with high efficiency.

In addition to the numerical results, we utilize the adiabatic approach to study the optimal equilibrium VCG concentration analytically (Appendix 1). We find that, for any choice of $L_V$, replication efficiency reaches its maximum when the fractions of dimerization (1+1) reactions and erroneous V+V ligations are equal (*Figure 2C* for $L_V = 6$ nt). This criterion can be used to derive a scaling law for the optimal equilibrium concentration ratio $c_V^{eq}/c_F^{eq}$ as a function of the oligomer length $L_V$,

$$\left. \frac{c_V^{eq}}{c_F^{eq}} \right|_{opt} \sim \sqrt{\frac{1}{\Lambda_{F+F}} - \frac{1}{L_V}} \exp\left(-\frac{|\gamma| L_V}{2}\right), \tag{2}$$

which is shown as dashed lines in *Figure 2F* (the length-scale $\Lambda_{F+F}$ is defined in Appendix 1). The optimal equilibrium concentration ratio decreases exponentially with $L_V$, while the hybridization energy $\gamma/2$ sets the inverse length scale of the exponential decay. Analytical estimate and numerical solution agree well, as long as the hybridization is weak and oligomers are sufficiently short. For strong

binding and long oligomers, complexes involving more than three strands play a non-negligible role, but such complexes are neglected in the analytical approximation.

*Figure 2G* shows how the maximal replication efficiency depends on the VCG oligomer length. Consistent with the qualitative trend observed in *Figure 2D*, longer oligomers enable higher maxima in replication efficiency. Regardless of the choice of $\gamma$, replication efficiency reaches 100% if $L_V$ is sufficiently high. Starting from *Equation 2*, we find the following approximation for the maximal replication efficiency attainable at a given oligomer length (dashed lines in *Figure 2G*),

$$\eta_{\max} \approx 1 - \eta^\circ \sqrt{\frac{1}{\Lambda_{F+F}} - \frac{1}{L_V}} L_V \exp\left(-\frac{|\gamma| L_V}{2}\right), \tag{3}$$

where $\eta^\circ$ is a genome-dependent constant (Appendix 1). This equation can provide guidance for the construction of VCG pools with high replication efficiency: Given a target efficiency, the necessary oligomer length and hybridization energy, that is temperature, can be calculated. In *Figure 2G*, we determine the oligomer length necessary to achieve $\eta_{\max} = 95\%$ for varying hybridization energies $\gamma$. At higher temperatures (weaker binding), VCG pools require longer oligomers to replicate with high efficiency.

*Equation 3* is not restricted to the specific example genome of length $L_G = 16$ nt, but applies more generally to genomes of arbitrary length. Any genome of length $L_G$ can contain at most $2 L_G$ distinct motifs. Consequently, the minimum length required to specify a unique address on the genome equals $L_U^{\min} = \left\lceil \frac{\ln(2L_G)}{\ln 4} \right\rceil$. By the same logic, the longest motif length for which all $4^L$ possible sequences can be exhaustively represented within the genome is $L_E^{\max} = \left\lfloor \frac{\ln(2L_G)}{\ln 4} \right\rfloor$. Both of these characteristic lengths scale logarithmically with genome size. For genomes where $L_E$ is taken to be maximal and $L_U$ minimal, we find that the characteristic VCG oligomer length $L_V^\star$ required for replication with high efficiency ($\eta_{\max} = 95\%$) also scales logarithmically with genome length (*Figure 2H*). Across genome lengths, the offset between $L_U$ and $L_V^\star$ is roughly constant.

In genomes with different choices of $L_E$ and $L_U$ ($L_E < L_E^{\max}$ and $L_U > L_U^{\min}$), the characteristic VCG oligomer length required for efficient replication, $L_V^\star$, is primarily determined by the unique motif length $L_U$. Specifically, the length of VCG oligomers, $L_V$, must exceed $L_U$, regardless of the value of $L_E$. This is shown for genomes of length $L_G = 64$ nt in Appendix 2, where we generate genomes with specified length scales $L_E$ and $L_U$ using a Metropolis–Hastings algorithm and analyze their replication efficiency. Intuitively, the required VCG oligomer length $L_V^\star$ is set by $L_U$, because $L_U$ defines the minimal hybridization region needed to ensure correct ligation via sequence-specific recognition. The precise offset between $L_U$ and $L_V^\star$ depends on genome-specific features, such as the frequency distribution of motifs with lengths between and $L_U$. If this distribution is nearly uniform (noting that at least one motif must repeat, or else it would constitute a unique address), then $L_V^\star$ will be close to $L_U$ (*Appendix 2—figure 1B*). In contrast, strongly biased motif distributions require larger $L_V^\star$ to achieve reliable replication (*Appendix 2—figure 1A*), though even in this case, $L_V^\star$ typically exceeds by only a few nucleotides.

## Replication in multi-length VCG pools is dominated by the longest oligomers

In the previous section, we characterized the behavior of pools containing VCG oligomers of a single length. We observed that V+V ligations are error-prone due to insufficient overlap between the educt strands and the template, whereas F+V ligations extend the VCG oligomers with high efficiency. The F+V ligations will gradually broaden the length distribution of the VCG pool, raising the question of how this broadening affects the replication behavior. In principle, introducing multiple oligomer lengths into the VCG pool might even improve the fidelity of V+V ligations, since a long VCG oligomer could serve as a template for the correct ligation of two shorter VCG oligomers.

To analyze this question quantitatively, we first consider the simple case of a VCG pool that only contains monomers, tetramers, and octamers. The concentration of monomers is set to $c(1) = 0.1$mM, while the concentrations of the VCG oligomers are varied independently (*Figure 3A*). Replication efficiency reaches its maximum at $c(8) \approx 0.1$μM and very low tetramer concentration, $c(4) \approx 7.4$pM, effectively resembling a single-length VCG pool containing only octamers. As shown in *Figure 3B*, the maximal efficiency is surrounded by a plateau of close-to-optimal efficiency. The octamer concentration

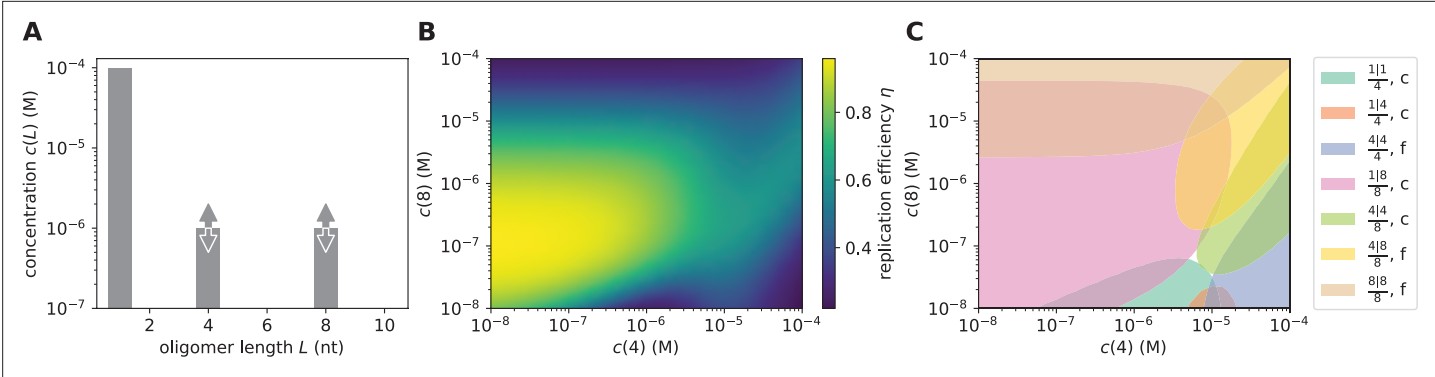

**Figure 3.** Replication performance of pools containing VCG oligomers of two different lengths. (**A**) The pool contains a fixed concentration of monomers, $c_F^{tot} = 0.1\,\text{mM}$, as well as tetramers and octamers at variable concentration. The hybridization energy per nearest-neighbor block is $\gamma = -2.5\,k_B T$. (**B**) Replication efficiency reaches its maximum for $c(8) \approx 0.1\,\mu\text{M}$ and significantly lower tetramer concentration, $c(4) \approx 7.4\,\text{pM}$. Efficiency remains close to maximal on a plateau around the maximum spanning almost two orders of magnitude in tetramer and octamer concentration. In addition, efficiency exhibits a ridge of increased efficiency for high tetramer concentration and intermediate octamer concentration. (**C**) Complexes that facilitate templated ligation are grouped by the length of the template and the educts, $\frac{L_{\text{educt,1}} | L_{\text{educt,2}}}{L_{\text{template}}}$. We distinguish complexes producing correct (labeled 'c') and false products (labeled 'f'). For each relevant type of complex, we highlight the region in the concentration plane where it contributes most significantly, that is at least 20% of the total ligation flux. The plateau of high efficiency is dominated by the ligation of monomers to octamers, whereas the ridge of increased efficiency is due to the correct ligation of two tetramers templated by an octamer.

The online version of this article includes the following figure supplement(s) for figure 3:

**Figure supplement 1.** Replication performance in pools containing tetramers and octamers for strong binding affinity, $\gamma = -5.0\,k_B T$.

**Figure supplement 2.** Replication performance in pools containing heptamers and octamers for weak binding affinity, $\gamma = -2.5\,k_B T$.

**Figure supplement 3.** Replication performance in pools containing heptamers and octamers for strong binding affinity, $\gamma = -5.0\,k_B T$.

can be varied by more than one order of magnitude without significant change in efficiency. Similarly, adding tetramers does not affect efficiency as long as the tetramer concentration does not exceed the octamer concentration. *Figure 3C* illustrates that the plateau of close-to-optimal efficiency coincides with the concentration regime where the ligation of a monomer to an octamer with another octamer acting as template, $\frac{1|8}{8}$, is the dominant ligation reaction. For high tetramer concentration and intermediate octamer concentration, templated ligation of tetramers on octamer templates, $\frac{4|4}{8}$, surpasses the contribution of $\frac{1|8}{8}$ ligations (green shaded area in *Figure 3C*). The $\frac{4|4}{8}$ reactions give rise to a ridge of increased efficiency, which, however, is small compared to the plateau of close-to-optimal efficiency (*Figure 3B*). Even though reactions of the type $\frac{4|4}{8}$ produce correct products in most cases, they compete with error-prone ligations like $\frac{4|8}{8}$ or $\frac{4|4}{4}$, which reduce fidelity. Increasing the binding affinity, that is lowering $\gamma$, enhances the contribution of $\frac{4|4}{8}$ ligations at the expense of $\frac{4|8}{8}$ ligations, resulting in an increased local maximum in replication efficiency (*Figure 3—figure supplement 1*). However, very strong hybridization energy, $\gamma = -5.0\,k_B T$, is necessary for $\frac{4|4}{8}$ ligations to give rise to similar efficiency as $\frac{1|8}{8}$ ligations. Thus, in this example, high efficiency replication facilitated by the ligation of short VCG oligomers on long VCG oligomers is in theory possible, but requires unrealistically strong binding affinity. Moreover, this mechanism is only effective if educts and template differ significantly with respect to their length. If the involved VCG oligomers are too similar in size, templated ligation of two short oligomers tends to be error-prone, irrespective of the choice of binding affinity (*Figure 3—figure supplement 2* and *Figure 3—figure supplement 3*). In that case, F+V ligations remain the most reliable replication mechanism.

We observe similar behavior in VCG pools containing a range of oligomer lengths from $L_V^{\min}$ to $L_V^{\max}$, rather than just tetramers and octamers. For a uniform VCG concentration profile (*Figure 4A*, dark gray), the maximal replication efficiency (at the optimal VCG concentration) is attained when F+V ligations involving long VCG oligomers dominate the templated ligation (see *Figure 4D* showing the contribution of different ligation types in pools with varying $L_V^{\min}$ at fixed $L_V^{\max} = 10\,\text{nt}$). Importantly, the maximal replication efficiency is always bounded by the efficiency of the longest VCG oligomer in the pool, independent of the presence or length of shorter oligomers (blue curve in *Figure 4B*; identical to the orange curve in *Figure 2G*). Including short VCG oligomers has minimal effect on

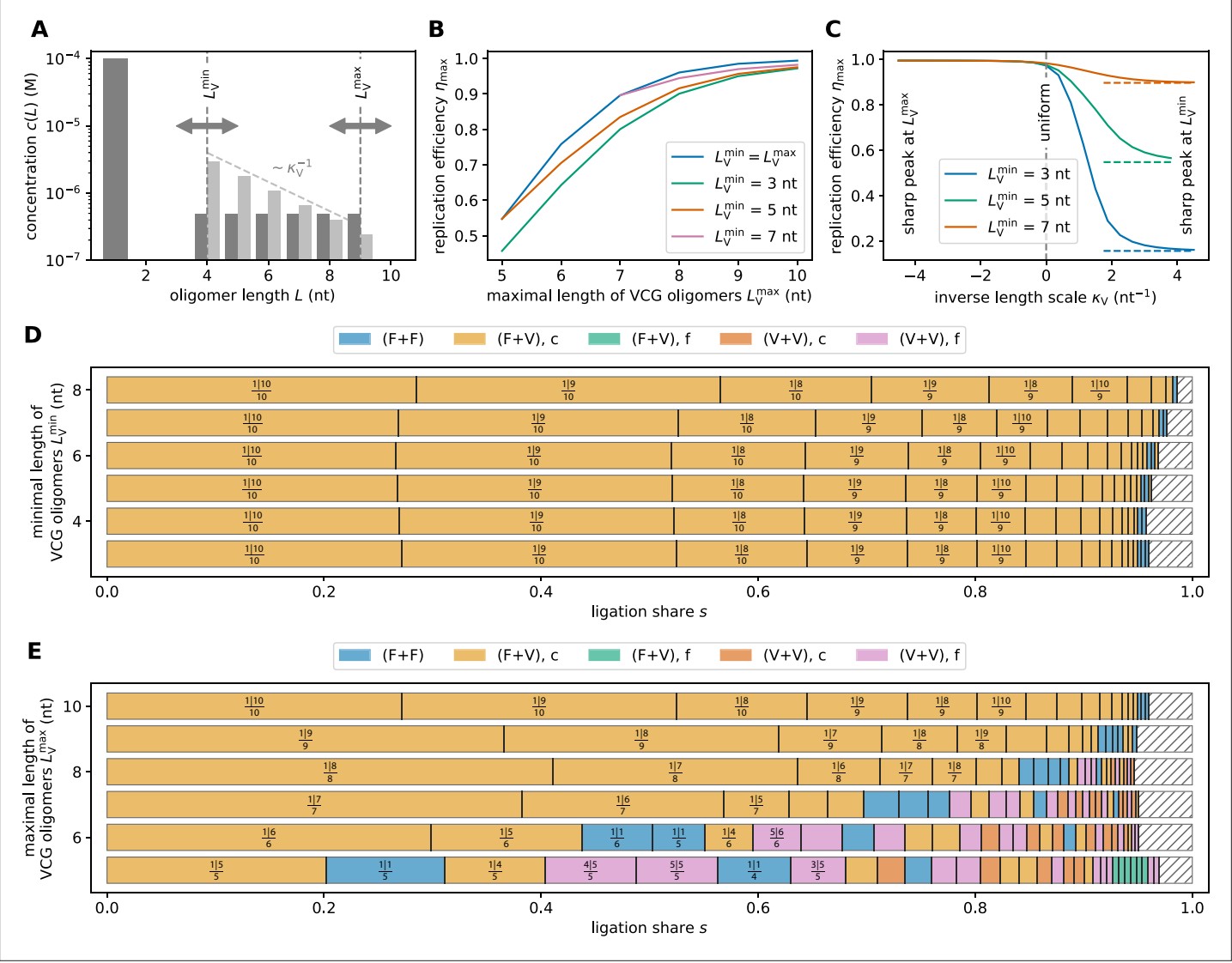

**Figure 4.** Replication performance of multi-length VCG pools. (**A**) The pool contains a fixed concentration of monomers, $c_F^{tot} = 0.1$mM, as well as long oligomers in the range $L_V^{min} \leq L_V \leq L_V^{max}$ at variable concentration $c_V^{tot}$. The length dependence of the concentration profile is assumed to be uniform (for panels B, D, and E) or exponential (for panel C); its steepness is set by the parameter $\kappa_V$. (**B**) If the length distribution is uniform, reducing $L_V^{min}$ decreases the maximal efficiency, whereas increasing $L_V^{max}$ increases it. Pools containing a range of oligomer lengths are always outperformed by single-length VCGs (blue curve). (**C**) Assuming an exponential length distribution of VCG oligomers allows us to tune from a poorly-performing regime (dominated by oligomers of length $L_V^{min}$) to a well-performing regime (dominated by oligomers of length $L_V^{max}$). In the limit $\kappa_V \to \infty$, $\eta_{max}$ approaches the replication efficiency of single-length pools containing only oligomers of length $L_V^{min}$ (dashed lines). (**D**) For high $L_V^{max}$, replication is dominated by primer extension of the long oligomers in the VCG (here $L_V^{max} = 10$ nt). In this limit, addition of shorter oligomers leaves the dominant F+V ligations almost unchanged. (**E**) Reducing $L_V^{max}$ for fixed $L_V^{min} = 3$ nt increases the fraction of unproductive (i.e. dimerization) or erroneous ligation reactions.

the dominant ligation types and only slightly increases the proportion of unproductive F+F ligations. Consequently, reducing $L_V^{min}$ while keeping $L_V^{max}$ fixed leads to only a modest decline in maximal efficiency. In contrast, decreasing $L_V^{max}$ while holding $L_V^{min}$ constant causes a substantial reduction in efficiency (**Figure 4B**), because the longest oligomer in the pool sets an upper bound on replication efficiency. Moreover, at low $L_V^{max}$, short and long oligomers become more similar in length, giving rise to a spectrum of erroneous V+V ligations that compete with the productive F+V ligations (**Figure 4E**).

In a realistic prebiotic scenario, the concentration profile of the VCG pool would not be uniform. Depending on the mechanism producing the pool and its coupling to the non-equilibrium environment, it might have a concentration profile that decreases or increases exponentially with length. We use the parameter $\kappa_V$ to control this exponential length dependence (**Figure 4A**): For negative $\kappa_V$,

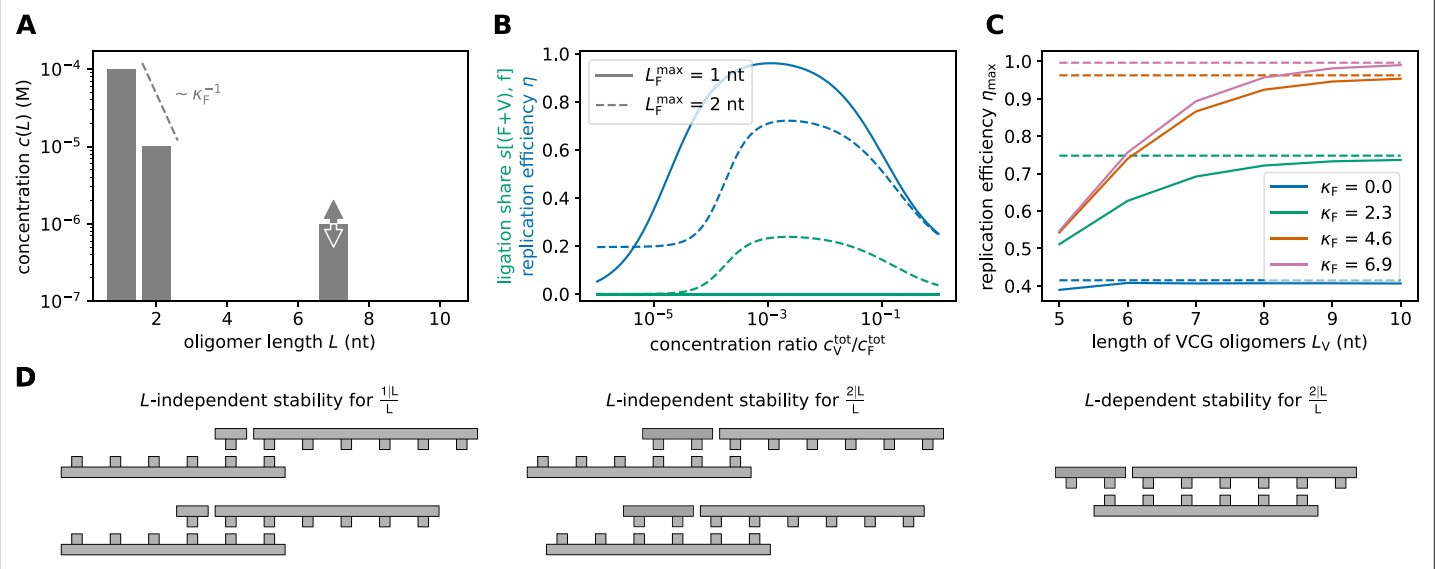

**Figure 5.** Replication performance of single-length VCG pools containing monomers and dimers as feedstock. (**A**) The pool contains a fixed total concentration of feedstock, $c_F^{tot} = 0.1$ mM, partitioned into monomers and dimers, as well as VCG oligomers of a single length, $L_V$. The proportion of monomers and dimers can be adjusted via $\kappa_F$, and the concentration of the VCG oligomers is a free parameter, $c_V^{tot}$. (**B**) Replication efficiency exhibits a maximum at intermediate VCG concentration in systems with (dashed blue curve) and without dimers (solid blue curve). The presence of dimers reduces replication efficiency significantly, as they enhance the ligation share of incorrect F+V ligations (dashed green curve). The panel depicts the behavior for $L_V = 7$ nt and $\kappa_F = 2.3$. (**C**) Optimal replication efficiency increases as a function of oligomer length, $L_V$, and asymptotically approaches a plateau (dashed lines, *Equation 4*). The value of this plateau, $\eta_{max}^{\infty}$, is determined by the competition between correct and false 2+V reactions, both of which grow exponentially with $L_V$. Thus, $\eta_{max}^{\infty}$ depends on the relative concentration of the dimers in the pool: the more dimers are included, the lower is $\eta_{max}^{\infty}$. (**D**) Erroneous 1+V ligations are possible if the educt oligomer has a short overlap region with the template. The hybridization energy for such configurations is small and independent of the length of the VCG oligomers (left). While 2+V ligations may produce incorrect products via the same mechanism (middle), incorrect product can also be caused by complexes in which two VCG oligomers hybridize perfectly to each other, but the dimer has a dangling end. The stability of these complexes increases exponentially with oligomer length (right).

the concentration increases as a function of length, while exponentially decaying length distributions have positive $\kappa_V$. We find that replication efficiency is high if the concentration of long VCG oligomers exceeds or at least matches the concentration of short VCG oligomers ($\kappa_V \leq 0$ in *Figure 4C*). In that case, replication efficiency is dominated by the long oligomers in the pool, since these form the most stable complexes. As the concentration of long oligomers is decreased further ($\kappa_V > 0$), the higher stability of complexes formed by longer oligomers is eventually insufficient to compensate for the reduced concentration of long oligomers. Replication efficiency is then governed by short VCG oligomers, which exhibit lower replication efficiency. In the limit $\kappa_V \rightarrow \infty$, replication efficiency approaches the replication efficiency of a single-length VCG pool containing only oligomers of length $L_V^{min}$ (*Figure 4C* and *Figure 2G*).

## Adding dinucleotides to the feedstock decreases replication efficiency

So far, we focused on ensembles that contain solely monomers as feedstock. However, examining the influence of dimers on replication in VCG pools is of interest, since dinucleotides have proven to be interesting candidates for enzyme-free RNA copying (*Leveau et al., 2022*; *Sosson et al., 2019*; *Walton and Szostak, 2016*). For this reason, we study oligomer pools like those illustrated in *Figure 5A*: The ensemble contains monomers, dimers, and VCG oligomers of a single length, $L_V$. As our default scenario, the dimer concentration is set to 10% of the monomer concentration, corresponding to $\kappa_F = 2.3$, but this ratio can be modified by changing $\kappa_F$.

*Figure 5B* compares the replication efficiency of a pool with ($L_F^{max} = 2$ nt) and without dimers ($L_F^{max} = 1$ nt). In both cases, pools that are rich in VCG oligomers exhibit low efficiency of replication. As erroneous V+V ligations are the dominant type of reaction in this limit, all pools achieve the same efficiency regardless of the presence of dimers. In contrast, when the pool is rich in feedstock (small $c_V^{tot}/c_F^{tot}$), pools with and without dimers behave differently: If only monomers are included, the

efficiency approaches zero, as dimerizing monomers do not contribute to the yield, and thus not to the efficiency. However, the presence of dimers enables the ligation of monomers and dimers to form trimers and tetramers, which lead to a non-zero yield. Given the low VCG concentration, the ligations are likely to proceed using dimers as template. As a consequence, educt oligomers can only hybridize to the template with a single-nucleotide-long hybridization region, leading to frequent formation of incorrect products and, consequently, low efficiency.

Replication efficiency reaches its maximum at intermediate VCG concentrations, where replication is dominated by F+V ligations. Notably, the maximal attainable efficiency is significantly lower for pools with dimers than without, as dimers increase the number and the stability of complex configurations that can form incorrect products (*Figure 5C*). Without dimers, ligation products are only incorrect if the overlap between the VCG educt oligomer and template is shorter than the unique motif length, $L_U$. With dimers, however, dangling end dimers can cause incorrect products even in case of long overlap of educt oligomer and template (right column in *Figure 5D*). The stability of the latter complexes depends on the length of the VCG oligomers, $L_V$, whereas the stability of complexes facilitating incorrect monomer addition is independent of oligomer length (*Figure 5D*).

In the presence of dimers, the length-dependent stability of complexes allowing for correct and incorrect F+V ligations causes a competition, which sets an upper bound on the efficiency of replication (Appendix 3),

$$\eta_{\max} \leq \frac{\mathcal{K}^{a,\circ}_{1+V,c} + 0.5\,\mathcal{K}^{a,\circ}_{2+V,c}\,e^{-\kappa_F}}{\mathcal{K}^{a,\circ}_{1+V,c} + 0.5\,(\mathcal{K}^{a,\circ}_{2+V,c} + \mathcal{K}^{a,\circ}_{2+V,f})\,e^{-\kappa_F}}. \tag{4}$$

Here, we introduced effective association constants $\mathcal{K}^a$, which depend differently on the VCG oligomer length, $L_V$. While the effective association constant $\mathcal{K}^a_{1+V,f}$ of complexes enabling incorrect 1+V ligations is length-independent, the effective association constant for incorrect 2+V ligations, $\mathcal{K}^a_{2+V,f}$, scales exponentially with $L_V$,

$$\mathcal{K}^a_{2+V,f} = \mathcal{K}^{a,\circ}_{2+V,f}\exp(|\gamma|L_V)$$

The effective association constants for correct 1+V and 2+V ligations also scale exponentially with the oligomer length (*Appendix 3—figure 1*).

In systems without dimers, that is $\kappa_F \to \infty$, $\eta_{\max}$ approaches 100%, which is consistent with the behavior observed in the previous sections. Conversely, in systems containing dimers, the maximal efficiency remains at a value below 100%, which depends on the concentration of dimers in the feedstock. *Figure 5C* shows the dependence of maximal replication efficiency on the length of VCG oligomers in pools containing monomers, dimers, and VCG oligomers. As $L_V$ increases, $\eta_{\max}$ converges towards the upper bound defined in *Equation 4* (dashed line in *Figure 5C*).

## Error-prone ligation of VCG oligomers can be kinetically suppressed

In all scenarios considered so far, the efficiency of replication is limited by a common mechanism, regardless of the specifics of the VCG pool: Ligations involving two oligomers that hybridize to the template over a region shorter than $L_U$ are prone to generate incorrect products. In previous sections, we minimized these erroneous ligations by fine-tuning the concentration and length of VCG oligomers. However, such control may become unnecessary if the typically error-prone ligation of two oligomers (V+V) is kinetically suppressed. Kinetic suppression can be an intrinsic property of the activation chemistry: Templated ligation of two oligomers can be several orders of magnitude slower than the extension of an oligomer by a single monomer (*Ding et al., 2022*; *Prywes et al., 2016*). As a result, V+V ligations are disfavored purely by their slower kinetics. In addition, it is conceivable that only monomers are chemically activated while longer oligomers remain inactive, which would further reduce the likelihood of erroneous ligations. This scenario has already been explored experimentally (*Ding et al., 2023*). In natural environments, it could occur, for instance, when activated monomers are produced externally and then diffuse into compartments containing the VCG but lacking internal activation pathways (*Kriebisch et al., 2024*; *Toparlak et al., 2023*).

Within our model, we capture the kinetic suppression by introducing two different rates of ligation, $k_{\mathrm{lig},1}$ for ligations involving a monomer and $k_{\mathrm{lig},>1}$ for ligations involving no monomer, allowing for kinetic discrimination between these processes. We explore the resulting replication efficiency in the

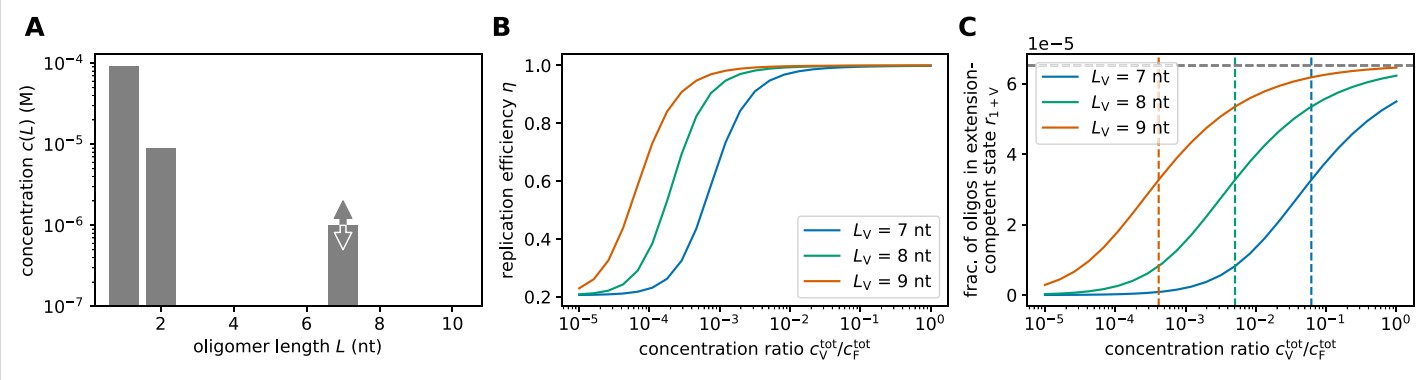

**Figure 6.** Replication performance of single-length VCG pools with kinetic suppression of ligation between oligomers. (**A**) The pool contains reactive monomers alongside non-reactive dimers and VCG oligomers of a single length. The concentrations of monomers and dimers are fixed, $c(1) = 0.091$ mM and $c(2) = 9.1$ μM, adding up to a total feedstock concentration of $c_F^{tot} = 0.1$ mM, while the concentration of VCG oligomers, $c_V^{tot}$, is varied. (**B**) Unlike in pools in which all ligation processes occur, replication efficiency does not decrease at high VCG concentration if ligations that do not involve monomers are kinetically suppressed. Instead, replication efficiency approaches an asymptotic value of 100%, as erroneous V+V ligations are impossible. (**C**) The fraction of oligomers that are in a monomer-extension-competent state depends on the total concentration of VCG oligomers. At low VCG concentration, most oligomers are single-stranded, and extension of oligomers by monomers is scarce. At high VCG concentration, $r_{1+V}$ approaches the asymptotic value $r_{1+V}^\infty$ (grey dashed line, **Figure 7**.). In this limit, almost all oligomers form duplexes, which facilitate monomer addition upon hybridization of a monomer. Thus, the asymptotic fraction of oligomers that gets extended by monomers is not determined by the oligomer length, but by the binding affinity of monomers to existing duplexes. Conversely, the threshold concentration at which $r_{1+V} = r_{1+V}^\infty/2$ depends on oligomer length (colored dashed lines): Longer oligomers reach higher $r_{1+V}$ at lower VCG concentration.

limit of perfect kinetic discrimination ($k_{lig,>1}/k_{lig,1} \to 0$) where only monomers are reactive for ligation. We first consider a pool where the reactive monomers are mixed with VCG oligomers of a single length as well as non-reactive dimers (**Figure 6A**). We vary the concentration of VCG oligomers, but keep the feedstock concentrations constant. For small VCG concentrations, we observe low efficiencies (**Figure 6B**), as the ligation of two monomers, or one monomer and one dimer, is most likely. Conversely, high $c_V^{tot}$ facilitates the formation of complexes in which VCG oligomers are extended by monomers, which implies high efficiency (**Figure 6B**). Note that, unlike in systems where all oligomers are reactive, replication efficiency does not decrease for high VCG concentration, as erroneous V+V ligations are impossible. Instead, perfect replication efficiency (100%) is reached for sufficiently high $L_V$.

While replication efficiency characterizes the relative amount of nucleotides used for the correct elongation of VCG oligomers, it is also interesting to analyze which fraction of VCG oligomers is in a monomer-extension-competent state,

$$r_{1+V} = c_{1+V}^{eq}/c_V^{tot}.$$

Here, $c_{1+V}^{eq}$ denotes the equilibrium concentration of all complexes enabling the addition of a monomer to any VCG oligomer. We find that $r_{1+V}$ depends on the VCG concentration qualitatively in the same way as the efficiency: $r_{1+V}$ is small for small VCG concentration but approaches a value $r_{1+V}^\infty$ asymptotically for high VCG concentration (**Figure 6C**). In this limit, almost all VCG oligomers are part of a duplex. Monomers can bind to these duplexes to form complexes allowing for 1+V ligations. Since almost all VCG oligomers are already part of a duplex, increasing $c_V^{tot}$ further does not increase the fraction of VCG oligomers that can be extended by monomers. Instead, the asymptotic value is determined by the concentration of monomers and their binding affinity $K_d(1)$ to an existing duplex (Appendix 4),

$$r_{1+V}^\infty \approx \frac{1}{6}\left(\frac{c^\circ}{K_d(1)} + 1 - \frac{2\,K_d(1)}{3\,c^\circ}\right)\frac{c(1)}{c^\circ}. \tag{5}$$

While the asymptotic value $r_{1+V}^\infty$ does not depend on the length of the VCG oligomers, the threshold VCG concentration at which $r_{1+V} = r_{1+V}^\infty/2$ scales exponentially with $L_V$ (Appendix 4),

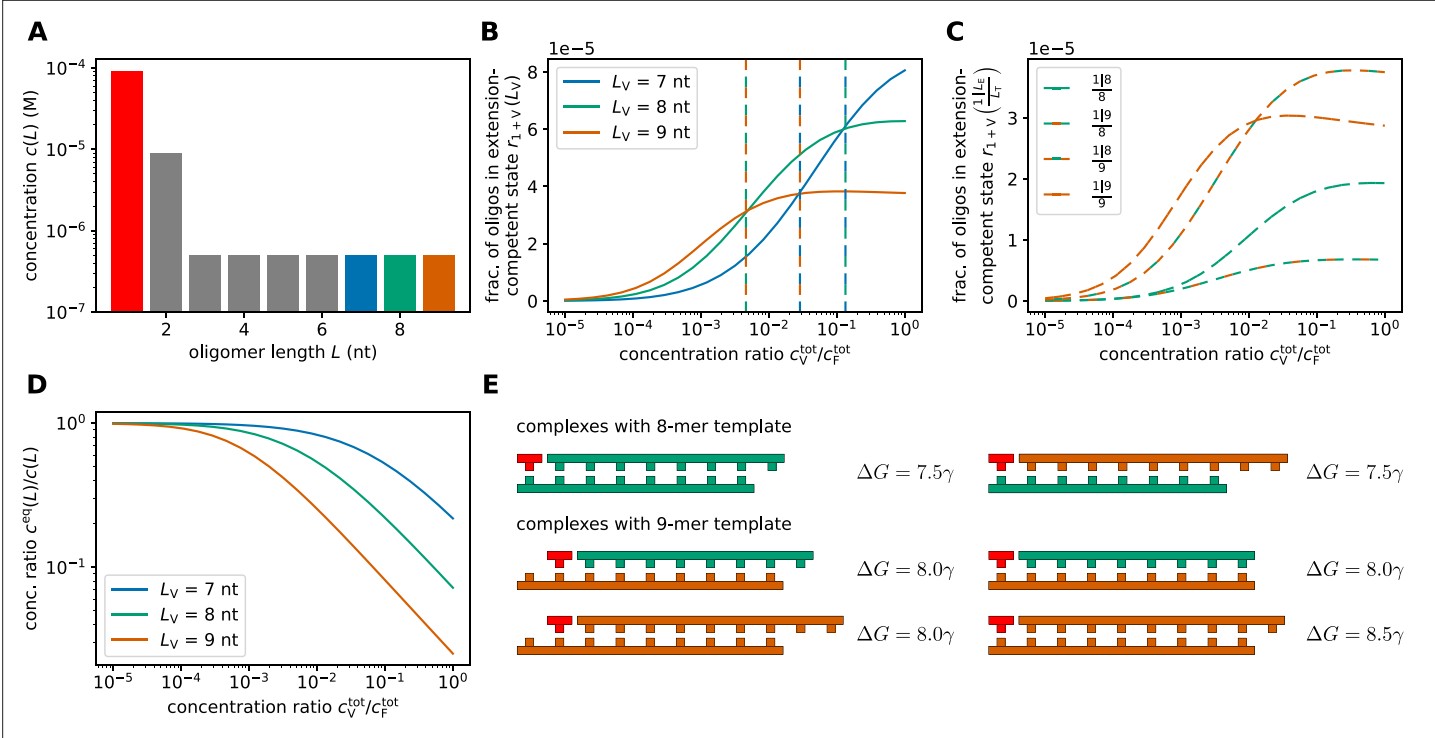

**Figure 7.** Replication performance of multi-length VCG pools with kinetic suppression of ligation between oligomers. (**A**) The pool contains reactive monomers as well as non-reactive dimers and VCG oligomers. The concentrations of monomers and dimers are fixed, $c(1) = 0.091\text{mM}$ and $c(2) = 9.1\,\mu\text{M}$, adding up to a total feedstock concentration of $c_V^{tot} = 0.1\,\text{mM}$, while the total concentration of VCG oligomers, $c_V^{tot}$, is varied. All VCG oligomers are assumed to have the same concentration. (**B**) At low VCG concentration, long oligomers are more likely in a monomer-extension-competent state than short oligomers, whereas at high VCG concentration, the trend reverses and short oligomers are more likely to be extended by monomers ('productivity inversion'). The threshold concentration at which a short oligomer starts to outperform a longer oligomer depends on the lengths of the compared oligomers (dashed lines). (**C**) The mechanism underlying productivity inversion can be understood based on the pair-wise competition of different VCG oligomers, for example 8-mers vs. 9-mers. Over the entire range of VCG concentrations, complexes with 8-mer templates have a lower relative equilibrium concentration than complexes with 9-mer templates (bottom two curves vs. top two curves). As the concentration of VCG oligomers is increased, ligations of type $\frac{1|8}{9}$ exceed ligations of type $\frac{1|9}{9}$, that is the fraction of 8-mers that are extended by monomers using a 9-mer as a template exceeds the fraction of extended 9-mers. (**D**) The equilibrium concentration of free oligomer decreases with increasing $c_V^{tot}$. For longer oligomers, the equilibrium fraction of free oligomers is lower, as they can form more stable complexes with longer hybridization sites. (**E**) Complexes in which 8-mers serve as template are less stable than complexes with 9-mer templates, explaining why complexes with 8-mer templates are more abundant than complexes with 9-mer templates (see panel C). Complexes with 9-mer template have similar stability regardless of the length of the educt oligomer, that is $\frac{1|8}{9}$ and $\frac{1|9}{9}$ are similarly stable. This similar stability, together with the higher concentration of free 8-mers compared to 9-mers (see panel D), is the reason why the fraction of monomer-extended 8-mers exceeds the one of 9-mers (see panel C).

$$c_V^{tot} \sim \exp(-L_V|\gamma|).$$

This scaling implies that longer oligomers require exponentially lower VCG concentration to achieve a given ratio $r_{1+V}$ (**Figure 6C**), as their greater length allows them to form more stable complexes. This observation implies that pools with longer oligomers will always be more productive than pools with shorter oligomers (at equal VCG concentration).

The behavior becomes more complex in pools containing VCG oligomers of multiple lengths, due to the competitive binding within such heterogeneous pools. To illustrate this, we examine an ensemble containing VCG oligomers ranging from $L_V^{min} = 3\,\text{nt}$ to $L_V^{max} = 9\,\text{nt}$, along with the same feedstock as previously (reactive monomers and non-reactive dimers, see **Figure 7A**). For simplicity, we assume that the length distribution of VCG oligomers is uniform. We study the fraction of oligomers in a monomer-extension-competent state as a function of oligomer length,

$$r_{1+V}(L) = c_{1+V}^{eq}(L)\,/\,c(L),$$

where $c^{eq}_{1+V}(L)$ denotes the equilibrium concentration of all complexes that allow monomer-extension of VCG oligomers of length $L$. At low VCG concentration, longer oligomers are more likely to be extended by monomers than shorter ones (*Figure 7B*). This behavior is intuitive, as longer oligomers tend to form more stable complexes, which lead to higher productivity. Surprisingly, increasing the VCG concentration reverses the length-dependence of the productivity, such that short oligomers are more likely to be extended by monomers than long ones (note the three crossings of the curves in *Figure 7B*). For example, 8-mers are more likely to undergo primer extension than 9-mers once the VCG concentration exceeds $\approx 0.5\,\mu M$. We derived a semi-analytical expression for the threshold VCG concentrations at which oligomers of two different lengths have equal productivity (dashed lines in *Figure 7B*, Appendix 5 and Appendix 6).

To understand the mechanism underlying this inversion of productivity, we analyze how different complex types contribute to $r_{1+V}(L)$. We introduce

$$r_{1+V}\left(\frac{1|L_E}{L_T}\right) = c^{eq}_{1+V}\left(\frac{1|L_E}{L_T}\right) \Big/ c(L_E),$$

which denotes the fraction of oligomers of length $L_E$ that are in a monomer-extension-competent complex configuration that uses an oligomer of length $L_T$ as template. The term $c^{eq}_{1+V}\left(\frac{1|L_E}{L_T}\right)$ includes the sum over all possible configurations of complexes with the given lengths. Focusing on 8-mers and 9-mers as an example, we observe that complexes utilizing the 9-mer as template are responsible for the inversion of productivity (top two curves in *Figure 7C*): As the VCG concentration increases, the fraction of monomer-extendable 8-mers eventually surpasses the fraction of monomer-extendable 9-mers, that is $r_{1+V}\left(\frac{1|8}{9}\right) > r_{1+V}\left(\frac{1|9}{9}\right)$ (*Figure 7C*). Two factors give rise to this feature: (i) Ternary complexes of type $\frac{1|8}{9}$ and $\frac{1|9}{9}$ have similar stability (*Figure 7E*), and (ii) the equilibrium concentration of free 8-mers is higher than that of 9-mers (*Figure 7D*). As a result, 8-mers are more likely than 9-mers to hybridize to an existing duplex $\frac{1}{9}$, and, given the stability of the complex $\frac{1|8}{9}$, 8-mers remain bound almost as stably as 9-mers. In summary, short oligomers sequester long oligomers as templates to enhance their monomer-extension rate, while long oligomers cannot make use of short oligomers as templates due to the relative instability of the corresponding complexes.

It is noteworthy that Ding et al. have already observed productivity inversion experimentally (*Ding et al., 2023*). In their study, they included activated monomers, activated imidazolium-bridged dinucleotides and oligomers up to a length of 12 nt and observed that the primer extension rate for short primers is higher than the extension rate of long primers. Even though our model differs from their setup in some aspects (e.g. different circular genome, no bridged dinucleotides), evaluating our model using parameters similar to those of the experimentally studied system predicts productivity inversion that qualitatively agrees with the experimental findings (Appendix 7). We therefore assume that the mechanism underlying productivity inversion described here also applies to the experimental observations.

## Discussion

While significant progress has been made in understanding the prebiotic formation of ribonucleotides (*Becker et al., 2016*; *Benner et al., 2012*; *Kim et al., 2011*; *Powner et al., 2009*) and characterizing ribozymes that might play a role in an RNA world (*Attwater et al., 2018*; *Mutschler et al., 2015*; *Pressman et al., 2019*; *Tjhung et al., 2020*), a convincing scenario bridging the gap between prebiotic chemistry and ribozyme-catalyzed replication is still missing. Here, we studied a scenario proposed by *Zhou et al., 2021* (the 'Virtual Circular Genome', VCG) using theoretical and computational approaches. We analyzed the process whereby template-directed ligation replicates the genomic information that is collectively stored in the VCG oligomers. Our analysis revealed a trade-off between the fidelity and the yield of this process: At low concentration of VCG oligomers, most of the ligations produce oligomers that are too short to specify a unique locus on the genome, resulting in a low yield of replication (*Figure 2B-C*). At high VCG concentration, erroneous templated ligations cause sequence scrambling and consequently limit the fidelity of replication (*Figure 2C-D*). We considered two solutions to these issues: (i) a VCG pool composition that optimizes its replication

behavior within the bounds of the fidelity-yield trade-off, and (ii) breaking the fidelity-yield trade-off given that error-prone ligations can be kinetically suppressed.

The first solution maximizes the yield of replication for fixed fidelity. In pools containing only monomers and VCG oligomers of a single length, replication efficiency can be maximized by increasing the length of VCG oligomers and decreasing their concentration (*Figure 2F-G*). This reduces the likelihood of error-prone templated ligation of long oligomers. When the pool contains VCG oligomers of multiple lengths, replication efficiency is typically governed by the longest oligomer in the pool (*Figure 4D*). Including dimers as feedstock for the replication increases the error fraction (*Figure 5B-C*), as dimers that bind to a template with a dangling end are prone to form an incorrect product (*Figure 5D*).

The second solution eliminates the error-prone templated ligation of two VCG oligomers by suppressing them kinetically, for example by assuring that only monomers are chemically activated. This enables both fidelity and yield to remain high at high VCG concentrations (*Figure 6B*), effectively breaking the fidelity-yield trade-off. Longer VCG oligomers are then more likely to be extended than shorter oligomers at equal concentration (*Figure 6C*). However, this is only true for pools with VCG oligomers of a single length — once multiple VCG oligomer lengths compete with each other, shorter oligomers can be more productive than longer ones (*Figure 7B*). This feature, which has also been observed experimentally (*Ding et al., 2023*), is caused by an asymmetry in the productive interaction between short and long oligomers (*Figure 7C*): While short oligomers can sequester longer oligomers as templates for their extension by a monomer, short oligomers are unlikely to serve as templates for longer oligomers (*Figure 7D-E*).

As we intended to study the pathways responsible for sequence scrambling and to explore possible mitigation strategies, we based our analysis on a coarse-grained model that neglects some experimental details. First, we assumed that a complex instantaneously dehybridizes if it contains a non-complementary base pair, whereas in reality, short duplexes can tolerate a limited number of mismatches (*Todisco et al., 2024a*). While such mismatches can facilitate incorrect hybridization and introduce additional replication errors, we expect this effect to be moderate: Mismatches preferentially occur near the ends of the hybridized region, where their destabilizing effect on binding is weakest (*Todisco et al., 2024a*). However, such terminal mismatches have also been shown to significantly reduce ligation rates (*Rajamani et al., 2010 Leu et al., 2013*), which in turn limits the likelihood of forming incorrect products.

Second, we simplified the hybridization dynamics by assuming that all oligomers bind to each other at equal rates, and that dehybridization rates are determined by the hybridization energy computed via a nearest-neighbor model. However, recent work has shown that hybridization to a gap flanked by two oligomers proceeds more slowly than binding to an unoccupied template. Moreover, the resulting nicked complexes (two oligomers hybridized adjacently on a template) are more stable than predicted by standard nearest-neighbor models due to enhanced stacking interactions at the nick site (*Todisco et al., 2024b*). While this added stability is not expected to affect overall replication efficiency of the VCG (since all productive complexes, correct or incorrect, contain a nick), it can impact the kinetics of the system. In particular, the extended lifetime of such complexes may challenge the adiabatic approximation used in much of our analysis, which assumes ligation is always slower than hybridization and dehybridization.

Third, we do not model the activation chemistry explicitly, but instead assume that all monomers (and, depending on the scenario, also oligomers) are always reactive. As a result, some activated intermediates that are known to form in experiments, such as imidazolium-bridged dinucleotides (*Walton and Szostak, 2016*), are not modeled. Nonetheless, we include aspects of activation chemistry in a coarse-grained manner. Specifically, to capture the experimentally observed difference in reactivity between monomer incorporation and templated ligation of oligomers under amino-imidazolium activation, we introduce two distinct ligation rate constants. With this approach, we describe the experimental setup well enough to qualitatively reproduce features observed in experiments, for example, the preferential extension of shorter oligomers by monomers in pools containing VCG oligomers of varying lengths (*Ding et al., 2023*).

The VCG scenario was proposed to close the gap between prebiotic chemistry and ribozyme-catalyzed replication. To this end, VCG pools need to be capable of replicating (parts of) ribozymes that play a role in the emergence of life. While there are cases of small ribozymes (*Pressman et al.,*

*2019*) or ribozymes with small active sites (e.g. the Hammerhead ribozyme *Scott, 2013*), ribozymes obtained experimentally via in vitro evolution are often more than a hundred nucleotides long (*Attwater et al., 2013*; *Johnston et al., 2001*; *Müller and Bartel, 2008*; *Wochner et al., 2011*). Remarkably, our model suggests that the VCG scenario enables high-fidelity replication of long genomes, even in pools containing relatively short VCG oligomers. For a genome of length $L_G$, a sequence of at least $L_U^{min} = \left\lceil \frac{\ln(2L_G)}{\ln 4} \right\rceil$ nucleotides is required to uniquely specify a position within the genome. If the oligomers in the pool exceed this length by about three nucleotides, accurate replication becomes feasible (*Figure 2H*). For example, genomes of length $L_G = 1000 \, \text{nt}$ ($L_U^{min} = 6, \text{nt}$) can be replicated in VCG pools containing $10 \, \text{nt}$ oligomers. However, $L_U$ equals $L_U^{min}$ only for a restricted set of genome sequences; more generally, $L_U > L_U^{min}$. In such cases, reliable replication requires correspondingly longer oligomers. While the precise margin between oligomer length and $L_U$ depends on genome-specific features (particularly the motif distribution at sub-$L_U$ scales), it typically amounts to only a few additional nucleotides (Appendix 2).

It is noteworthy that replication in the VCG scenario imposes a selection pressure on prebiotic genomes to reduce their unique motif length, $L_U$. A circular genome requiring many nucleotides to specify a unique locus (high $L_U$) replicates less efficiently than one with a shorter $L_U$, assuming all other properties of the VCG pool (particularly the oligomer length distribution) remain identical. This length distribution arises from the interplay between the chemical kinetics and molecular transport governed by the physical environment. For instance, templated ligation in an open system with continuous oligomer influx and outflux can produce a non-monotonic length distribution, with a concentration peak at a characteristic oligomer length $L_c$, determined by the interplay between dehybridization and outflux (*Rosenberger et al., 2021*). Through this emergent length scale, the environment shapes replication in the VCG scenario. If the environment facilitates long oligomers ($L_c > L_U$), replication proceeds efficiently. Conversely, in environments with a small $L_c$, repeating motifs longer than $L_c$ are selected against. In such cases, mutational errors may replace long repeated motifs with functionally equivalent sequences composed of shorter unique motifs, thereby increasing replication efficiency.

Given the broad range of prebiotically plausible non-equilibrium environments (*Ianeselli et al., 2023*), it is reasonable to expect that some environments provide the required conditions for efficient replication. The constraints formulated in this work can help to guide the search for self-replicating oligomer pools, in the vast space of possible concentration profiles and non-equilibrium environments.

## Methods
### Constructing circular genomes
In the Virtual Circular Genome (VCG) scenario, genomes are encoded in a pool of oligomers. The encoded genomes are assumed to be circular sequences of length $L_G$, containing both the original sequence and its reverse complement. Each genome is characterized by two fundamental length scales that reflect different aspects of motif distribution along the sequence. The minimal unique motif length, $L_U$, is defined as the shortest subsequence length for which all motifs of length $L \geq L_U$ appear at most once in the genome. In contrast, the exhaustive coverage length, $L_E$, denotes the largest motif length for which all $4^{L_E}$ possible motifs are present within the genome. Since only $2L_G$ distinct motifs can be encoded in a genome (including its complement), $L_E$ cannot exceed

$$L_E^{max} = \left\lfloor \frac{\ln(2L_G)}{\ln 4} \right\rfloor.$$

Similarly, for a motif to be uniquely addressable on the genome, its length must be at least

$$L_U^{min} = \left\lceil \frac{\ln(2L_G)}{\ln 4} \right\rceil.$$

We note that $L_E^{max}$ and $L_U^{min}$ are essentially the same length scale (differing by at most one nucleotide).

The characteristic length scales $L_E$ and $L_U$ impose constraints on how motifs are distributed. For example, when $L = L_U$, all motifs of length $L$ must appear at most once, while at least one motif of length $L - 1$ must occur more than once. To quantify the motif distribution, we introduce the motif entropy,

$$S(L) = - \sum_{i=1}^{4^L} f_i \ln f_i,$$

where $f_i$ denotes the relative frequency of motif $i$ across the genome and its reverse complement. Motif entropy ranges from zero (a homogeneous sequence with only one motif) to a maximum value that depends on the subsequence length,

$$S^{\text{max}}(L) = \begin{cases} L & \text{if } L \leq L_{\text{G}}, \\ \dfrac{\ln(2L_{\text{G}})}{\ln 4} & \text{if } L > L_{\text{G}}. \end{cases}$$

For motif length $L$ to qualify as the unique motif length $L_{\text{U}}$, its entropy must be maximal, $S(L) = S^{\text{max}}(L)$, while $S(L-1)$ must be smaller than its respective maximum, $S(L-1) < S^{\text{max}}(L-1)$.

The correspondence between characteristic length scales and motif entropies provides a way to construct genome sequences with specified motif characteristics. By treating the entropy function as an effective 'Hamiltonian' $\mathcal{H}$, we can generate genome sequences through Metropolis–Hastings sampling. In our implementation, each update step in the Metropolis-Hastings algorithm involves either a single-nucleotide mutation or a cut-and-paste operation that relocates a segment of the genome to a new position (the cut-and-paste operation is 10 times more likely than the single nucleotide mutation). The acceptance criterion follows the standard Metropolis rule: modifications that reduce the Hamiltonian are always accepted, while increases in 'energy' are accepted with probability $\exp\left[-\beta(E_{\text{old}} - E_{\text{new}})\right]$. Here, $\beta^{-1}$ is an effective temperature chosen to be small compared to the typical energy to ensure convergence to the minimum. Simulations are either run until a predefined entropy target is reached or until the energy converges to a plateau.

To generate genomes with $L_{\text{E}} = L_{\text{E}}^{\text{max}}$ and $L_{\text{U}} = L_{\text{U}}^{\text{min}}$, we minimize the Hamiltonian

$$\mathcal{H} = -S(L_{\text{E}}) - S(L_{\text{U}})$$

Starting from a random sequence, we perform 10,000 Metropolis–Hastings steps at an inverse temperature $\beta = 10^{-5}$ to construct genome sequences of lengths $L_{\text{G}} = 16\,\text{nt}$ (listed in Results) and $L_{\text{G}} = 64\,\text{nt}$ (listed in **Supplementary file 1**). To explore genomes with $L_{\text{E}} < L_{\text{E}}^{\text{max}}$ and $L_{\text{U}} > L_{\text{U}}^{\text{min}}$, we initialize the algorithm from the maximally diverse genomes ($L_{\text{E}} = L_{\text{E}}^{\text{max}}$, $L_{\text{U}} = L_{\text{U}}^{\text{min}}$) and then reduce entropy across the range $L_{\text{E}} < L < L_{\text{U}}$. This is done by minimizing the Hamiltonian

$$\mathcal{H} = \sum_{L=L_{\text{E}}}^{L_{\text{U}}} S(L),$$

via two different sampling protocols. In the first protocol, the simulation is terminated as soon as the genome reaches the desired values of $L_{\text{E}}$ and $L_{\text{U}}$. The resulting motif distributions on the intermediate length scales ($L_{\text{E}} < L < L_{\text{U}}$) remain close to uniform, with only minor biases sufficient to enforce the length-scale constraints. In the second protocol, entropy minimization continues beyond the point at which the target values are achieved, leading to more strongly biased motif distributions on intermediate length scales. These construction strategies allow us to systematically tune genome complexity and motif structure, enabling controlled investigations of how the characteristic length scales influence replication dynamics (see Appendix 2 for details).

## Computing replication observables based on the kinetic simulation

We simulate the dynamics of VCG pools using a kinetic simulation that is based on the Gillespie algorithm. In the simulation, oligomers can hybridize to each other to form complexes or dehybridize from an existing complex. Moreover, two oligomers can undergo templated ligation if they are hybridized adjacent to each other on a third oligomer. At each time $t$, the state of the system is determined by a list of all single-stranded oligomers and complexes as well as their respective copy number. We refer to the state of the system at the time $t$ as the ensemble of compounds $\mathcal{E}_t$. Given the copy numbers, the rates $r_i$ of all possible chemical reactions $i \in \mathcal{I}$ can be computed. To evolve the system in time, we need to perform two steps: (i) We sample the waiting time until the next reaction, $\tau$, from an exponential distribution with mean $(\sum_{i \in \mathcal{I}} r_i)^{-1}$, and update the simulation time, $t \to t + \tau$. (ii) We pick which

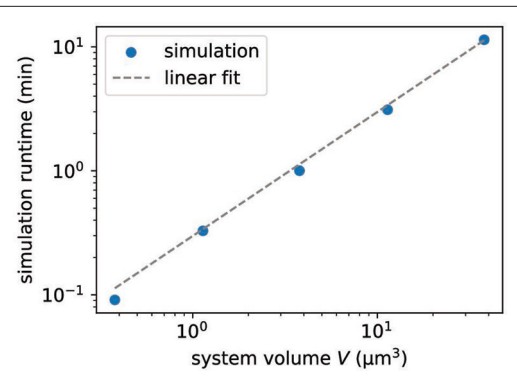

**Figure 8.** Simulation runtime of the full kinetic simulation for a VCG pool that includes monomers and VCG oligomers of length $L = 8$. The total concentration of feedstock monomers equals $c_F^{tot} = 0.1$ mM, while the total concentration of VCG oligomers is $c_V^{tot} = 1$ µM. The energy contribution per matching nearest-neighbor block is set to $\gamma = -2.5\, k_B T$. The volume of the system is varied, and the time evolution is simulated until $t = 5.0 \cdot 10^7 t_0$. The runtime of the simulation scales linearly with the volume of the system.

reaction to perform by sampling from a categorical distribution. Here, the probability to pick reaction $i$ equals $r_i/(\sum_{i \in \mathcal{I}} r_i)$. The copy numbers are updated according to the sampled reaction, yielding $\mathcal{E}_{t+\tau}$. Steps (i) and (ii) are repeated until the simulation time $t$ reaches the desired final time, $t_{final}$. A more detailed explanation of the kinetic simulation is presented in *Göppel et al., 2022*; *Rosenberger et al., 2021*.

Our goal is to compute observables characterizing replication in the VCG scenario based on the full kinetic simulation. In the following derivation, we focus on one particular observable (yield) for clarity. The results for other observables are stated directly, as their derivations follow analogously. Recall the definition of the yield introduced in the Results section,

$$y = \frac{\# \text{ nucleotides incorporated in VCG oligomers until } \tau_{lig}}{\# \text{incorporated nucleotides until } \tau_{lig}}.$$

As we are interested in the initial replication performance of the VCG, we compute the yield based on the ligation events that take place until the characteristic timescale of ligations $\tau_{lig} = k_{lig}^{-1} \approx 10^{12}\, t_0$. In principle, we would like to compute the yield based on the templated ligation events that we observe in the simulation. Unfortunately, for reasonable system parameters, it is impossible to simulate the system long enough to observe sufficiently many ligation events to compute $y$ to reasonable accuracy. For example, for a VCG pool containing monomers at a total concentration of $c_F^{tot} = 0.1$ mM and VCG oligomers of length $L = 8$ nt at a total concentration of $c_V^{tot} = 1$ µM, it would take about 1700 hr of simulation time to reach $t = 5 \cdot 10^{12} t_0$ (*Figure 8*). Multiple such runs would be needed to estimate the mean and the variance of the observables of interest, rendering this approach unfeasible.

Instead, we compute the replication observables based on the copy number of complexes that could potentially perform a templated ligation, that is complexes in which two strands are hybridized adjacent to each other, such that they could form a covalent bond. We can show analytically that the number of productive complexes is a good approximation for the number of incorporated nucleotides: The number of incorporated nucleotides can be computed as the integral over the ligation flux, weighted by the number of nucleotides that are added in each templated ligation reaction,

$$(\# \text{ incorporated nucleotides until } \tau_{lig}) = \int_0^{\tau_{lig}} dt \sum_{C \in \mathcal{E}_t} N(C)\min(L_{e,1}, L_{e,2})\, \mathbb{1}(C \text{ allows templated ligation}).$$

Here, $N(C)$ denotes the copy number of the complex $C$ in the pool $\mathcal{E}_t$. $L_{e,1}$ and $L_{e,2}$ denote the lengths of the oligomers that undergo ligation, and $\mathbb{1}$ is an indicator function which enforces that only complexes in a ligation-competent configuration contribute to the reaction flux. As only a few ligation events are expected to happen until $\tau_{lig}$, it is reasonable to assume that the ensembles $\mathcal{E}_t$ do not change significantly during $t \in [0, \tau_{lig}]$. Therefore, the integration over time may be interpreted as a multiplication by $\tau_{lig}$,

$$(\# \text{ incorporated nucleotides until } \tau_{lig}) \approx \tau_{lig} \left\langle \sum_{C \in \mathcal{E}} N(C)\min(L_{e,1}, L_{e,2})\mathbb{1}(C \text{ allows templated ligation}) \right\rangle, \quad (6)$$

where $\langle \dots \rangle$ denotes the average over realizations of the ensembles $\mathcal{E}_t$ within the time interval $t \in [\tau_{eq}, \tau_{lig}]$. This average corresponds to the average number of complexes in a ligation-competent configuration. Note that, at this point, we made the additional assumption that no templated ligations are taking place between $[0, \tau_{eq}]$. This assumption is reasonable, as (i) the equilibration process is very

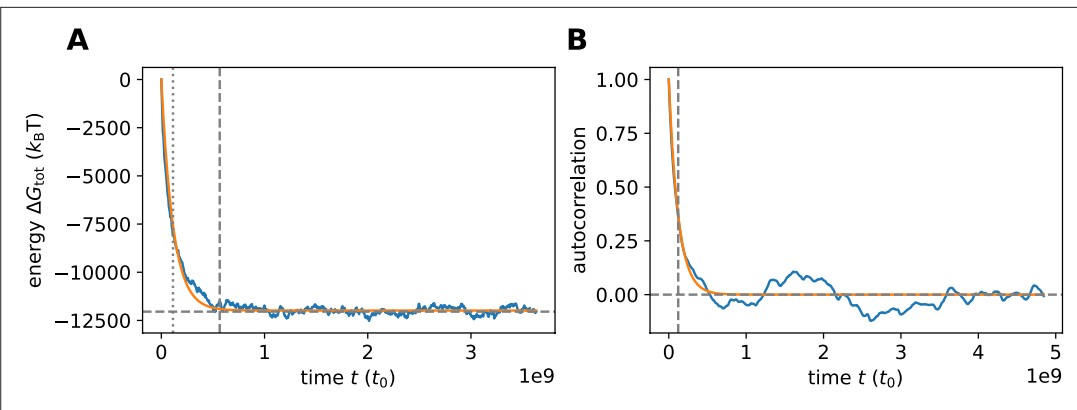

**Figure 9.** Characteristic timescales in the kinetic simulation. (**A**) The equilibration timescale is determined based on the total hybridization energy of all strands in the pool, $\Delta G_{\text{tot}}$. By fitting an exponential function to $\Delta G_{\text{tot}}$, we obtain a characteristic timescale $\tau^*$ (vertical dotted line), which is then used to calculate the equilibration time as $\tau_{\text{eq}} = 5\tau^*$ (vertical dashed line). The horizontal dashed line shows the total hybridization energy expected in (de)hybridization equilibrium according to the coarse-grained adiabatic approach (Methods). (**B**) The correlation timescale is determined based on the autocorrelation of $\Delta G_{\text{tot}}$. We obtain $\tau_{\text{corr}}$ (vertical dashed line) by fitting an exponential function to the autocorrelation. In both panels, we show simulation data obtained for a VCG pool containing monomers and VCG oligomers with a concentration of $c_{\text{F}}^{\text{tot}} = 0.1\text{ mM}$ as well as oligomers of length $L = 8\text{ nt}$ with a concentration of $c_{\text{V}}^{\text{tot}} = 1\,\mu\text{M}$.

short compared to the characteristic timescale of ligation, and (ii) the number of complexes that might allow for templated ligation during equilibration is lower than in equilibrium (we start the simulation with an ensemble of single-stranded oligomers). Both aspects imply that the rate of templated ligation is negligible during the interval $[0, \tau_{\text{eq}}]$.

In order to compute the average over different realizations of ensembles $\mathcal{E}$ (as required in *Equation 6*), we need to sample a set of uncorrelated ensembles that have reached the hybridization equilibrium, which can be done using the full kinetic simulation. The simulation starts with a pool containing only single-stranded oligomers and reaches the (de)hybridization equilibrium after a time $\tau_{\text{eq}}$. We identify this timescale of equilibration by fitting an exponential function to the total hybridization energy of all complexes in the system, $\Delta G_{\text{tot}}$ (*Figure 9A*). In the set of ensembles used to evaluate the average in *Equation 6*, we only include ensembles for time $t > \tau_{\text{eq}}$ to ensure that the ensembles have reached (de)hybridization equilibrium. To ensure that the ensembles are uncorrelated, we require that the time between two ensembles that contribute to the average is at least $\tau_{\text{corr}}$. The correlation time, $\tau_{\text{corr}}$, is determined via an exponential fit to the autocorrelation function of $\Delta G_{\text{tot}}$ (*Figure 9B*). Besides computing the expectation value (*Equation 6*), we are also interested in the 'uncertainty' of this expectation value, that is in the standard deviation of the sample mean $\sigma_{\langle X \rangle}$. (We use $X$ as a short-hand notation for $\sum_{\text{C} \in \mathcal{E}} N(\text{C}) \min(L_{\text{e},1}, L_{\text{e},2}) \mathbb{1}(\text{C allows templated ligation})$). The standard deviation of the sample mean, $\sigma_{\langle X \rangle}$, is related to the standard deviation of $X$, $\sigma_X$, by the number of samples, $\sigma_{\langle X \rangle} = (N_s)^{-1/2} \sigma_X$. Moreover, based on the van-Kampen system size expansion, we expect the standard deviation of $X$ to be proportional to $V^{-1/2}$, such that $\sigma_{\langle X \rangle} \propto (N_s V)^{-1/2}$.

Using *Equation 6* (as well as an analogous expression for the number of nucleotides that are incorporated in VCG oligomers), the yield can be expressed as

$$y = \frac{\left\langle \sum_{\text{C} \in \mathcal{E}} N(\text{C}) \min(L_{\text{e},1}, L_{\text{e},2}) \mathbb{1}(\text{C allows templated ligation}) \mathbb{1}(L_{\text{e},1} + L_{\text{e},2} \geq L_{\text{U}}) \right\rangle}{\left\langle \sum_{\text{C} \in \mathcal{E}} N(\text{C}) \min(L_{\text{e},1}, L_{\text{e},2}) \mathbb{1}(\text{C allows templated ligation}) \right\rangle}.$$

The additional condition $\mathbb{1}(L_{\text{e},1} + L_{\text{e},2} \geq L_{\text{U}})$ in the numerator ensures that the product oligomer is long enough to be counted as a VCG oligomer, that is at least $L_{\text{U}}$ nucleotides long. Analogously, the expression for the fidelity of replication reads

$$f = \frac{\left\langle \sum_{C \in \mathcal{E}} N(C) \min(L_{e,1}, L_{e,2}) \mathbb{1}(C \text{ allows templated ligation}) \mathbb{1}(L_{e,1} + L_{e,2} \geq L_U) \mathbb{1}(\text{product correct}) \right\rangle}{\left\langle \sum_{C \in \mathcal{E}} N(C) \min(L_{e,1}, L_{e,2}) \mathbb{1}(C \text{ allows templated ligation}) \mathbb{1}(L_{e,1} + L_{e,2} \geq L_U) \right\rangle}.$$

Multiplying fidelity and yield results in the efficiency of replication,

$$\eta = \frac{\left\langle \sum_{C \in \mathcal{E}} N(C) \min(L_{e,1}, L_{e,2}) \mathbb{1}(C \text{ allows templated ligation}) \mathbb{1}(L_{e,1} + L_{e,2} \geq L_U) \mathbb{1}(\text{product correct}) \right\rangle}{\left\langle \sum_{C \in \mathcal{E}} N(C) \min(L_{e,1}, L_{e,2}) \mathbb{1}(C \text{ allows templated ligation}) \right\rangle}.$$

The ligation share of a particular type of templated ligation $s(\text{type})$, that is, the relative contribution of this templated-ligation type to the nucleotide extension flux, can be represented in a similar form as the other observables,

$$s(\text{type}) = \frac{\left\langle \sum_{C \in \mathcal{E}} N(C) \min(L_{e,1}, L_{e,2}) \mathbb{1}(C \text{ allows templated ligation of given type}) \right\rangle}{\left\langle \sum_{C \in \mathcal{E}} N(C) \min(L_{e,1}, L_{e,2}) \mathbb{1}(C \text{ allows templated ligation}) \right\rangle}.$$

As all observables are expressed as the ratio of two expectation values, $\mathcal{Z} = \langle X \rangle / \langle Y \rangle$, we can compute the uncertainty of the observables via Gaussian error propagation,

$$\sigma_{\mathcal{Z}} = \sqrt{\frac{\sigma^2_{\langle X \rangle}}{\langle Y \rangle^2} + \frac{\langle X \rangle^2 \, \sigma^2_{\langle Y \rangle}}{\langle Y \rangle^4} - \frac{2 \langle X \rangle \, \sigma^2_{\langle X \rangle, \langle Y \rangle}}{\langle Y \rangle^3}}.$$

Since the variances, $\sigma^2_{\langle X \rangle}$ and $\sigma^2_{\langle Y \rangle}$, as well as the covariance, $\sigma^2_{\langle X \rangle, \langle Y \rangle}$, are proportional to $(N_s V)^{-1}$, the standard deviation of the observable mean, $\sigma_Z$, scales with the inverse square root of the number of samples and the system volume, that is $\sigma_Z \propto (N_s V)^{-1/2}$. Therefore, the variance of the computed observable can be reduced by either increasing the system volume or increasing the number of samples used for averaging. Both approaches incur the same computational cost: (i) Increasing the

**Table 1.** Input parameters and resulting observables (yield and efficiency) from the full kinetic simulation of replication in pools containing monomers and VCG oligomers of a single length $L_V$. The observables (yield and efficiency) listed in this table are shown in **Figure 2**.

| VCG oligo. length | conc. ratio $c_V^{tot}/c_F^{tot}$ | volume | equilibration time | correlation time | number of samples | yield $y$ | efficiency $\eta$ |
|---|---|---|---|---|---|---|---|
| 6 | $1.0 \cdot 10^{-4}$ | $5.0 \cdot 10^4$ | $3.4 \cdot 10^6$ | $1.9 \cdot 10^6$ | 3805 | $0.04 \pm 0.01$ | $0.04 \pm 0.01$ |
| 6 | $1.0 \cdot 10^{-3}$ | $5.0 \cdot 10^3$ | $1.2 \cdot 10^7$ | $2.6 \cdot 10^6$ | 3264 | $0.38 \pm 0.02$ | $0.36 \pm 0.02$ |
| 6 | $3.3 \cdot 10^{-3}$ | $8.0 \cdot 10^2$ | $1.3 \cdot 10^7$ | $2.7 \cdot 10^6$ | 5400 | $0.68 \pm 0.02$ | $0.64 \pm 0.02$ |
| 6 | $1.0 \cdot 10^{-2}$ | $9.1 \cdot 10^1$ | $1.4 \cdot 10^7$ | $2.7 \cdot 10^6$ | 5440 | $0.87 \pm 0.01$ | $0.77 \pm 0.03$ |
| 6 | $3.3 \cdot 10^{-2}$ | $9.1 \cdot 10^0$ | $1.3 \cdot 10^7$ | $2.4 \cdot 10^6$ | 6170 | $0.96 \pm 0.01$ | $0.63 \pm 0.03$ |
| 7 | $1.0 \cdot 10^{-4}$ | $3.9 \cdot 10^4$ | $1.7 \cdot 10^8$ | $2.6 \cdot 10^7$ | 784 | $0.33 \pm 0.05$ | $0.33 \pm 0.05$ |
| 7 | $1.0 \cdot 10^{-3}$ | $7.6 \cdot 10^2$ | $1.9 \cdot 10^8$ | $4.0 \cdot 10^7$ | 2041 | $0.87 \pm 0.02$ | $0.81 \pm 0.05$ |
| 7 | $3.3 \cdot 10^{-3}$ | $7.7 \cdot 10^1$ | $1.9 \cdot 10^8$ | $3.3 \cdot 10^7$ | 2980 | $0.95 \pm 0.01$ | $0.87 \pm 0.04$ |
| 7 | $1.0 \cdot 10^{-2}$ | $1.1 \cdot 10^1$ | $1.9 \cdot 10^8$ | $2.6 \cdot 10^7$ | 3465 | $0.99 \pm 0.01$ | $0.81 \pm 0.05$ |
| 7 | $3.3 \cdot 10^{-2}$ | $1.7 \cdot 10^0$ | $1.9 \cdot 10^8$ | $3.1 \cdot 10^7$ | 3235 | $0.99 \pm 0.04$ | $0.73 \pm 0.05$ |
| 8 | $1.0 \cdot 10^{-4}$ | $6.3 \cdot 10^3$ | $2.5 \cdot 10^9$ | $1.1 \cdot 10^8$ | 466 | $0.81 \pm 0.05$ | $0.81 \pm 0.05$ |
| 8 | $1.0 \cdot 10^{-3}$ | $9.9 \cdot 10^1$ | $1.9 \cdot 10^9$ | $3.6 \cdot 10^8$ | 615 | $0.99 \pm 0.01$ | $0.99 \pm 0.01$ |
| 8 | $3.3 \cdot 10^{-3}$ | $1.6 \cdot 10^1$ | $1.0 \cdot 10^9$ | $2.2 \cdot 10^8$ | 1100 | $0.95 \pm 0.03$ | $0.95 \pm 0.03$ |
| 8 | $1.0 \cdot 10^{-2}$ | $3.8 \cdot 10^0$ | $5.6 \cdot 10^8$ | $1.4 \cdot 10^8$ | 1700 | $1.00 \pm 0.01$ | $0.93 \pm 0.05$ |
| 8 | $3.3 \cdot 10^{-2}$ | $0.9 \cdot 10^0$ | $4.9 \cdot 10^8$ | $7.4 \cdot 10^7$ | 3195 | $1.00 \pm 0.03$ | $0.82 \pm 0.05$ |

number of samples, $N_s$, requires running the simulation for a longer duration, with the additional runtime scaling linearly with the number of samples. (ii) Similarly, the additional runtime needed due to increased system volume, $V$, also scales linearly with $V$ (*Figure 8*). One update step in the simulation always takes roughly the same amount of runtime, but the change in simulation time per update step depends on the total rate of all reactions in the system. The total rate is dominated by the association reactions, and their rate is proportional to the volume. Therefore, the change in simulation time per update step is proportional to $V^{-1}$. The runtime, which is necessary to reach the same simulation time in a system with volume $V$ as in a system with volume 1, is a factor of $V$ longer in the larger system. With this in mind, it makes no difference whether the variance is reduced by increasing the volume or the number of samples. For practical reasons (post-processing of the simulations is less memory- and time-consuming), we opt to choose a moderate number of samples, but slightly higher system volumes to compute the observables of interest. The simulation parameters (length of oligomers, concentrations, hybridization energy, volume, number of samples, characteristic timescales) used to obtain the results presented in *Figure 2* are summarized in *Table 1*.

## Coarse-grained representation of complexes in the adiabatic approach

To characterize the replication performance of VCG pools across a broad range of system parameters, we developed an adiabatic approach that enables faster computation of replication observables than full kinetic simulations. In this method, we compute the equilibrium concentrations of complexes in productive configurations under the assumption that templated ligation is much slower than (de) hybridization. The approach relies on a coarse-grained representation of the oligomers in the pool, which we introduce in this section.

### Single strands

In the coarse-grained description, oligomers of identical length are assumed to have equal concentration, irrespective of their sequence. This assumption is justified for two reasons: (i) We initialize the VCG pool without sequence bias, that is all oligomers compatible with the genome sequence are included at equal concentration. (ii) Hybridization energy in our simplified energy model (and therefore also the stability of complexes) only depends on the length of the hybridization site, not on its sequence, such that oligomers of equal length have similar probabilities of being free or bound in complexes. Under this assumption, each coarse-grained oligomer is uniquely identified by its length $L$, and it represents a group of oligomers with $\mathfrak{C}(L)$ distinct sequences. We refer to the number $\mathfrak{C}(L)$ as the combinatorial multiplicity of the coarse-grained oligomer. The value of $\mathfrak{C}(L)$ depends on the choice of the encoded genome. We assume that by construction, all possible oligomer sequences of length $L \leq L_\mathrm{E}^\mathrm{max}$ are included in the genome (see Methods for the definition of $L_\mathrm{E}^\mathrm{max}$ and $L_\mathrm{U}^\mathrm{min}$). For $L \geq L_\mathrm{U}^\mathrm{min} = L_\mathrm{E}^\mathrm{max} + 1$, only a subset of all possible $4^L$ sequences is included, but no sequence is repeated multiple times across the genome. Therefore, the combinatorial multiplicity equals

$$\mathfrak{C}(L) = \begin{cases} 4^L & \text{if } L < L_\mathrm{U}^\mathrm{min}, \\ 2L_\mathrm{G} & \text{if } L \geq L_\mathrm{U}^\mathrm{min}. \end{cases}$$

### Duplexes

Two strands can form a duplex by hybridizing to each other. We refer to the bottom oligomer as 'Crick' strand C and to the top oligomer as 'Watson' strand W. A duplex is uniquely characterized by the lengths of the oligomers, $L_\mathrm{C}$ and $L_\mathrm{W}$, as well as their relative alignment (*Figure 10A*). The alignment index $i$ denotes the position of the Watson strand with respect to the Crick strand. As there needs to be at least one nucleotide of overlap between the strands for a duplex to exist, the alignment index needs to be in the interval $i \in [-(L_\mathrm{W} - 1), L_\mathrm{C} - 1]$. Using the alignment index, we can also determine if the duplex has a left (or right) dangling end. The corresponding indicator variables are called $d_l$ (or $d_r$),

$$d_l = \begin{cases} 1 & \text{if } i = 0, \\ 0 & \text{otherwise,} \end{cases} \qquad d_r = \begin{cases} 1 & \text{if } i + L_\mathrm{W} = L_\mathrm{C}, \\ 0 & \text{otherwise.} \end{cases}$$

**Figure 10.** Schematic representation of complexes considered in the adiabatic approach. (**A**) A duplex is comprised of two strands, which we refer to as W (Watson) and C (Crick) strands. The relative position of the strands is characterized by the alignment index $i$; for the depicted duplex, $i = -2$. The length of the hybridization region is called $L_o$. (**B**) A ternary complex contains three strands. By convention, we denote the two strands that are on the same 'side'' of the complex as W1 and W2, and the complementary strand as C. The alignment indices $i$ and $j$ denote the positions of W1 and W2 relative to C. For the depicted complex, $i = -2$ and $j = 3$. The length of the hybridization regions is called $L_{o,1}$ and $L_{o,2}$.

Moreover, the length of the hybridization region $L_o$ can be computed via

$$L_o = \min(L_C, i + L_W) - \max(i, 0).$$

The hybridization energy of the duplex depends on the length of the hybridization region as well as on the existence/absence of dangling ends. For a hybridization site of length $L_o$, there are $L_o - 1$ nearest-neighbor energy blocks each of which contributes $\gamma$ to the energy. Moreover, each dangling end contributes $\frac{\gamma}{2}$ to the energy,

$$E = \gamma(L_o - 1) + \frac{\gamma}{2}(d_l + d_r).$$

To compute the combinatorial multiplicity for a duplex with fixed $L_C$, $L_W$ and alignment index $i$, we need to multiply the combinatorial multiplicity of the Crick strand by the number of possible hybridization partners. We assume that a hybridization partner is possible if its sequence is perfectly complementary to the lower strand within the hybridization region, whereas hybridization partners with mismatches are not accounted for. This is sensible as long as the energetic penalty for mismatches in the full kinetic simulation is sufficiently large to suppress mismatches. The number of possible hybridization partners is determined by the length of the overlap region $L_o$: If $L_o \geq L_U^{\min}$, the pool contains only one oligomer sequence that can act as hybridization partner by construction of the genome. For shorter hybridization regions, multiple hybridization partners might be possible. Their number is set by the combinatorial multiplicity of the Watson oligomer divided by the combinatorial multiplicity of the hybridization region,

$$\mathfrak{C}(L_C, L_W, i) = \begin{cases} \dfrac{\mathfrak{C}(L_C)\mathfrak{C}(L_W)}{4^{L_o}} & \text{if } L_o < L_U^{\min}, \\ \mathfrak{C}(L_C) & \text{if } L_o \geq L_U^{\min}. \end{cases}$$

To avoid double-counting, we only account for complexes in which the Crick strand is at least as long as the Watson strand, $L_C \geq L_W$, and multiply $\mathfrak{C}(L_W, L_C, i)$ by 1/2 if $L_W = L_C$.

### Ternary complexes

Ternary complexes, that is complexes comprised of three strands, are uniquely characterized by the length of the three oligomers, $L_C, L_{W,1}, L_{W,2}$, as well as their respective alignment (**Figure 10B**). The alignment index $i$ denotes the position of strand W1 relative to strand C. Analogously, $j$ denotes the relative position of W2 relative to oligomer C. Two strands that are hybridized to each other need to have a hybridization region of at least one nucleotide. Moreover, the strands W1 and W2 must not occupy the same position on the template strand C. Taking both requirements together, the alignment indices fall within the intervals,

$$i \in [-(L_{W,1} - 1), L_C - L_{W,1} - 1], \quad \text{and} \quad j \in [i + L_{W,1}, L_C - 1].$$

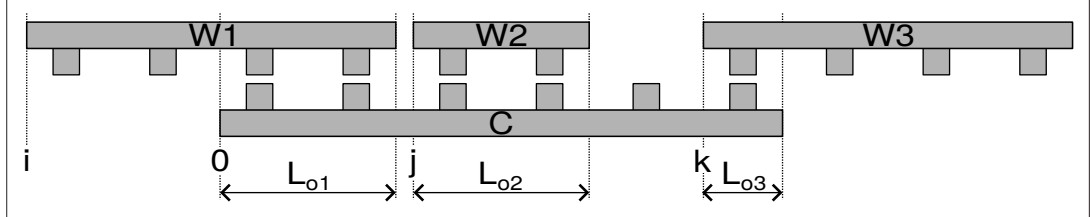

**Figure 11.** Schematic representation of a 3-1 quaternary complex. Three strands (referred to as Watson strands `W1`, `W2`, and `W3`) hybridize to a single template strand (Crick strand `C`). The positions relative to the left end of the `C` strand are given by the alignment indices $i, j$, and $k$; here, $i = -2, j = 2, k = 5$. The length of the overlap regions is denoted as $L_{o,1}, L_{o,2}$, and $L_{o,3}$.

A ternary complex may have a dangling end not only on its left or right end, but also in the gap between strands `W1` and `W2`. Three boolean variables are necessary to denote the presence/absence of the respective dangling ends,

$$d_l = \begin{cases} 0 & \text{if } i = 0, \\ 1 & \text{otherwise,} \end{cases} \qquad d_m = \begin{cases} 0 & \text{if } i + L_{W,1} = j, \\ 1 & \text{otherwise,} \end{cases} \qquad d_r = \begin{cases} 0 & \text{if } j + L_{W,2} = L_C, \\ 1 & \text{otherwise.} \end{cases}$$

The length of the two hybridization regions is given by

$$L_{o,1} = i + L_{W,1} - \max(i, 0), \quad \text{and} \quad L_{o,2} = \min(j + L_{W,2}, L_C) - j.$$

The hybridization energy depends on the length of the overlap regions as well as on the existence of dangling ends: As in the duplex, each overlap region of length $L_{o,1}$ (or $L_{o,2}$) comprises $L_{o,1} - 1$ (or $L_{o,2} - 1$) nearest neighbor blocks, each of which contributes $\gamma$ to the total energy. Moreover, every dangling end contributes $\gamma/2$. Note that the presence of a gap between strands `W1` and `W2`, that is $d_m = 1$, implies that there are two dangling ends, one for `W1` and another for `W2`. Gaps in between two complexes contribute $\gamma/2$ per each dangling end, adding up to $\gamma$. If there is no gap between the strands, that is $d_m = 0$, there are no dangling end contributions, but a new full nearest neighbor block emerges, which contributes $\gamma$ to the energy. Therefore, the total energy reads,

$$E = \gamma(L_{o,1} + L_{o,2} - 2) + \frac{\gamma}{2}(d_l + d_r + 2d_m) + \gamma(1 - d_m).$$

The combinatorial multiplicity of a ternary complex is computed in the same way as for the duplex: The combinatorial multiplicity of the strand `C` is multiplied by the number of possible hybridization partners `W1` and `W2`. Again, the number of possible partners is set by the length of the hybridization regions,

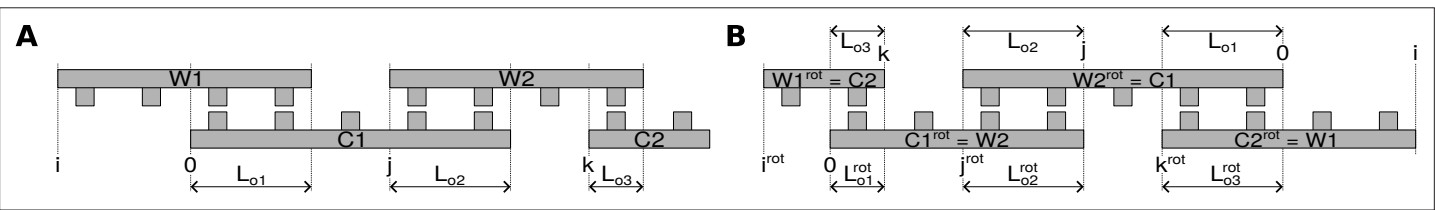

**Figure 12.** Schematic representation of a left-tilted 2-2 quaternary complex. (**A**) Two Watson strands (`W1` and `W2`) are hybridized to two Crick strands (`C1` and `C2`). Both Watson strands are hybridized to the left Crick strand `C1`, whereas only `W2` is hybridized to the right Crick strand `C2`. The alignment indices $i, j$ and $k$ denote the position of the strands relative to the left end of `C1`; here, $i = -2$, $j = 3$ and $k = 6$. The length of the hybridization regions is called $L_{o,1}, L_{o,2}$, and $L_{o,3}$. (**B**) Rotating the schematic representation of a left-tilted 2-2 quaternary complex by $180°$ produces an alternative representation of the same complex, which is again a left-tilted 2-2 complex. The panel depicts the rotated complex representation (variables with superscript 'rot') as well as the non-rotated representation (variables without superscript). There is a unique linear mapping between non-rotated and rotated representation, for example `C2` after rotation always corresponds to `W1` before rotation.

**Figure 13.** Schematic representation of a right-tilted 2-2 quaternary complex. (**A**) Two Watson strands (`W1` and `W2`) are hybridized to two Crick strands (`C1` and `C2`). Unlike in the left-tilted 2-2 quaternary complex, both Watson strands are hybridized to the right Crick strand `C2`, whereas only `W1` is hybridized to the left Crick strand `C1`. The alignment indices $i, j$, and $k$ denote the positions of the strands relative to `C1`; here, $i = 1$, $k = 3$, and $j = 6$. The length of the overlap regions is called $L_{o,1}, L_{o,2}$ and $L_{o,3}$. (**B**) Rotating the schematic representation of a right-tilted 2-2 quaternary complex by $180^\circ$ produces an alternative representation of the same complex, which is again a right-tilted 2–2 complex. The panel depicts the rotated complex representation (variables with superscript 'rot') as well as the non-rotated representation (variables without superscript). There is a unique linear mapping between non-rotated and rotated representation, for example `C2` after rotation always corresponds to `W1` before rotation. The mapping is identical for left- and right-tilted 2–2 quaternary complexes.

$$\mathfrak{C}(L_C, L_{W,1}, L_{W,2}, i, j) = \mathfrak{C}(L_C) \times \left[ \frac{\mathfrak{C}(L_{W,1})}{4^{L_{o,1}}} \mathbb{1}\left(L_{o,1} < L_U^{\min}\right) + \mathbb{1}\left(L_{o,1} \geq L_U^{\min}\right) \right]$$
$$\times \left[ \frac{\mathfrak{C}(L_{W,2})}{4^{L_{o,2}}} \mathbb{1}\left(L_{o,2} < L_U^{\min}\right) + \mathbb{1}\left(L_{o,2} \geq L_U^{\min}\right) \right].$$

We use $\mathbb{1}$ to denote the indicator function which returns 1 in case the condition in the bracket is fulfilled and zero otherwise. As all ternary complexes are asymmetric, there is no need to introduce a symmetry correction factor.

## Quaternary complexes

The largest complexes to be accounted for in our coarse-grained adiabatic approach are quaternary complexes, that is complexes comprised of four strands. We need to distinguish three types of such complexes: (i) 3-1 quaternary complexes, (ii) left-tilted 2-2 quaternary complexes, and (iii) right-tilted 2-2 quaternary complexes. In 3-1 quaternary complexes, three Watson strands are hybridized to one Crick strand (*Figure 11*), whereas in 2-2 quaternary complexes, two Watson strands are hybridized to two Crick strands (*Figure 12* and *Figure 13*).

### 3-1 Quaternary complexes

*Figure 11* depicts a typical 3-1 quaternary complex. Such a complex is uniquely characterized by the length of its oligomers, $L_C, L_{W,1}, L_{W,2}, L_{W,3}$, as well as their relative position to each other denoted by the alignment indices $i, j$, and $k$. All positions within the ternary complex are measured relative to the left end of the `C` strand. Any `W` strand needs to have at least one nucleotide of overlap with the `C` strand, but two `W` strands must never occupy the same position on the `C` strand. Consequently, the alignment indices fall within the intervals,

$$i \in [-(L_{W,1} - 1), L_C - L_{W,1} - L_{W,2} - 1], \quad j \in [i + L_{W,1}, L_C - L_{W,2} - 1], \quad \text{and} \quad k \in [j + L_{W,2}, L_C - 1].$$

There are two dangling ends (left and right) and potentially two gaps between the `W` strands: one gap between `W1` and `W2` and another one between `W2` and `W3`. The following boolean variables indicate the presence/absence of the respective dangling ends,

$$d_l = \begin{cases} 0 & \text{if } i = 0, \\ 1 & \text{otherwise,} \end{cases} \qquad d_{m1} = \begin{cases} 0 & \text{if } i + L_{W,1} = j, \\ 1 & \text{otherwise,} \end{cases}$$

$$d_{m2} = \begin{cases} 0 & \text{if } i + L_{W,2} = k, \\ 1 & \text{otherwise,} \end{cases} \qquad d_r = \begin{cases} 0 & \text{if } k + L_{W,3} = L_C, \\ 1 & \text{otherwise.} \end{cases}$$

The length of the hybridization regions is given by

$$L_{\text{o},1} = i + L_{\text{W},1} - \max(i, 0), \quad L_{\text{o},2} = L_{\text{W},2}, \quad \text{and} \quad L_{\text{o},3} = \min(L_C, k + L_{\text{W},3}) - k.$$

Following the same reasoning as in the case of ternary complexes, the energy equals

$$E = \gamma(L_{\text{o},1} + L_{\text{o},2} + L_{\text{o},3} - 3) + \gamma(1 - d_{m1}) + \gamma(1 - d_{m2}) + \frac{\gamma}{2}(d_l + d_r + 2d_{m1} + 2d_{m2}).$$

Similarly, the combinatorial multiplicity of 3-1 quaternary complexes is constructed using the same reasoning as in the case of ternary complexes,

$$\mathfrak{C}(L_C, L_{\text{W},1}, L_{\text{W},2}, L_{\text{W},3}, i, j, k) = \mathfrak{C}(L_C) \times \left[ \frac{\mathfrak{C}(L_{\text{W},1})}{4^{L_{\text{o},1}}} \mathbb{1}\left(L_{\text{o},1} < L_U^{\min}\right) + \mathbb{1}\left(L_{\text{o},1} \geq L_U^{\min}\right) \right]$$
$$\times \left[ \frac{\mathfrak{C}(L_{\text{W},2})}{4^{L_{\text{o},2}}} \mathbb{1}\left(L_{\text{o},2} < L_U^{\min}\right) + \mathbb{1}\left(L_{\text{o},2} \geq L_U^{\min}\right) \right]$$
$$\times \left[ \frac{\mathfrak{C}(L_{\text{W},3})}{4^{L_{\text{o},3}}} \mathbb{1}\left(L_{\text{o},3} < L_U^{\min}\right) + \mathbb{1}\left(L_{\text{o},3} \geq L_U^{\min}\right) \right].$$

As 3-1 quaternary complexes are not symmetric under rotation, no symmetry correction of the combinatorial multiplicity is necessary.

## Left-tilted 2-2 quaternary complexes

A 2-2 quaternary complex is comprised of two `C` strands and two `W` strands. We call a 2-2 complex left-tilted if strand `W1` is connected to strand `W2` via strand `C1` (**Figure 12A**). The lengths of the oligomers are called $L_{\text{W},1}, L_{\text{W},2}, L_{\text{C},1}$, and $L_{\text{C},2}$. The positions of the strands relative to each other are governed by the alignment indices. All positions are measured relative to the position of the left end of strand `C1`. The alignment indices may take on the following values,

$$i \in [-(L_{\text{W},1} - 1), L_{\text{C},1} - L_{\text{W},1} - 1], \quad j \in [i + L_{\text{W},1}, L_{\text{C},1}], \quad \text{and} \quad k \in [L_{\text{C},2}, j + L_{\text{W},2} - 1].$$

The complex can have dangling ends on the right and on the left end of the complex; the presence of these dangling ends is indicated by the boolean variables $d_l$ and $d_r$. Moreover, two gaps are possible: There might be a gap between strands `W1` and `W2`, or a gap between `C1` and `C2`. The respective boolean variables read

$$d_l = \begin{cases} 0 & \text{if } i = 0, \\ 1 & \text{otherwise,} \end{cases} \qquad d_{m1} = \begin{cases} 0 & \text{if } i + L_{\text{W},1} = j, \\ 1 & \text{otherwise,} \end{cases}$$

$$d_{m2} = \begin{cases} 0 & \text{if } L_{\text{C},1} = k, \\ 1 & \text{otherwise,} \end{cases} \qquad d_r = \begin{cases} 0 & \text{if } j + L_{\text{W},2} = k + L_{\text{C},2}, \\ 1 & \text{otherwise.} \end{cases}$$

We refer to the hybridization region of strand `W1` and `C1` as overlap region 1, to the hybridization region of strand `W2` and `C1` as overlap region 2 and to the hybridization region of strand `W2` and `C2` as overlap region 3. The length of these hybridization regions is computed via

$$L_{\text{o},1} = i + L_{\text{W},1} - \max(i, 0), \quad L_{\text{o},2} = L_{\text{C},1} - j, \quad \text{and} \quad L_{\text{o},3} = \min(j + L_{\text{W},2}, k + L_{\text{C},2}) - k.$$

Given the length of the hybridization region as well as the presence/absence of dangling ends, we can compute the hybridization energy,

$$E = \gamma(L_{\text{o},1} + L_{\text{o},2} + L_{\text{o},3} - 3) + \gamma(1 - d_{m1}) + \gamma(1 - d_{m2}) + \frac{\gamma}{2}(d_l + d_r + 2d_{m1} + 2d_{m2}).$$

The combinatorial multiplicity of a left-tilted 2-2 quaternary complex is constructed using the same reasoning as in the case of a 3-1 complex,

$$\mathfrak{C}(L_{C,1}, L_{C,2}, L_{W,1}, L_{W,2}, i, j, k) = \mathfrak{C}(L_C) \times \left[ \frac{\mathfrak{C}(L_{W,1})}{4^{L_{o,1}}} \mathbb{1}\left(L_{o,1} < L_U^{\min}\right) + \mathbb{1}\left(L_{o,1} \geq L_U^{\min}\right) \right]$$
$$\times \left[ \frac{\mathfrak{C}(L_{W,2})}{4^{L_{o,2}}} \mathbb{1}\left(L_{o,2} < L_U^{\min}\right) + \mathbb{1}\left(L_{o,2} \geq L_U^{\min}\right) \right]$$
$$\times \left[ \frac{\mathfrak{C}(L_{C,1})}{4^{L_{o,3}}} \mathbb{1}\left(L_{o,3} < L_U^{\min}\right) + \mathbb{1}\left(L_{o,3} \geq L_U^{\min}\right) \right].$$

To prevent double-counting the same quaternary complex, we include either the complex or its rotated representation in the container of possible complexes, but not both. If the complex is symmetric under rotation, we multiply the combinatorial multiplicity by 1/2. Given a left-tilted 2-2 quaternary complex $(L_{C,1}, L_{C,2}, L_{W,1}, L_{W,2}, i, j, k)$, we can compute the corresponding rotated complex $(L_{C,1}^{\text{rot}}, L_{C,2}^{\text{rot}}, L_{W,1}^{\text{rot}}, L_{W,2}^{\text{rot}}, i^{\text{rot}}, j^{\text{rot}}, k^{\text{rot}})$ by applying a linear map. The mapping of the oligomer lengths is illustrated in *Figure 12B*. We see that strand `C2` after rotation corresponds to strand `W1` before rotation, implying that $L_{C,2}^{\text{rot}} = L_{W,1}$. The same reasoning can be applied to all strands leading to the map,

$$L_{C,1}^{\text{rot}} = L_{W,2}, \quad L_{C,2}^{\text{rot}} = L_{W,1}, \quad L_{W,1}^{\text{rot}} = L_{C,2}, \quad \text{and} \quad L_{W,2}^{\text{rot}} = L_{C,1}.$$

In order to compute the map of the alignment indices under rotation, we need to express the relative positions of the strands with respect to the position of strand `C1` after rotation, which corresponded to `W2` before rotation. For example, $|i^{\text{rot}}|$ corresponds to the number of nucleotides by which strand `C2` (before rotation) protrudes beyond strand `W2` (before rotation). Expressed in terms of variables before rotation, this distance may be written as $k + L_{C,2} - jL_{W,2}$. Analogous relations can be derived for all alignment indices,

$$i^{\text{rot}} = j - k + L_{W,2} - L_{C,2}, \quad j^{\text{rot}} = j + L_{W,2} - L_{C,1}, \quad \text{and} \quad k^{\text{rot}} = j - i + L_{W,1} - L_{W,2}.$$

## Right-tilted 2-2 quaternary complexes

A 2-2 quaternary complex is called right-tilted if strand `W1` is connected to strand `W2` via strand `C2` (*Figure 13*). As in the case of the left-tilted 2-2 complex, the oligomer lengths are again called $L_{W,1}, L_{W,2}, L_{C,1}$ and $L_{C,2}$, but the values of the alignment indices that are possible for the right-tilted quaternary complex differ from the ones of the left-tilted complex,

$$i \in [L_{C,1} - L_{W,1} - 1, L_{C,1} - 1], \quad j \in [L_{C,1}, i + L_{W,1} - 1], \quad \text{and} \quad k \in [i + L_{W,1}, k + L_{C,2} - 1].$$

Note that the range of $i$ is chosen such that at least one nucleotide of strand `W1` always extends to the right beyond the end of strand `C1`, allowing for a hybridization region between strand `C2` and `W1`. The boolean variables denoting the presence/absence of dangling ends read

$$d_l = \begin{cases} 0 & \text{if } i = 0, \\ 1 & \text{otherwise,} \end{cases} \qquad d_{m1} = \begin{cases} 0 & \text{if } i + L_{W,1} = j, \\ 1 & \text{otherwise,} \end{cases}$$

$$d_{m2} = \begin{cases} 0 & \text{if } L_{C,1} = k, \\ 1 & \text{otherwise,} \end{cases} \qquad d_r = \begin{cases} 0 & \text{if } j + L_{W,2} = k + L_{C,2}, \\ 1 & \text{otherwise.} \end{cases}$$

The length of the overlap regions is given by

$$L_{o,1} = L_{C,1} - \max(i, 0), \quad L_{o,2} = i + L_{W,1} - k, \quad \text{and} \quad L_{o,3} = \min(k + L_{C,2}, j + L_{W,2}) - j.$$

Like in the case of left-tilted 2-2 quaternary complexes, the total hybridization energy is computed via

$$E = \gamma(L_{o,1} + L_{o,2} + L_{o,3} - 3) + \gamma(1 - d_{m1}) + \gamma(1 - d_{m2}) + \frac{\gamma}{2}(d_l + d_r + 2d_{m1} + 2d_{m2}),$$

and the combinatorial multiplicity via

$$\mathfrak{C}(L_{\mathrm{C},1}, L_{\mathrm{C},2}, L_{\mathrm{W},1}, L_{\mathrm{W},2}, i, j, k) = \mathfrak{C}(L_{\mathrm{C}}) \times \left[ \frac{\mathfrak{C}(L_{\mathrm{W},1})}{4^{L_{\mathrm{o},1}}} \mathbb{1}\left(L_{\mathrm{o},1} < L_{\mathrm{U}}^{\min}\right) + \mathbb{1}\left(L_{\mathrm{o},1} \geq L_{\mathrm{U}}^{\min}\right) \right]$$
$$\times \left[ \frac{\mathfrak{C}(L_{\mathrm{W},2})}{4^{L_{\mathrm{o},2}}} \mathbb{1}\left(L_{\mathrm{o},2} < L_{\mathrm{U}}^{\min}\right) + \mathbb{1}\left(L_{\mathrm{o},2} \geq L_{\mathrm{U}}^{\min}\right) \right]$$
$$\times \left[ \frac{\mathfrak{C}(L_{\mathrm{C},1})}{4^{L_{\mathrm{o},3}}} \mathbb{1}\left(L_{\mathrm{o},3} < L_{\mathrm{U}}^{\min}\right) + \mathbb{1}\left(L_{\mathrm{o},3} \geq L_{\mathrm{U}}^{\min}\right) \right].$$

We include either the quaternary complex or its rotated representation in the list of possible complexes to avoid double-counting. Moreover, the combinatorial multiplicity is divided by 2 if the complex is symmetric under rotation. It turns out that the rotation map for the right-tilted 2-2 quaternary complex is identical to the one of the left-tilted 2-2 complex,

$$L_{\mathrm{C},1}^{\mathrm{rot}} = L_{\mathrm{W},2}, \quad L_{\mathrm{C},2}^{\mathrm{rot}} = L_{\mathrm{W},1}, \quad L_{\mathrm{W},1}^{\mathrm{rot}} = L_{\mathrm{C},2}, \quad L_{\mathrm{W},2}^{\mathrm{rot}} = L_{\mathrm{C},1},$$
$$i^{\mathrm{rot}} = j - k + L_{\mathrm{W},2} - L_{\mathrm{C},2}, \quad j^{\mathrm{rot}} = j + L_{\mathrm{W},2} - L_{\mathrm{C},1}, \quad \text{and} \quad k^{\mathrm{rot}} = j - i + L_{\mathrm{W},1} - L_{\mathrm{W},2}.$$

## Numerical solution of the (De)hybridization equilibrium in the adiabatic approach

Based on the list of complexes constructed in the previous section, we can compute the equilibrium concentration of strands and complexes reached in the (de)hybridization equilibrium. In the following, we denote the concentration of a coarse-grained oligomer with length $L$ as $c(L)$, and the concentration of an oligomer with length $L$ and known sequence as $c_s(L)$. Recall that we assumed that all sequences of a given length that are compatible with the circular genome are equally likely in the pool. Thus, the concentration of the coarse-grained oligomer and the concentration of an oligomer with specified sequence are related by the combinatorial multiplicity,

$$c(L) = \mathfrak{C}(L) c_s(L).$$

In order to compute the concentration of a complex based on the concentration of single strands, we make use of the law of mass action. The concentration of a specific sequence realization of a complex is computed as the product of concentrations of the strands forming the complex divided by the dissociation constant $K_{\mathrm{d}}$,

$$c_s(\vec{L}, \vec{i}) = \frac{1}{K_{\mathrm{d}}(\vec{L}, \vec{i})} \prod_j c_s(L_j).$$

Here, $\vec{L}$ is the vector denoting the lengths of the strands comprising the complex, and $\vec{i}$ are the alignment indices. The dissociation constant is set by the hybridization energy, $\Delta G$, of the complex,

$$K_{\mathrm{d}}(\vec{L}, \vec{i}) = (c^\circ)^{n-1} \exp(\beta \Delta G),$$

where $c^\circ = 1\mathrm{M}$ is the standard concentration. Just as in the case of single strands, the concentration of the sequence-independent coarse-grained complex is related to the concentration of a complex with specific sequence realization via the combinatorial prefactor,

$$c(\vec{L}, \vec{i}) = \mathfrak{C}(\vec{L}, \vec{i}) c_s(\vec{L}, \vec{i}).$$

It can be useful to combine the combinatorial multiplicity and the dissociation constant into a single effective association constant,

$$\mathcal{K}_{\mathrm{a}}(\vec{L}, \vec{i}) = \frac{\mathfrak{C}(\vec{L}, \vec{i})}{K_{\mathrm{d}}(\vec{L}, \vec{i})}.$$

Note that the effective association constant including the combinatorial multiplicity is denoted by $\mathcal{K}_{\mathrm{a}}$ (in curly font), while the association constant without combinatorial multiplicity is denoted $K_{\mathrm{a}} = K_{\mathrm{d}}^{-1}$ (in regular font).

In the adiabatic approach, we study the behavior of the system on timescales that are long enough for the system to reach the (de)hybridization equilibrium, but too short for templated ligation events to take place. Therefore, the length of the oligomers is expected not to change throughout the

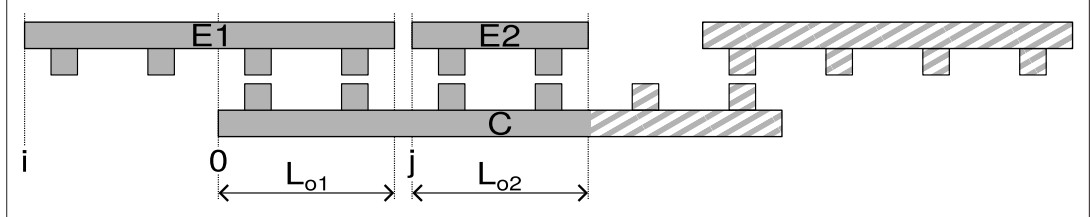

**Figure 14.** Schematic representation of a complex that allows templated ligation. The strands `E1` and `E2` are adjacent to each other, such that a covalent bond can form between their ends. The length of the product strand, $L_p$, is set by the length of the educt strands, $L_{e,1}$ and $L_{e,2}$. The likelihood for the complex to form a product oligomer whose sequence is compatible with the true circular genome, $p_{corr}$, is determined by the length of the educts and the length of their hybridization region with the template. The parts of the complex shown with hatching do not affect $p_{corr}$.

equilibration process, and we need to introduce a separate mass conservation law for each coarse-grained oligomer, that is each oligomer length, that is included in the pool. For each length, the concentration of single-stranded coarse-grained oligomers of length $L$ and the concentration of coarse-grained oligomers of length $L$ bound in a complex need to add up to their total concentration $c(L)$ set by the initial condition,

$$c^{eq}\left(L\right) + \sum_{(\vec{L},\vec{i})\,|\,\exists\,j\,\text{s.t.}\,\vec{L}_j=L} c^{eq}\left(\vec{L},\vec{i}\right) = c^{tot}(L).$$

In this equation, $(\vec{L},\vec{i})\,|\,\exists\,j\,\text{s.t.}\,\vec{L}_j = L$ denotes the summation over all complexes that contain at least one strand of length $L$. Note that we referred to the total concentration of oligomers of length $L$ as $c(L)$ in the main text, but for clarity, we use $c^{tot}(L)$ in the following.

Combining the mass conservation requirement with the mass action law gives a set of coupled polynomial equations. The number of equations equals the number of distinct oligomer lengths included in the pool. The polynomial equations are of degree 4, as quaternary complexes are the largest complexes to be accounted for and their concentrations equal the product of the four strands comprising the complex. We determine the equilibrium concentrations by finding the root of this set of fourth-degree polynomial equations using the Levenberg-Marquardt algorithm.

## Computing replication observables based on the adiabatic approach

As we are not modeling templated ligation events explicitly in the adiabatic approach, we compute replication observables based on the equilibrium concentration of complexes that are in a configuration which allows for a templated ligation reaction to happen. Templated ligations are possible if two strands in the complex are adjacent to each other, that is there is no gap in between two oligomers that are hybridized to the same template strand (*Figure 14*). Recall that the absence of a gap between two oligomers in the complex implies that the dangling end indicator variable $d_m = 0$. The length of the product strand `P` is equal to the sum of the lengths of the two educt strands `E1` and `E2`, $L_p = L_{e,1} + L_{e,2}$. We can use the information about the length of the product strand to compute the yield of replication. By definition, the yield equals the fraction of nucleotides used to form VCG oligomers, that is strands that are at least $L_U$ long,

$$y = \frac{\#\,\text{nucleotides incorporated in VCG oligomers}}{\#\,\text{incorporated nucleotides}}. \tag{7}$$

We can express this quantity in terms of the equilibrium concentration of complexes facilitating templated ligation,

$$y = \frac{\sum_{(\vec{L},\vec{i})\,|\,(d_m=0\,\wedge\,L_{e,1}+L_{e,2}\geq L_U^{min})}\,\min(L_{e,1},L_{e,2})\,c^{eq}\left(\vec{L},\vec{i}\right)}{\sum_{(\vec{L},\vec{i})\,|\,d_m=0}\,\min(L_{e,1},L_{e,2})\,c^{eq}\left(\vec{L},\vec{i}\right)}.$$

$(\vec{L}, \vec{i}) \,|\, (d_m = 0 \wedge L_{e,1} + L_{e,2} \geq L_U^{min})$ denotes the summation over all complexes, in which (i) the strands E1 and E2 are adjacent to each other, that is $d_m = 0$, and (ii) the length of the product $L_{e,1} + L_{e,2} \geq L_U^{min}$. We multiply the equilibrium concentration by the length of the shorter educt strand to account for the number of incorporated nucleotides in line with the definition of the yield (**Equation 7**).

In order to compute the fidelity of replication, we need to distinguish between product oligomer sequences that are compatible with the genomes (correct sequences) and sequences that are incompatible with the genome (false sequences). As we do not know about the details of the sequences due to the coarse-grained representation of the complexes, we need to invoke a combinatorial argument to determine the fraction of correct products. To this end, we compare the number of product sequences that might be produced in a complex of given oligomer lengths and alignment indices to the number of correct products associated with the same complex configuration. The combinatorial multiplicity of the products that could be produced by a complex of given configuration is set by the combinatorial multiplicity of the possible templates, $\mathfrak{C}(L_{o,1} + L_{o,2})$, multiplied by the multiplicity of the educt strands hybridizing to the template with given lengths of the hybridization regions, $L_{o,1}$ and $L_{o,2}$,

$$\mathfrak{C}(\text{possible products}) = \mathfrak{C}(L_{o,1} + L_{o,2}) \frac{\mathfrak{C}(L_{e,1})}{\mathfrak{C}(L_{o,1})} \frac{\mathfrak{C}(L_{e,2})}{\mathfrak{C}(L_{o,2})}.$$

The multiplicity of correct products equals the combinatorial multiplicity of strands that have the same length as the product,

$$\mathfrak{C}(\text{correct products}) = \mathfrak{C}(L_{e,1} + L_{e,2}).$$

This implies that the probability for a complex of given shape $(\vec{L}, \vec{i})$ to form a correct product is given by

$$p_{corr}(\vec{L}, \vec{i}) = \frac{\mathfrak{C}(\text{correct products})}{\mathfrak{C}(\text{possible products})} = \frac{\mathfrak{C}(L_{e,1} + L_{e,2}) \, \mathfrak{C}(L_{o,1}) \, \mathfrak{C}(L_{o,2})}{\mathfrak{C}(L_{o,1} + L_{o,2}) \, \mathfrak{C}(L_{e,1}) \, \mathfrak{C}(L_{e,2})}.$$

Using this probability, we can compute the fidelity of replication,

$$f = \frac{\sum_{(\vec{L}, \vec{i}) \,|\, (d_m = 0 \wedge L_{e,1} + L_{e,2} \geq L_U^{min})} p_{corr}(\vec{L}, \vec{i}) \min(L_{e,1}, L_{e,2}) c^{eq}(\vec{L}, \vec{i})}{\sum_{(\vec{L}, \vec{i}) \,|\, (d_m = 0 \wedge L_{e,1} + L_{e,2} \geq L_U^{min})} \min(L_{e,1}, L_{e,2}) c^{eq}(\vec{L}, \vec{i})}$$

as well as the replication efficiency,

$$\eta = \frac{\sum_{(\vec{L}, \vec{i}) \,|\, (d_m = 0 \wedge L_{e,1} + L_{e,2} \geq L_U^{min})} p_{corr}(\vec{L}, \vec{i}) \min(L_{e,1}, L_{e,2}) c^{eq}(\vec{L}, \vec{i})}{\sum_{(\vec{L}, \vec{i}) \,|\, d_m = 0} \min(L_{e,1}, L_{e,2}) c^{eq}(\vec{L}, \vec{i})}.$$

## Acknowledgements

We thank Paul Higgs and members of the Gerland group for stimulating discussions. This work was supported by the Deutsche Forschungsgemeinschaft (DFG, German Research Foundation) via the CRC/TRR 392 Molecular Evolution (Project-ID 521256690), and under Germany's Excellence Strategy (EXC-2094-390783311, ORIGINS).

## Additional information

### Funding

| Funder | Grant reference number | Author |
|---|---|---|
| Deutsche Forschungsgemeinschaft | CRC 392 Project-ID 521256690 | Ulrich Gerland |

| Funder | Grant reference number | Author |
|--------|------------------------|--------|
| Deutsche Forschungsgemeinschaft | ORIGINS EXC-2094-390783311 | Ulrich Gerland |

The funders had no role in study design, data collection and interpretation, or the decision to submit the work for publication.

## Author contributions

Ludwig Burger, Conceptualization, Software, Validation, Investigation, Visualization, Methodology, Writing – original draft, Writing – review and editing; Ulrich Gerland, Conceptualization, Supervision, Funding acquisition, Investigation, Writing – original draft, Project administration, Writing – review and editing

## Author ORCIDs

Ludwig Burger ⓘ https://orcid.org/0009-0000-0699-6302
Ulrich Gerland ⓘ https://orcid.org/0000-0002-0859-6422

Reviewer #1 (Public review): https://doi.org/10.7554/eLife.104043.3.sa1
Reviewer #2 (Public review): https://doi.org/10.7554/eLife.104043.3.sa2
Author response https://doi.org/10.7554/eLife.104043.3.sa3

# Additional files

## Supplementary files

Supplementary file 1. List of genomes sampled via the Metropolis-Hastings algorithm for a genome length of 64 nucleotides.

## Data availability

This manuscript presents a computational study, employing both stochastic simulations and a deterministic coarse-grained approximation (the adiabatic approach). Means and standard deviations from the stochastic simulations are shown in Figure 2 and listed in Appendix 2 - Table 1. Data from the adiabatic approach are shown in Figures 2-7 and can be reproduced using the code available at https://github.com/gerland-group/VirtualCircularGenome (copy archived at *Burger, 2025*). This repository also includes the code for sampling circular genomes via a Metropolis-Hastings-type algorithm.

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

## Appendix 1

## Scaling of replication efficiency with oligomer length in single-length VCG pools

We consider a pool containing monomers and VCG oligomers of a single length. As the feedstock only contains monomers, the total monomer concentration and the total feedstock concentration are identical, $c_F^{tot} = c^{tot}(1)$. Similarly, the total VCG concentration equals the total concentration of oligomers of length $L_V$, $c_V^{tot} = c^{tot}(L_V)$. In systems of this type, the replication efficiency can be expressed as

$$\eta = \frac{c_{F+V,c} + L_V \, c_{V+V,c}}{c_{F+F} + c_{F+V} + L_V \, c_{V+V}},$$

where the concentrations denote the equilibrium concentration of all complexes facilitating the indicated type of templated ligation. Note that we do not include F+F ligations (i.e. dimerizations) in the numerator as they do not contribute to the yield. Moreover, the V+V ligations are multiplied by the length of the oligomers $L_V$, as each V+V extends an existing oligomer by $L_V$ nucleotides.

Assuming that complexes comprised of three strands are the dominant type of complexes, we can express the equilibrium concentrations as a product of the equilibrium concentrations of the free strands and the effective association constants,

$$c_{F+F} = \mathcal{K}_{F+F}^a (c_{F,s}^{eq})^2 \, c_{V,s}^{eq}, \tag{8a}$$

$$c_{F+V} = \mathcal{K}_{F+V}^a \, c_{F,s}^{eq} (c_{V,s}^{eq})^2, \quad c_{F+V,c} = \mathcal{K}_{F+V,c}^a \, c_{F,s}^{eq} (c_{V,s}^{eq})^2 \tag{8b}$$

$$c_{V+V} = \mathcal{K}_{V+V}^a \, (c_{V,s}^{eq})^3, \quad c_{V+V,c} = \mathcal{K}_{V+V,c}^a \, (c_{V,s}^{eq})^3 \tag{8c}$$

Here, $c_{V,s}^{eq}$ denotes the equilibrium concentration of a VCG oligomer with a specific sequence; the equilibrium concentration of all VCG oligomers for any sequence $c_V^{eq}$ is computed by multiplying with the combinatorial multiplicity, $c_V^{eq} = \mathfrak{C}(L_V) \, c_{V,s}^{eq}$ (analogously, $c_F^{eq} = \mathfrak{C}(1) \, c_{F,s'}^{eq}$ see Methods for details). *Equation 8* allows us to write the replication efficiency as

$$
\eta = \frac{\mathcal{K}_{F+V,c}^a \, c_{F,s}^{eq} (c_{V,s}^{eq})^2 + \mathcal{K}_{V+V,c}^a \, (c_{V,s}^{eq})^3}{\mathcal{K}_{F+F}^a (c_{F,s}^{eq})^2 \, c_{V,s}^{eq} + \mathcal{K}_{F+V}^a \, c_{F,s}^{eq} (c_{V,s}^{eq})^2 + \mathcal{K}_{V+V}^a \, (c_{V,s}^{eq})^3}
$$
$$
= \frac{\mathcal{K}_{F+V,c}^a \, r_s^2 + \mathcal{K}_{V+V,c}^a \, r_s^3}{\mathcal{K}_{F+F}^a \, r_s + \mathcal{K}_{F+V}^a \, r_s^2 + \mathcal{K}_{V+V}^a \, r_s^3} = \frac{\mathcal{K}_{F+V,c}^a \, r_s + \mathcal{K}_{V+V,c}^a \, r_s^2}{\mathcal{K}_{F+F}^a + \mathcal{K}_{F+V}^a \, r_s + \mathcal{K}_{V+V}^a \, r_s^2},
\tag{9}
$$

where we introduced the ratio of equilibrium concentrations, $r_s := c_{V,s}^{eq}/c_{F,s}^{eq}$. Maximizing *Equation 9* with respect to $r_s$ yields

$$r_s^{opt} = \frac{L_V \mathcal{K}_{F+F}^a \mathcal{K}_{V+V,c}^a + \sqrt{L_V^2 (\mathcal{K}_{F+F}^a)^2 (\mathcal{K}_{V+V,c}^a)^2 + L_V \mathcal{K}_{F+F}^a \mathcal{K}_{F+V,c}^a (\mathcal{K}_{V+V}^a \mathcal{K}_{F+V,c}^a - \mathcal{K}_{V+V,c}^a \mathcal{K}_{F+V}^a)}}{L_V (\mathcal{K}_{V+V}^a \mathcal{K}_{F+V,c}^a - \mathcal{K}_{V+V,c}^a \mathcal{K}_{F+V}^a)}$$

The effective association constants are computed using the combinatorial rules outlined in the Methods section. We find that $\mathcal{K}_{F+V,c}^a$ is similar to $\mathcal{K}_{F+V}^a$, deviating at most by 10%. Moreover,

$$\frac{L_V (\mathcal{K}_{F+F}^a)^2 (\mathcal{K}_{V+V,c}^a)^2}{\mathcal{K}_{F+F}^a \mathcal{K}_{F+V,c}^a \left( \mathcal{K}_{V+V}^a \mathcal{K}_{F+V,c}^a - \mathcal{K}_{V+V,c}^a \mathcal{K}_{F+V}^a \right)} \le 5 \cdot 10^{-2} \ll 1,$$

and,

$$\frac{\mathcal{K}_{F+F}^a \mathcal{K}_{V+V,c}^a}{\mathcal{K}_{V+V}^a \mathcal{K}_{F+V,c}^a - \mathcal{K}_{V+V,c}^a \mathcal{K}_{F+V}^a} \le 10\% \; r_s^{opt}.$$

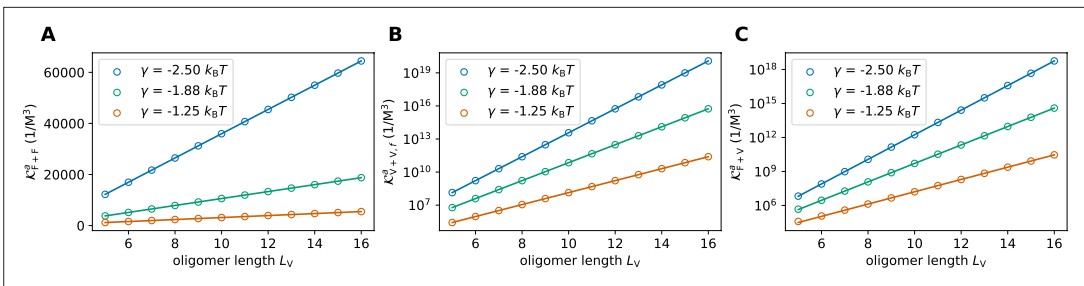

**Appendix 1—figure 1.** Effective association constants of complexes facilitating F+F ligations (**A**), false V+V ligations (**B**), and F+V ligations (**C**). The dots depict the effective association constants derived based on the combinatorial rules presented in the Methods section, and the solid lines represent the respective scaling laws introduced in *Equations 10 and 11*. Different colors correspond to different hybridization energies per matching nearest neighbor block γ.

These observations allow us to obtain a significantly simplified approximate expression for $r_s^{\text{opt}}$,

$$r_s^{\text{opt}} \approx \sqrt{\frac{\mathcal{K}_{\text{F+F}}^{\text{a}}}{L_{\text{V}} \left( \mathcal{K}_{\text{V+V}}^{\text{a}} - \mathcal{K}_{\text{V+V,c}}^{\text{a}} \right)}} = \sqrt{\frac{\mathcal{K}_{\text{F+F}}^{\text{a}}}{L_{\text{V}} \, \mathcal{K}_{\text{V+V,f}}^{\text{a}}}} \, .$$

We used the identity $\mathcal{K}_{\text{V+V}}^{\text{a}} = \mathcal{K}_{\text{V+V,c}}^{\text{a}} + \mathcal{K}_{\text{V+V,f}}^{\text{a}}$ in the last step. To derive the scaling of $r_s^{\text{opt}}$ as a function of $L_{\text{V}}$, we need to analyze how the effective association constants depend on $L_{\text{V}}$.

1. $\mathcal{K}_{\text{F+F}}^{\text{a}}$ scales linearly with $L_{\text{V}}$. In F+F ligations, a VCG oligomer acts as a template facilitating the ligation of two monomers. Independent of the length of the VCG oligomer, the two monomers always have the same binding affinity to their template. However, the number of possible positions that the two adjacent monomers can have on the template scales linearly in template length, causing the effective association constant to depend linearly on $L_{\text{V}}$,

$$\mathcal{K}_{\text{F+F}}^{\text{a}} = \mathcal{K}_{\text{F+F}}^{\text{a,} \circ} \left( \frac{L_{\text{V}}}{\Lambda_{\text{F+F}}} - 1 \right) . \tag{10}$$

2. $\mathcal{K}_{\text{V+V,f}}^{\text{a}}$ scales exponentially with $L_{\text{V}}$. As illustrated in *Figure 2E*, the length of the hybridization region in complexes facilitating incorrect V+V ligations equals $L_{\text{V}}$. As the hybridization energy is proportional to the length of the hybridization site, $L_{\text{V}}$, the effective association constant $\mathcal{K}_{\text{V+V,f}}^{\text{a}}$ scales exponentially with $L_{\text{V}}$. The number of complexes facilitating incorrect product formation is a function of the unique subsequence length $L_U$, but does not depend on the oligomer length $L_{\text{V}}$. Therefore, there is no additional dependence of $\mathcal{K}_{\text{V+V}}^{\text{a}}$ on $L_{\text{V}}$,

$$\mathcal{K}_{\text{V+V,f}}^{\text{a}} = \mathcal{K}_{\text{V+V,f}}^{\text{a,} \circ} \exp \left( |\gamma| L_{\text{V}} \right) . \tag{11}$$

*Appendix 1—figure 1* shows the $L_{\text{V}}$-dependence of the effective association constants. The circles represent the effective association constants derived based on the combinatorial rules discussed in the Methods section, while the solid lines show the length-dependent scaling computed via *Equation 10* and *Equation 11*.

Given the scaling of $\mathcal{K}_{\text{F+F}}^{\text{a}}$ and $\mathcal{K}_{\text{V+V,f}}^{\text{a}}$, we find that the optimal ratio of equilibrium concentrations $r_s^{\text{opt}}$ equals

$$r_s^{\text{opt}} = \sqrt{\frac{\mathcal{K}_{\text{F+F}}^{\text{a,} \circ}}{\mathcal{K}_{\text{V+V,f}}^{\text{a,} \circ}}} \sqrt{\frac{1}{\Lambda_{\text{F+F}}} - \frac{1}{L_{\text{V}}}} \exp \left( -\frac{\gamma L_{\text{V}}}{2} \right) .$$

We obtain $r_{\text{opt}}$ (*Figure 2*) from $r_s^{\text{opt}}$ by multiplying with the appropriate combinatorial multiplicity,

$$r_{\text{opt}} = \frac{\mathfrak{C}(L_{\text{V}})}{\mathfrak{C}(1)} r_s^{\text{opt}} .$$

Combining the expression for $r_s^{\text{opt}}$ with **Equation 9**, the optimal efficiency of replication can be expressed as

$$\eta_{\text{opt}} = \frac{\mathcal{K}_{\text{F+V,c}}^{\text{a}}\, r_s^{\text{opt}} + \mathcal{K}_{\text{V+V,c}}^{\text{a}}\, (r_s^{\text{opt}})^2}{\mathcal{K}_{\text{F+F}}^{\text{a}} + \mathcal{K}_{\text{F+V}}^{\text{a}}\, r_s^{\text{opt}} + \mathcal{K}_{\text{V+V}}^{\text{a}}\, (r_s^{\text{opt}})^2} = \frac{1 + \dfrac{L_V\,\mathcal{K}_{\text{V+V,c}}^{\text{a}}}{\mathcal{K}_{\text{F+V}}^{\text{a}}}\, r_s^{\text{opt}}}{1 + \dfrac{L_V\,\mathcal{K}_{\text{V+V}}^{\text{a}}}{\mathcal{K}_{\text{F+V}}^{\text{a}}}\, r_s^{\text{opt}} + \dfrac{\mathcal{K}_{\text{F+F}}^{\text{a}}}{\mathcal{K}_{\text{F+V}}^{\text{a}}\, r_s^{\text{opt}}}}.$$

As $\dfrac{L_V\,\mathcal{K}_{\text{V+V}}^{\text{a}}}{\mathcal{K}_{\text{F+V}}^{\text{a}}}\, r_s^{\text{opt}} + \dfrac{\mathcal{K}_{\text{F+F}}^{\text{a}}}{\mathcal{K}_{\text{F+V}}^{\text{a}}\, r_s^{\text{opt}}} \ll 1$ (for sufficiently high $L_V$), we can expand in this small argument,

$$\eta_{\text{opt}} \approx 1 - \frac{L_V\,\left(\mathcal{K}_{\text{V+V}}^{\text{a}} - \mathcal{K}_{\text{V+V,c}}^{\text{a}}\right)}{\mathcal{K}_{\text{F+V}}^{\text{a}}}\, r_s^{\text{opt}} - \frac{\mathcal{K}_{\text{F+F}}^{\text{a}}}{\mathcal{K}_{\text{F+V}}^{\text{a}}\, r_s^{\text{opt}}} = 1 - \frac{L_V\,\mathcal{K}_{\text{V+V,f}}^{\text{a}}}{\mathcal{K}_{\text{F+V}}^{\text{a}}}\, r_s^{\text{opt}} - \frac{\mathcal{K}_{\text{F+F}}^{\text{a}}}{\mathcal{K}_{\text{F+V}}^{\text{a}}\, r_s^{\text{opt}}}. \tag{12}$$

To understand the scaling of $\eta_{\text{opt}}$ as function of $L_V$, we need to characterize the scaling of $\mathcal{K}_{\text{F+V}}^{\text{a}}$ with $L_V$. A typical complex allowing for a F+V ligation is depicted in **Figure 5D**. The length of the overlap $L_o$ region between template and educt VCG oligomer can vary; it is at least one nucleotide, and at most $L_V - 1$ nucleotides (one base pairing position needs to remain available for the monomer). We obtain the effective association constant by summing the contributions of all of these configurations,

$$\mathcal{K}_{\text{F+V}}^{\text{a}} \sim \sum_{L_o=1}^{L-1} \exp(L_o|\gamma|) \sim \exp(|\gamma|L)$$

Therefore, we use the following scaling ansatz for $\mathcal{K}_{\text{F+V}}^{\text{a}}$,

$$\mathcal{K}_{\text{F+V}}^{\text{a}} = \mathcal{K}_{\text{F+V}}^{\text{a,}\circ}\, \exp\left(|\gamma|L_V\right). \tag{13}$$

Combining the scaling laws for $\mathcal{K}_{\text{F+F}}^{\text{a}}$, $\mathcal{K}_{\text{F+V}}^{\text{a}}$, and $\mathcal{K}_{\text{V+V,f}}^{\text{a}}$ with **Equation 12**, we find

$$\begin{aligned}
\eta_{\text{opt}} &= 1 - 2\frac{\sqrt{\mathcal{K}_{\text{F+F}}^{\text{a,}\circ}\mathcal{K}_{\text{V+V,f}}^{\text{a,}\circ}}}{\mathcal{K}_{\text{F+V}}^{\text{a,}\circ}}\sqrt{\frac{1}{\Lambda_{\text{F+F}}} - \frac{1}{L_V}}L_V \exp\left(-\frac{|\gamma|L_V}{2}\right) \\
&= 1 - \eta^{\circ}\sqrt{\frac{1}{\Lambda_{\text{F+F}}} - \frac{1}{L_V}}L_V \exp\left(-\frac{|\gamma|L_V}{2}\right).
\end{aligned}$$

## Appendix 2

### Dependence of replication efficiency on the sequence of the genome

The coarse-grained representation used in the adiabatic approach applies only to a specific class of genomes, namely those in which the exhaustive coverage length $L_E$ is maximal and the unique motif length $L_U$ is minimal. This corresponds to genomes that satisfy $L_E = L_E^{max}$ and $L_U = L_U^{max}$ (see Methods for details).

To analyze replication behavior beyond this limited case, we developed a fully sequence-resolved extension of the adiabatic approach. Rather than using a coarse-grained view of oligomers, this method considers each distinct strand sequence as a separate chemical species. For example, to study a genome of length $L_G = 64$ nt that is encoded in a pool containing monomers and octamers, a total of 132 individual oligomers (four monomers and 128 distinct octamers) must be accounted for. This contrasts sharply with the coarse-grained scenario, which involves only two variables (total monomer and total octamer concentrations). Starting from the single-stranded oligomers, all possible complexes involving up to three strands are enumerated, and their hybridization free energies are calculated using the same energetic model applied in the coarse-grained framework. In the aforementioned example, this results in 351,200 distinct sequence-resolved complexes, compared to just 135 complexes in the coarse-grained model, highlighting the increased computational demands in terms of memory and runtime. The hybridization equilibrium is computed by solving the algebraic system defined by the law of mass action and mass conservation. The procedure mirrors that of the coarse-grained approach (Methods), with the key difference that combinatorial prefactors are no longer required: these are inherently encoded in the full enumeration of unique sequences and their binding configurations.

For our further analysis, we fix the genome length at $L_G = 64$ nt and systematically vary the motif length scales $L_E$ and $L_U$. Genomes with desired values of $L_E$ and $L_U$ are constructed as described in the Methods section: Starting from a genome with maximal motif entropy ($L_E = L_E^{max} = 3$ nt, $L_U = L_U^{min} = 4$ nt), we reduce the motif entropy on intermediate length scales $L_E < L < L_U$ to achieve the target characteristics. *Supplementary file 1* provides an overview of all sampled genome sequences. We consider two limiting cases for motif distributions between $L_E$ and $L_U$. In the weakly biased case, the motif distribution (for motifs of length $L_E < L < L_U$) is nearly uniform: most motifs of length $L$ appear exactly once, except for a few motifs that occur twice. This leads to genomes where $L \neq L_U$, but the deviation from maximal entropy is minimal. In the strongly biased case, many motifs appear multiple times, resulting in a much less uniform motif distribution.

Each genome is mapped to a VCG pool containing monomers and oligomers of variable length $L_V$. Assuming all oligomers are chemically activated, we compute the replication efficiency of each pool as a function of the relative oligomer concentration, for different values of $L_V$ and across genome types (*Appendix 2—figure 1*). As expected, replication efficiency exhibits a maximum at intermediate oligomer concentrations. The maximum efficiency achieved depends on both the VCG oligomer length and the genome's motif properties. In general, longer oligomers enable more efficient replication. However, at a fixed oligomer length, genomes with higher unique motif lengths $L_U$ (or lower $L_E$) replicate with lower efficiency, implying that these genomes require longer oligomers for successful replication. For example, a VCG pool with oligomers of length $L_V = 9$ nt can replicate a genome with $L_E = 3$ nt and $L_U = 4$ nt at 97% efficiency (see solid green curve in *Appendix 2—figure 1B*). In contrast, replicating a genome with $L_E = 3$ nt and $L_U = 10$ nt requires oligomers of at least $L_V = 11$ nt to reach comparable efficiency (see dotted green curve in *Appendix 2—figure 1C*).

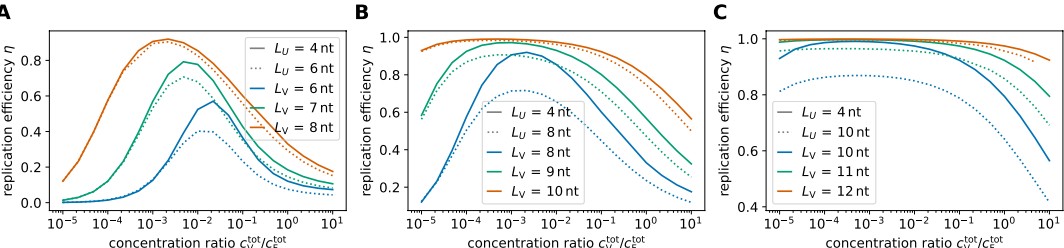

**Appendix 2—figure 1.** Replication efficiency as a function of the concentration of VCG oligomers for different choices of genomes and varying VCG oligomer length. All genomes are $L_G = 64\,\text{nt}$ long and include all motifs up to length $L_E = 2\,\text{nt}$, but differ with respect to their minimal unique subsequence length $L_U$: (**A**) $L_U = 6\,\text{nt}$, (**B**) $L_U = 8\,\text{nt}$, and (**C**) $L_U = 10\,\text{nt}$ (shown as dotted lines). For comparison, every panel shows the replication efficiency of a genome with $L_E = 3\,\text{nt}$, $L_U = 4\,\text{nt}$ (solid line). Different colors are used to distinguish different VCG oligomer lengths. Under otherwise identical conditions (e.g. identical oligomer length), replication proceeds with lower efficiency in genomes with higher unique subsequence length $L_U$.

Across all tested genomes, the oligomer length required to achieve $> 95\%$ efficiency (denoted $L_V^\star$) consistently exceeds the unique motif length $L_U$ (**Appendix 2—figure 2**). However, the difference $L_V^\star - L_U$ depends on the structure of the motif distribution in the intermediate length range $L_E < L < L_U$. Genomes with strongly biased motif distributions require longer oligomers (larger $L_V^\star$, see **Appendix 2—figure 2A**), whereas the replication behavior of weakly biased genomes closely mirrors that of the fully entropic genome ($L_E = 3\,\text{nt}$, $L_U = 4\,\text{nt}$) (**Appendix 2—figure 2B**). In realistic prebiotic settings, genomes are likely to fall somewhere between these extremes, depending on the strength of biases introduced during their emergence.

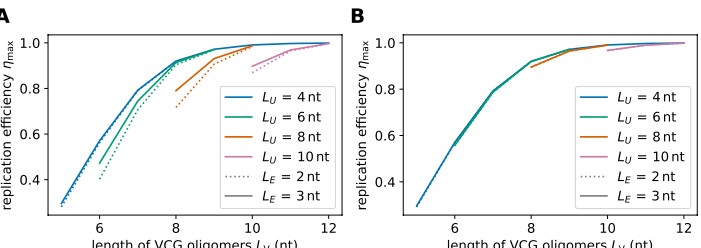

**Appendix 2—figure 2.** Maximal replication efficiency as a function of the oligomer length for different genomes (all $L_G = 64\,\text{nt}$ long). Regardless of the genome, the oligomer length needs to exceed $L_U$ to enable replication with high efficiency (e.g., higher than 95%). The difference between the genome length required for high efficiency replication and the unique motif length $L_U$ depends on the motif distribution on intermediate length scales ($L_E < L < L_U$): Genomes with strong bias require longer oligomers (**A**) than genomes with weak bias (**B**) (see **Supplementary file 1** for the genomes and their motif entropies).

## Appendix 3

### Influence of dimers on replication efficiency in single-length VCG pools

In a VCG pool comprised of monomers, dimers, and VCG oligomers of a single length, the replication efficiency can be expressed as

$$
\eta = \frac{c^{\text{eq}}\left(\left.\frac{1|L_V}{L_V}\right|_c\right) + 2\,c^{\text{eq}}\left(\left.\frac{2|L_V}{L_V}\right|_c\right)}{c^{\text{eq}}\left(\frac{1|L_V}{L_V}\right) + 2\,c^{\text{eq}}\left(\frac{2|L_V}{L_V}\right)},
$$

provided the pool operates in the concentration regime where all types of ligations other than F+V ligations are negligible. In this equation, $c^{\text{eq}}\left(\frac{L|M}{N}\right)$ denotes the equilibrium concentration of all complexes allowing for the templated ligation of oligomers of length $L$ to oligomers of length $M$ if templated by oligomers of length $N$. $c^{\text{eq}}\left(\left.\frac{L|M}{N}\right|_c\right)$ denotes the same concentration under the additional constraint that the product oligomers need to be correct, that is the sequence of the product needs to be part of the pre-defined genome sequence.

Complexes allowing for templated ligation need to involve at least three oligomers but may be comprised of more strands. For the following analytical derivation, we restrict ourselves to ternary complexes, while the full numerical solution (see continuous lines in **Figure 5C**) also includes quaternary complexes. If only ternary complexes are accounted for, each equilibrium concentration may be written as a product of the effective association constant and the equilibrium concentrations of the involved oligomers,

$$
c^{\text{eq}}\left(\frac{L|M}{N}\right) = \mathcal{K}^{\text{a}}_{\frac{L|M}{N}}\, c_s^{\text{eq}}(L)\, c_s^{\text{eq}}(M)\, c_s^{\text{eq}}(N).
$$

Using this expression, the equation for the replication efficiency may be simplified,

$$
\eta = \frac{\left.\mathcal{K}^{\text{a}}_{\frac{1|L_V}{L_V}}\right|_c + 2\,\left.\mathcal{K}^{\text{a}}_{\frac{2|L_V}{L_V}}\right|_c \frac{c_s^{\text{eq}}(2)}{c_s^{\text{eq}}(1)}}{\mathcal{K}^{\text{a}}_{\frac{1|L_V}{L_V}} + 2\,\mathcal{K}^{\text{a}}_{\frac{2|L_V}{L_V}} \frac{c_s^{\text{eq}}(2)}{c_s^{\text{eq}}(1)}}.
$$

The effective association constants in the denominator can be expressed as the sum of the association constants of correct and false products,

$$
\mathcal{K}^{\text{a}}_{\frac{1|L_V}{L_V}} = \left.\mathcal{K}^{\text{a}}_{\frac{1|L_V}{L_V}}\right|_c + \left.\mathcal{K}^{\text{a}}_{\frac{1|L_V}{L_V}}\right|_f,
$$

$$
\mathcal{K}^{\text{a}}_{\frac{2|L_V}{L_V}} = \left.\mathcal{K}^{\text{a}}_{\frac{2|L_V}{L_V}}\right|_c + \left.\mathcal{K}^{\text{a}}_{\frac{2|L_V}{L_V}}\right|_f.
$$

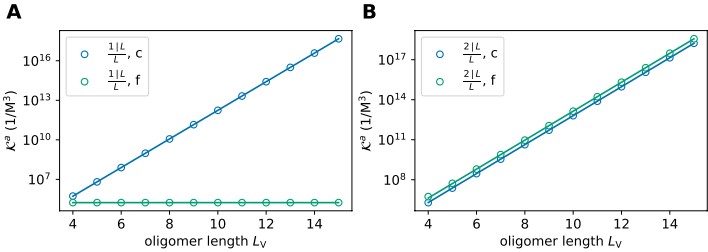

**Appendix 3—figure 1.** Effective association constants of complexes facilitating 1+V ligations (**A**) and 2+V ligations (**B**).

In the following, we simplify the notation by referring to $\frac{1|L_V}{L_V}$ as 1+V ligations, and to $\frac{2|L_V}{L_V}$ as 2+V ligations. The effective association constants differ with respect to their dependence on $L_V$: While $\mathcal{K}^a_{1+V,c}$, $\mathcal{K}^a_{2+V,c}$ and $\mathcal{K}^a_{2+V,f}$ increase exponentially with $L_V$, $\mathcal{K}^a_{1+V,f}$ is length-independent (**Appendix 3—figure 1**). Introducing the parametrizations,

$$\mathcal{K}^a_{1+V,c} = \mathcal{K}^{a,\circ}_{1+V,c} \exp(|\gamma|L_V), \quad \mathcal{K}^a_{1+V,f} = \mathcal{K}^{a,\circ}_{1+V,f},$$

$$\mathcal{K}^a_{2+V,c} = \mathcal{K}^{a,\circ}_{2+V,c} \exp(|\gamma|L_V), \quad \mathcal{K}^a_{2+V,f} = \mathcal{K}^{a,\circ}_{2+V,c} \exp(|\gamma|L_V),$$

we can express $\eta$ as

$$\eta = \frac{\mathcal{K}^{a,\circ}_{1+V,c} + 2\,\mathcal{K}^{a,\circ}_{2+V,c}\,\frac{c^{eq}_s(2)}{c^{eq}_s(1)}}{\mathcal{K}^a_{1+V,c} + \mathcal{K}^a_{1+V,f}\exp(-|\gamma|L_V) + 2\,(\mathcal{K}^{a,\circ}_{2+V,c} + \mathcal{K}^{a,\circ}_{2+V,f})\,\frac{c^{eq}_s(2)}{c^{eq}_s(1)}}.$$

Taking the limit $L_V \to \infty$ yields an upper bound for the replication efficiency,

$$\eta_{max} = \frac{\mathcal{K}^{a,\circ}_{1+V,c} + 2\,\mathcal{K}^{a,\circ}_{2+V,c}\,\frac{c^{eq}_s(2)}{c^{eq}_s(1)}}{\mathcal{K}^a_{1+V,c} + 2\,(\mathcal{K}^{a,\circ}_{2+V,c} + \mathcal{K}^{a,\circ}_{2+V,f})\,\frac{c^{eq}_s(2)}{c^{eq}_s(1)}}.$$

Provided that the ratio of equilibrium concentrations of dimers and monomers is comparable to the ratio of total concentration of dimers and monomers, $c^{eq}(2)/c^{eq}(1) \approx c^{tot}(2)/c^{tot}(1)$, we can write

$$\frac{c^{eq}_s(2)}{c^{eq}_s(1)} = \frac{\mathfrak{C}(1)}{\mathfrak{C}(2)}\frac{c^{eq}(2)}{c^{eq}(1)} \approx \frac{\mathfrak{C}(1)}{\mathfrak{C}(2)}\frac{c^{tot}(2)}{c^{tot}(1)} = \frac{\mathfrak{C}(1)}{\mathfrak{C}(2)}e^{-\kappa_F}.$$

Therefore, the upper bound of the replication efficiency reads

$$\eta_{max} = \frac{\mathcal{K}^{a,\circ}_{1+V,c} + 2\frac{\mathfrak{C}(1)}{\mathfrak{C}(2)}\,\mathcal{K}^{a,\circ}_{2+V,c}e^{-\kappa_F}}{\mathcal{K}^a_{1+V,c} + 2\frac{\mathfrak{C}(1)}{\mathfrak{C}(2)}\,(\mathcal{K}^{a,\circ}_{2+V,c} + \mathcal{K}^{a,\circ}_{2+V,f})\,e^{-\kappa_F}}.$$

For the genome that is analyzed in the main text, $\frac{\mathfrak{C}(2)}{\mathfrak{C}(1)} = \frac{16}{4} = 4$. Comparison of the analytical approach (involving only ternary complexes) to the numerical solution (involving quaternary complexes) reveals that the contribution of quaternary complexes is indeed negligible (**Figure 5**).

## Appendix 4

### Fraction of VCG oligomers extended by monomers in single-length VCG pools with kinetically suppressed V+V ligation

In pools containing monomers, dimers, and VCG oligomers of a single length, in which V+V ligations are kinetically suppressed (e.g. by only activating monomers), we observe that the fraction of oligomers that can be extended by a monomer reaches an asymptotic value for high VCG concentrations. Notably, this asymptotic value is independent of the oligomer length.

The fraction of oligomers that are in a monomer extension-competent state can be computed based on the equilibrium concentration of complexes facilitating the extension of an oligomer by a monomer,

$$
r_{1+V} = \frac{c^{\text{eq}}\left(\frac{1|L_V}{L_V}\right) + c^{\text{eq}}\left(\frac{L_V|1}{L_V}\right)}{c^{\text{tot}}(L_V)} = \frac{2\,c^{\text{eq}}\left(\frac{1|L_V}{L_V}\right)}{c^{\text{tot}}(L_V)}.
$$

Note that, in principle, the monomer can be added to the 5′- or the 3′-end of the oligomer, leading to two contributions in the numerator, which are identical due to symmetry, $c^{\text{eq}}\left(\frac{1|L_V}{L_V}\right) = c^{\text{eq}}\left(\frac{L_V|1}{L_V}\right)$. Here, $c^{\text{eq}}\left(\frac{L|M}{N}\right)$ denotes the equilibrium concentration of all complexes allowing for the templated ligation of oligomers of length $L$ to oligomers of length $M$ if templated by oligomers of length $N$. We assume that ternary complexes are the dominant type of complexes facilitating templated ligation, allowing us to express the equilibrium concentrations as

$$
c^{\text{eq}}\left(\frac{L|M}{N}\right) = \mathcal{K}^{\text{a}}_{\frac{L|M}{N}}\, c_s^{\text{eq}}(L)\, c_s^{\text{eq}}(M)\, c_s^{\text{eq}}(N).
$$

Therefore, the fraction of monomer-extendable oligomers reads

$$
r_{1+V} = \frac{\dfrac{2\,\mathcal{K}^{\text{a}}_{\frac{1|L_V}{L_V}}\, c_s^{\text{eq}}(1)\, c_s^{\text{eq}}(L_V)^2}{L_V}}{c^{\text{tot}}(L_V)}.
$$

In the limit of high VCG concentration, almost all VCG oligomers form duplexes. Hence, the equilibrium concentration of free oligomers is well-approximated by

$$
c_s^{\text{eq}}(L_V) = \sqrt{\frac{c^{\text{tot}}(L_V)}{2\mathcal{K}^{\text{a}}_{L_V/L_V}}},
$$

implying,

$$
r_{1+V}^{\infty} = \frac{\dfrac{2\,\mathcal{K}^{\text{a}}_{\frac{1|L_V}{L_V}}\, c_s^{\text{eq}}(1) c^{\text{tot}}(L_V)}{L_V}}{2\,\mathcal{K}^{\text{a}}_{L_V/L_V}\, c^{\text{tot}}(L_V)} = \frac{\dfrac{\mathcal{K}^{\text{a}}_{\frac{1|L_V}{L_V}}\, c_s^{\text{eq}}(1)}{L_V}}{\mathcal{K}^{\text{a}}_{L_V/L_V}}.
$$

As almost all monomers are free in solution, their equilibrium concentration can be approximated by the total concentration of monomers, $c_s^{\text{eq}}(1) = c^{\text{eq}}(1)/\mathfrak{C}(1) \approx c^{\text{tot}}(1)/\mathfrak{C}(1)$.

From this representation of $r_{1+V}^{\infty}$, it is not clear yet why $r_{1+V}^{\infty}$ should be independent of $L_V$. In order to derive an $L_V$-independent expression for $r_{1+V}^{\infty}$, we need to express the association constant of ternary complexes, $\mathcal{K}^{\text{a}}_{\frac{1|L_V}{L_V}}$, in terms of the duplex association constant, $\mathcal{K}^{\text{a}}_{L_V/L_V}$.

As illustrated in *Appendix 4—figure 1*, the effective association constant of the ternary complex may be obtained by multiplying the effective association constant of the duplex by the binding affinity of the monomer. We need to distinguish the contributions of duplexes based on the length of their hybridization region: If the two strands in the duplex form a hybridization region that is as long as the oligomers, there is free position for the monomer to hybridize to; such duplexes must be excluded from the computation of the association constant of ternary complexes. Moreover, it is necessary to take into account that monomers hybridizing to the end of a strand have different

binding affinity than monomers hybridizing to the center of an oligomer (*Appendix 4—figure 1A* and *Appendix 4—figure 1B-C*). All in all, the effective association constant for ternary complexes reads

$$\frac{\mathcal{K}^{\mathrm{a}}_{1|L_{\mathrm{V}}}}{L_{\mathrm{V}}} = 2\,\mathcal{K}^{\mathrm{a}}_{L_{\mathrm{V}}/L_{\mathrm{V}}}\big|_{L_{\mathrm{o}}=L_{\mathrm{V}}-1}\exp\left(\frac{|\gamma|}{2}\right)(c^{\circ})^{-1} + 2\sum_{L_{\mathrm{o}}=1}^{L_{\mathrm{V}}-2}\mathcal{K}^{\mathrm{a}}_{L_{\mathrm{V}}/L_{\mathrm{V}}}\big|_{L_{\mathrm{o}}}\exp(|\gamma|)(c^{\circ})^{-1}$$

$$\approx 2\,\mathcal{K}^{\mathrm{a}}_{L_{\mathrm{V}}/L_{\mathrm{V}}}\big|_{L_{\mathrm{o}}=L_{\mathrm{V}}-1}\exp\left(\frac{|\gamma|}{2}\right)(c^{\circ})^{-1} + 2\mathcal{K}^{\mathrm{a}}_{L_{\mathrm{V}}/L_{\mathrm{V}}}\big|_{L_{\mathrm{o}}=L_{\mathrm{V}}-2}\exp(|\gamma|)(c^{\circ})^{-1}$$

The notation $\cdot\big|_{L_{\mathrm{o}}}$ implies that only complex configurations with a specific overlap length $L_{\mathrm{o}}$ are included.

To proceed further, we express effective duplex association constant as a sum of effective duplex association constants with constrained overlap length $L_{\mathrm{o}}$,

$$\mathcal{K}^{\mathrm{a}}_{L_{\mathrm{V}}/L_{\mathrm{V}}} = \mathcal{K}^{\mathrm{a}}_{L_{\mathrm{V}}/L_{\mathrm{V}}}\big|_{L_{\mathrm{o}}=L_{\mathrm{V}}} + 2\sum_{L_{\mathrm{o}}=1}^{L_{\mathrm{V}}-1}\mathcal{K}^{\mathrm{a}}_{L_{\mathrm{V}}/L_{\mathrm{V}}}\big|_{L_{\mathrm{o}}}$$

$$\approx \mathcal{K}^{\mathrm{a}}_{L_{\mathrm{V}}/L_{\mathrm{V}}}\big|_{L_{\mathrm{o}}=L_{\mathrm{V}}} + 2\mathcal{K}^{\mathrm{a}}_{L_{\mathrm{V}}/L_{\mathrm{V}}}\big|_{L_{\mathrm{o}}=L_{\mathrm{V}}-1} + \mathcal{K}^{\mathrm{a}}_{L_{\mathrm{V}}/L_{\mathrm{V}}}\big|_{L_{\mathrm{o}}=L_{\mathrm{V}}-2}.$$

For each term contributing to the sum, the length of the hybridization region (overlap length) is known. Therefore, each contributing term can be evaluated by computing the energy associated with the known overlap length,

$$\mathcal{K}^{\mathrm{a}}_{L_{\mathrm{V}}/L_{\mathrm{V}}}\big|_{L_{\mathrm{o}}=L_{\mathrm{V}}} = \frac{1}{(c^{\circ})^2}\frac{\mathfrak{C}(L_{\mathrm{V}})}{2}\exp[(L_{\mathrm{V}}-1)|\gamma|],$$

$$\mathcal{K}^{\mathrm{a}}_{L_{\mathrm{V}}/L_{\mathrm{V}}}\big|_{L_{\mathrm{o}}=L_{\mathrm{V}}-1} = \frac{1}{(c^{\circ})^2}\frac{\mathfrak{C}(L_{\mathrm{V}})}{2}\exp[(L_{\mathrm{V}}-2)|\gamma|+2\cdot|\gamma|/2] = \frac{1}{(c^{\circ})^2}\frac{\mathfrak{C}(L_{\mathrm{V}})}{2}\exp[(L_{\mathrm{V}}-1)|\gamma|],$$

$$\mathcal{K}^{\mathrm{a}}_{L_{\mathrm{V}}/L_{\mathrm{V}}}\big|_{L_{\mathrm{o}}=L_{\mathrm{V}}-2} = \frac{1}{(c^{\circ})^2}\frac{\mathfrak{C}(L_{\mathrm{V}})}{2}\exp[(L_{\mathrm{V}}-3)|\gamma|+2\cdot|\gamma|/2] = \frac{1}{(c^{\circ})^2}\frac{\mathfrak{C}(L_{\mathrm{V}})}{2}\exp[(L_{\mathrm{V}}-2)|\gamma|].$$

Therefore, the asymptotic ratio of monomer-extended oligomers equals

$$r^{\infty}_{1+\mathrm{V}} = \frac{\frac{\mathcal{K}^{\mathrm{a}}_{1|L_{\mathrm{V}}}}{L_{\mathrm{V}}}}{\mathcal{K}^{\mathrm{a}}_{L_{\mathrm{V}}/L_{\mathrm{V}}}}\frac{c^{\mathrm{tot}}(1)}{\mathfrak{C}(1)} = \frac{2\,\mathcal{K}^{\mathrm{a}}_{L_{\mathrm{V}}/L_{\mathrm{V}}}\big|_{L_{\mathrm{o}}=L_{\mathrm{V}}-1}\exp\left(\frac{|\gamma|}{2}\right) + 2\,\mathcal{K}^{\mathrm{a}}_{L_{\mathrm{V}}/L_{\mathrm{V}}}\big|_{L_{\mathrm{o}}=L_{\mathrm{V}}-2}\exp(|\gamma|)}{\mathcal{K}^{\mathrm{a}}_{L_{\mathrm{V}}/L_{\mathrm{V}}}\big|_{L_{\mathrm{o}}=L_{\mathrm{V}}} + 2\,\mathcal{K}^{\mathrm{a}}_{L_{\mathrm{V}}/L_{\mathrm{V}}}\big|_{L_{\mathrm{o}}=L_{\mathrm{V}}-1} + \mathcal{K}^{\mathrm{a}}_{L_{\mathrm{V}}/L_{\mathrm{V}}}\big|_{L_{\mathrm{o}}=L_{\mathrm{V}}-2}}\frac{c^{\mathrm{tot}}(1)}{\mathfrak{C}(1)\,c^{\circ}}$$

$$= \frac{2\left[1+\exp\left(\frac{|\gamma|}{2}\right)\right]}{1+2+2\exp\left(-|\gamma|\right)}\frac{c^{\mathrm{tot}}(1)}{\mathfrak{C}(1)\,c^{\circ}} \approx \frac{2}{3}\left[1+\exp\left(\frac{|\gamma|}{2}\right)\right]\left[1-\frac{2}{3}\exp\left(-|\gamma|\right)\right]\frac{c^{\mathrm{tot}}(1)}{\mathfrak{C}(1)\,c^{\circ}}$$

$$\approx \frac{2}{3}\left[\exp\left(\frac{|\gamma|}{2}\right)+1-\frac{2}{3}\exp\left(-\frac{|\gamma|}{2}\right)\right]\frac{c^{\mathrm{tot}}(1)}{\mathfrak{C}(1)\,c^{\circ}}.$$

Addition of a single base pair will decrease the energy by the energy of a half nearest neighbor block, that is by $\gamma/2$. Thus, we assign $K_{\mathrm{d}}(1) = c^{\circ}\exp\left(-|\gamma|/2\right)$, and find

$$r^{\infty}_{1+\mathrm{V}} = \frac{2}{3}\left[\frac{c^{\circ}}{K_{\mathrm{d}}(1)}+1-\frac{2\,K_{\mathrm{d}}(1)}{3c^{\circ}}\right]\frac{c^{\mathrm{tot}}(1)}{\mathfrak{C}(1)\,c^{\circ}}.$$

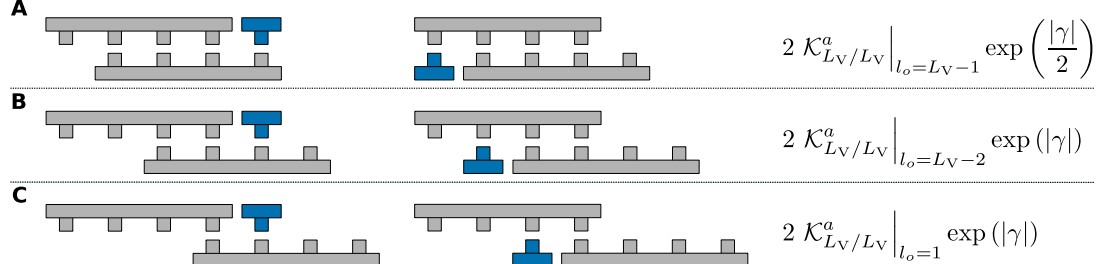

**A** $\qquad 2\,\mathcal{K}^{a}_{L_{\mathrm{V}}/L_{\mathrm{V}}}\big|_{l_{o}=L_{\mathrm{V}}-1}\exp\left(\frac{|\gamma|}{2}\right)$

**B** $\qquad 2\,\mathcal{K}^{a}_{L_{\mathrm{V}}/L_{\mathrm{V}}}\big|_{l_{o}=L_{\mathrm{V}}-2}\exp\left(|\gamma|\right)$

**C** $\qquad 2\,\mathcal{K}^{a}_{L_{\mathrm{V}}/L_{\mathrm{V}}}\big|_{l_{o}=1}\exp\left(|\gamma|\right)$

**Appendix 4—figure 1.** The effective association constants of reactive ternary complexes can be computed based on the effective association constant of the duplexes. (**A**) If the hybridization region of the two VCG oligomers is nucleotides long, the monomer hybridizes to the end (start) of the template strand. As the template has no

*Appendix 4—figure 1 continued on next page*

*Appendix 4—figure 1 continued*

dangling end, the energy contribution of the hybridizing monomer is $\gamma/2$. (**B**) and (**C**) for hybridization regions that are shorter than , but at least 1 nucleotide long, the energy contribution due to the added nucleotide is $\gamma$. In all cases, the factor 2 accounts for the two possible positions at which a monomer might be added.

## Appendix 5

### Analytical solution for equilibrium concentrations in multi-length VCG pools

We consider pools comprised of monomers, dimers, and VCG oligomers of multiple lengths. For simplicity, we restrict ourselves to concentration profiles with a uniform distribution of VCG oligomers, such that all VCG oligomers have the same concentration. Assuming that almost all oligomer mass is contained in single strands and duplexes, the mass conservation equations read

$$\mathfrak{C}(L)\, c_s^{\text{eq}}(L) + \sum_{M \neq L} \mathcal{K}_{L/M}^{\text{a}}\, c_s^{\text{eq}}(L)\, c_s^{\text{eq}}(M) + 2\mathcal{K}_{L/L}^{\text{a}}\, c_s^{\text{eq}}(L)^2 = c^{\text{tot}}(L).$$

Note that there is a mass conservation equation for each oligomer length individually, that is we are dealing with a set of multiple coupled quadratic equations. We can make the system of equations dimensionless by introducing a dimensionless concentration,

$$\tilde{c}_s^{\text{eq}}(L) = \tilde{c}_L = \frac{c_s^{\text{eq}}(L)}{\sqrt{\dfrac{c^{\text{tot}}(L)}{2\mathcal{K}_{L/L}^{\text{a}} + \sum_{M \neq L} \mathcal{K}_{L/M}^{\text{a}} \sqrt{\dfrac{c^{\text{tot}}(M)}{c^{\text{tot}}(L)}}}}}. \tag{14}$$

Rescaling $c_s^{\text{eq}}(L)$ by $\sqrt{\dfrac{c^{\text{tot}}(L)}{2\mathcal{K}_{L/L}^{\text{a}} + \sum_{M \neq L} \mathcal{K}_{L/M}^{\text{a}} \sqrt{\frac{c^{\text{tot}}(M)}{c^{\text{tot}}(L)}}}}$ is helpful, as this expression is the solution to the approximative mass conservation equation in the high concentration limit,

$$2\mathcal{K}_{L/L}^{\text{a}} c_s^{\text{eq}}(L)^2 + \sum_{M \neq L} \mathcal{K}_{L/M}^{\text{a}}\, c_s^{\text{eq}}(L)\, c_s^{\text{eq}}(M) = c^{\text{tot}}(L),$$

under the assumption that $c_s^{\text{eq}}(L) \approx c_s^{\text{eq}}(M)$. The latter assumption is reasonable given the concentration profile is uniform. As a consequence of the rescaling, we expect $\tilde{c}_L$ to be of order 1 in the high concentration limit.

Writing the full mass conservation equation in terms of the dimensionless concentration yields,

$$1 = \frac{\mathfrak{C}_L}{\sqrt{2\mathcal{K}_{L/L}^{\text{a}} + \sum_{P \neq L} \mathcal{K}_{L/P}^{\text{a}} \sqrt{\frac{c^{\text{tot}}(P)}{c^{\text{tot}}(L)}}} \sqrt{c^{\text{tot}}(L)}} \tilde{c}_L$$

$$+ \sum_{M \neq L} \frac{\mathcal{K}_{L/M}^{\text{a}}}{\sqrt{2\mathcal{K}_{L/L}^{\text{a}} + \sum_{P \neq L} \mathcal{K}_{L/P}^{\text{a}} \sqrt{\frac{c^{\text{tot}}(P)}{c^{\text{tot}}(L)}}} \sqrt{2\mathcal{K}_{M/M}^{\text{a}} + \sum_{R \neq M} \mathcal{K}_{M/R}^{\text{a}} \sqrt{\frac{c^{\text{tot}}(R)}{c^{\text{tot}}(M)}}}} \sqrt{\frac{c^{\text{tot}}(M)}{c^{\text{tot}}(L)}} \tilde{c}_L \tilde{c}_M$$

$$+ \frac{2\mathcal{K}_{L/L}^{\text{a}}}{2\mathcal{K}_{L/L}^{\text{a}} + \sum_{P \neq L} \mathcal{K}_{L/R}^{\text{a}} \sqrt{\frac{c^{\text{tot}}(R)}{c^{\text{tot}}(L)}}} \tilde{c}_L^2.$$

We drop all ratios of total concentrations, as we assume that the total concentration is the same for each oligomer length of VCG oligomer (uniform concentration profile),

$$1 = \frac{\mathfrak{C}_L}{\sqrt{2\mathcal{K}_{L/L}^{\text{a}} + \sum_{P \neq L} \mathcal{K}_{L/P}^{\text{a}}} \sqrt{c^{\text{tot}}(L)}} \tilde{c}_L$$

$$+ \sum_{M \neq L} \frac{\mathcal{K}_{L/M}^{\text{a}}}{\sqrt{2\mathcal{K}_{L/L}^{\text{a}} + \sum_{P \neq L} \mathcal{K}_{L/P}^{\text{a}}} \sqrt{2\mathcal{K}_{M/M}^{\text{a}} + \sum_{R \neq M} \mathcal{K}_{M/R}^{\text{a}}}} \tilde{c}_L \tilde{c}_M$$

$$+ \frac{2\mathcal{K}_{L/L}^{\text{a}}}{2\mathcal{K}_{L/L}^{\text{a}} + \sum_{P \neq L} \mathcal{K}_{L/R}^{\text{a}}} \tilde{c}_L^2.$$

By introducing the dimensionless prefactors,

$$\alpha_L = \frac{\mathfrak{c}_L}{\sqrt{2\mathcal{K}^{\mathrm{a}}_{L/L} + \sum_{P \neq L} \mathcal{K}^{\mathrm{a}}_{L/P}}\sqrt{c^{\mathrm{tot}}(L)}},$$

$$\beta_{L,M} = \begin{cases} \dfrac{\mathcal{K}^{\mathrm{a}}_{L/M}}{\sqrt{2\mathcal{K}^{\mathrm{a}}_{L/L} + \sum_{P \neq L} \mathcal{K}^{\mathrm{a}}_{L/P}}\sqrt{2\mathcal{K}^{\mathrm{a}}_{M/M} + \sum_{R \neq M} \mathcal{K}^{\mathrm{a}}_{M/R}}} & \text{if } L \neq M \\[2ex] \dfrac{2\mathcal{K}^{\mathrm{a}}_{L/L}}{2\mathcal{K}^{\mathrm{a}}_{L/L} + \sum_{P \neq L} \mathcal{K}^{\mathrm{a}}_{L/R}} & \text{if } L = M \end{cases}$$

we can rewrite the mass conservation equation,

$$\alpha_L \tilde{c}_L + \sum_M \beta_{L,M} \tilde{c}_L \tilde{c}_M = 1.$$

To solve for $\tilde{c}_L$, we make the assumption that the dimensionless equilibrium concentration is close to 1 for any length $L$. For this reason, we may assume that $\tilde{c}_L \approx \tilde{c}_M$. This assumption is crucial, as it allows us to de-couple the quadratic equations. Therefore, the mass conservation reads,

$$\alpha_L \tilde{c}_L + \sum_M \beta_{L,M} \tilde{c}_L^2 + \underbrace{\sum_M \beta_{L,M} \tilde{c}_L (\tilde{c}_M - \tilde{c}_L)}_{\text{small correction}} = 0$$

$$\alpha_L \tilde{c}_L + \sum_M \beta_{L,M} \tilde{c}_L^2 + \underbrace{\tilde{c}_L \sum_M \beta_{L,M} \tilde{\epsilon}_{M,L}}_{\text{small correction}} = 0$$

We can use this representation to compute the equilibrium concentrations recursively: We start the recursion with the assumption $\tilde{\epsilon}_{M,L} = 0$, and compute $\tilde{c}_L^{(0)}$ with this assumption,

$$\tilde{c}_L^{(0)} = \frac{-\alpha_L + \sqrt{\alpha_L^2 + 4\sum_M \beta_{L,M}}}{2\sum_M \beta_{L,M}}.$$

Next, we compute $\tilde{\epsilon}_{M,L}^{(0)} = \tilde{c}_M^{(0)} - \tilde{c}_L^{(0)}$, and solve the mass conservation equation for the equilibrium concentration (first recursion step),

$$\tilde{c}_L^{(1)} = \frac{-\alpha_L - \sum_M \beta_{L,M} \tilde{\epsilon}_{M,L}^{(0)} + \sqrt{\left(\alpha_L + \sum_M \beta_{L,M} \tilde{\epsilon}_{M,L}^{(0)}\right)^2 + 4\sum_M \beta_{L,M}}}{2\sum_M \beta_{L,M}} \tag{15}$$

This scheme can be applied until the approximated values of $\tilde{c}_L^{(i)}$ match the true values (obtained via numerical root finding) sufficiently well,

$$\tilde{\epsilon}_{M,L}^{(i)} = \tilde{c}_M^{(i)} - \tilde{c}_L^{(i)}$$

$$\tilde{c}_L^{(i+1)} = \frac{-\alpha_L - \sum_M \beta_{L,M} \tilde{\epsilon}_{M,L}^{(i)} + \sqrt{\left(\alpha_L + \sum_M \beta_{L,M} \tilde{\epsilon}_{M,L}^{(i)}\right)^2 + 4\sum_M \beta_{L,M}}}{2\sum_M \beta_{L,M}}$$

$$c_s^{\mathrm{eq},\,(i+1)}(L) = \sqrt{\frac{c^{\mathrm{tot}}(L)}{2\mathcal{K}^{\mathrm{a}}_{L/L} + \sum_{M \neq L} \mathcal{K}^{\mathrm{a}}_{L/M}}}\, \tilde{c}_L^{(i+1)}$$

Note that the equilibrium concentration obtained after the $(i+1)$-th iteration step $c_L^{(i+1)}$ depends on the total concentration $c^{\mathrm{tot}}(L)$ via the rescaling prefactor as well as via $\alpha_L$. It turns out that the approximation converges after about five iteration steps; the relative error between the true (numerically computed) equilibrium concentration and the approximation is already below 1% in the first iteration step.

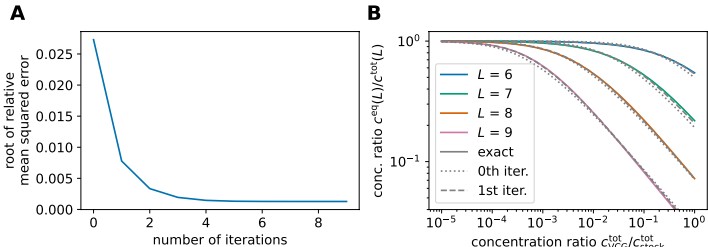

**Appendix 5—figure 1.** Comparison between approximate (analytical) and exact (numerical) solution of the (de)hybridization equilibrium in multi-length VCG pools. (**A**) The approximation converges after roughly five iteration steps. The relative error drops below 1% in the first iteration step already. (**B**) Equilibrium concentrations as a function of total VCG concentration. Continuous lines show the numerical solution, while the dotted and dashed lines depict the approximation obtained in the zeroth or first iteration step, respectively. The feedstock concentration is fixed, $c_V^{tot} = 0.1 \, \mathrm{mM}$.

## Appendix 6

### Threshold concentration for productivity inversion in multi-length VCG pools with kinetically suppressed V+V ligations

In VCG pools that contain VCG oligomers of multiple lengths as well as activated monomers, we observe 'productivity inversion': Short oligomers are more likely than long oligomers to be part of a complex capable of extending the oligomer by a monomer. However, productivity inversion can be only observed for sufficiently high concentration of VCG oligomers. The threshold concentration that is necessary for oligomers of length $M$ to exceed the monomer-extension fraction of oligomers of length $L$ (assuming $M < L$) is set by the condition $r_{1+V}(L) = r_{1+V}(M)$. To compute $r_{1+V}(L)$, we need to account for all possible complexes, in which an oligomer $L$ can be extended by a monomer,

$$r_{1+V}(L) = \frac{2\sum_N \mathcal{K}^{a}_{1|L} \dfrac{c^{eq}(L)c^{eq}(N)c^{eq}(1)}{N}}{c^{tot}(L)}.$$

As we are considering a uniform concentration profile, all VCG oligomers have the same total concentration, that is $c^{tot}(L) = c^{tot}(M)$. Thus, we can express the condition for the threshold concentration as

$$\sum_P \mathcal{K}^{a}_{1|M} \frac{c^{eq}(M)c^{eq}(P)}{P} - \sum_N \mathcal{K}^{a}_{1|L} \frac{c^{eq}(L)c^{eq}(N)}{N} = 0.$$

We combine this condition with the analytical approximation of the equilibrium concentration up to the first iteration step derived in Appendix 5 (*Equations 14 and 15*). As the equilibrium concentrations $c^{eq}(L)$, $c^{eq}(M)$, $c^{eq}(N)$, $c^{eq}(P)$ depend on the total concentration of VCG oligomers $c^{tot}(L) = c^{tot}(M) = c^{tot}(N) = c^{tot}(P)$, this yields a semi-analytical criterion for the threshold concentration $c^{tot}_{threshold}(L)$, at which the fraction of monomer-extended $M$-mers exceeds the fraction of monomer-extended $L$-mers.

# Appendix 7

## Productivity inversion in the experimental study by Ding et al.

In their experimental study, Ding et al. focus on a genome of length $L_G = 12\,\text{nt}$. In the VCG pool encoding the genome, Ding et al. include dimers with 11 different sequences, trimers with 20 different sequences, and tetramers up to 12-mers with 24 different sequences each (Tab. S1 in **Ding et al., 2023**).

Every oligomer sequence is included with a concentration of $1\,\mu\text{M}$ (so-called 1x profile), adding up to a total concentration of $\sum_{L=2}^{12} c^{\text{tot}}(L) = 247\,\mu\text{M}$. The feedstock for the replication is comprised of activated imidazolium-bridged homo-dinucleotides with a total concentration of $20\,\text{mM}$. Over a timescale of around $4\,\text{h}$, these homo-dinucleotides react with each other to form 6 additional imidazolium-bridged hetero-dinucleotides as well as activated mono-nucleotides (Fig. S1 in **Ding et al., 2023**).

We construct a genome that mimics the properties of the genome used by Ding et al., but obeys our genome construction principles outlined in the Methods section. To this end, we consider a genome of length $L_G = 12\,\text{nt}$ and a minimal unique subsequence length of $L_U = 3\,\text{nt}$. This implies that our VCG pool contains dimers with 16 different sequences and trimers up to 12-mers with 24 different sequences.

Our model does not include imidazolium-bridged dinucleotides explicitly. Instead, we assume that the concentration of activated mononucleotides in our model equals the total concentration of activated homo-dinucleotides ($20\,\text{mM}$) used in the experimental study. The total concentration of non-activated oligomers is treated as a free parameter.

We find that the system exhibits inversion of productivity: Over the entire studied concentration range, 10-mers are more likely to be extended by monomers than 12-mers (**Appendix 7—figure 1**). Moreover, 8-mers are more productive than 12-mers (provided the concentration of non-activated oligomers $\sum_{L=2}^{12} c^{\text{tot}}(L)$ exceeds roughly $0.3\,\mu\text{M}$), and more productive than 10-mers (provided $\sum_{L=2}^{12} c^{\text{tot}}(L) \gtrsim 5\,\mu\text{M}$). Thus, for the experimentally used concentration of non-activated oligomers ($\sum_{L=2}^{12} c^{\text{tot}}(L) = 247\,\mu\text{M}$, see vertical dashed line in **Appendix 7—figure 1**), 8-mers are more productive than 10-mers, which are in turn more productive than 12-mers. Unlike in the experimental system, 6-mers are less likely than 8-mers and 10-mers to be extended by a monomer. We suppose that this difference can be attributed to the differences between the experimental setup and our theoretical model: (i) We choose a (slightly) different genome than Ding et al. (ii) We model imidazolium-bridged dinucleotides as mononucleotides, as imidazolium-bridged dinucleotides only incorporate one mononucleotide at a time, just like activated mononucleotides. However, dinucleotides bind more stably to an existing complex than mononucleotides, which will affect the fraction of monomer-extended oligomers predicted.

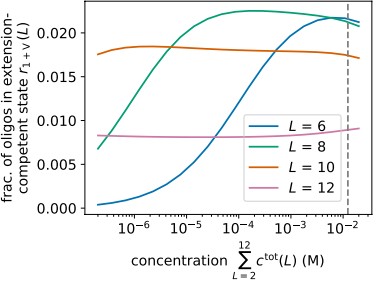

**Appendix 7—figure 1.** Replication performance of multi-length VCG pools, in which only the monomers are activated. The pool includes activated monomers ($c^{\text{tot}}(1) = 20\,\text{mM}$), as well as non-activated oligomers of length $L = 2\,\text{nt}$ up to $L = 12\,\text{nt}$ with variable total concentration $\sum_{L=2}^{12} c^{\text{tot}}(L)$. The system exhibits inversion of productivity: 10-mers are more likely to be in a monomer-extension competent state than 12-mers, 8-mers are more likely to be extended by monomers than 10-mers (for provided $\sum_{L=2}^{12} c^{\text{tot}}(L) \geq 5\,\mu\text{M}$). For the experimentally used concentration (vertical dashed line), 8-mers are more productive than 10-mers, and those are more productive than 12-mers. However, unlike in the experimental system, 6-mers are less productive than 8-mers and 10-mers.

