## [Editor Report · eLife Assessment]

This **important** theoretical study examines the possibility of encoding genomic information in a collective of short overlapping strands (e.g., the Virtual Circular Genome (VCG) model). The study presents **convincing** theoretical arguments, simulations and comparisons to experimental data to point at potential features and limitations of such distributed collective encoding of information. The work should be of relevance to colleagues interested in molecular information processing and to those interested in pre-Central Dogma or prebiotic models of self-replication.

---

## [Referee Report · Reviewer #1 (Public review)]

Summary:

This is an interesting theoretical study examining the viability of Virtual Circular Genome (VCG) model, a recently proposed scenario of prebiotic replication in which a relatively long sequence is stored as a collection of its shorter subsequences (and their compliments). It was previously pointed out that VCG model is prone to so-called sequence scrambling which limits the overall length of such a genome. In the present paper, additional limitations are identified. Specifically, it is shown that VCG is well replicated when the oligomers are elongated by sufficiently short chains from "feedstock" pool. However, ligation of oligomers from VCG itself results in a high error rate. I believe the research is of high quality and well written. However, the presentation could be improved and the key messages could be clarified.

(1) It is not clear from the paper whether the observed error has the same nature as sequence scrambling

(2) The authors introduce two important lengths LS1 and LS2 only in the conclusions and do not explain enough which each of them is important. It would make sense to discuss this early in the manuscript.

(3) It is not entirely clear why specific length distribution for VCG oligomers has to be assumed rather than emerged from simulations.

(4) Furthermore, the problem has another important length, L0 that is never introduced or discussed: a minimal hybridization length with a lifetime longer than the ligation time. From the parameters given, it appears that L0 is sufficiently long (~10 bases). In other words, it appears that the study is done is a somewhat suboptimal regime: most hybridization events do not lead to a ligation. Am I right in this assessment? If that is the case, the authors might want to explore another regime, L0

Strengths:

High-quality theoretical modeling of an important problem is implemented.

---

## [Referee Report · Reviewer #2 (Public review)]

Summary:

This important theoretical and computational study by Burger and Gerland attempts to set environmental, compositional, kinetic, and thermodynamic constraints on the proposed virtual circular genome (VCG) model for the early non-enzymatic replication of RNA. The authors create a solid kinetic model using published kinetic and thermodynamic parameters for non-enzymatic RNA ligation and (de)hybridization, which allows them to test a variety of hypotheses about the VCG. Prominently, the authors find that the length (longer is better) and concentration (intermediate is better) of the VCG oligos have an outsized impact on the fidelity and yield of VCG production with important implications for future VCG design. They also identify that activation of only RNA monomers, which can be achieved using environmental separation of the activation and replication, can relax the constraints on the concentration of long VCG component oligos by avoiding the error-prone oligo-oligo ligation. Finally, in a complex scenario with multiple VCG oligo lengths, the authors demonstrate a clear bias for the extension of shorter oligos compared to the longer ones. This effect has been observed experimentally (Ding et al., JACS 2023) but was unexplained rigorously until now. Overall, this manuscript will be of interest to scientists studying the origin of life and the behavior of complex nucleic acid systems.

Strengths:

- The kinetic model is carefully and realistically created, enabling the authors to probe the VCG thoroughly.

- Fig. 6 outlines important constraints for scientists studying the origin of life. It supports the claim that the separation of activation and replication chemistry is required for efficient non-enzymatic replication. One could easily imagine a scenario where activation of molecules occurs, followed by their diffusion into another environment containing protocells that encapsulate a VCG. The selective diffusion of activated monomers across protocell membranes would then result in only activated monomers being available to the VCG, which is the constraint outlined in this work. The proposed exclusive replication by monomers also mirrors the modern biological systems, which nearly exclusively replicate by monomer extension.

- Another strength of the work is that it explains why shorter oligos extend better compared to the long ones in complex VCG mixtures. This point is independent of the activation chemistry used (it simply depends on the kinetics and thermodynamics of RNA base-pairing) so it should be very generalizable.

---

## [Author Response]

The following is the authors’ response to the original reviews

**Public Reviews:**

**Reviewer #1 (Public review):**
Summary:This is an interesting theoretical study examining the viability of Virtual Circular Genome (VCG) model, a recently proposed scenario of prebiotic replication in which a relatively long sequence is stored as a collection of its shorter subsequences (and their compliments). It was previously pointed out that VCG model is prone to socalled sequence scrambling which limits the overall length of such a genome. In the present paper, additional limitations are identified. Specifically, it is shown that VCG is well replicated when the oligomers are elongated by sufficiently short chains from ”feedstock” pool. However, ligation of oligomers from VCG itself results in a high error rate. I believe the research is of high quality and well written. However, the presentation could be improved and the key messages could be clarified.Strengths:High-quality theoretical modeling of an important problem is implemented.Weaknesses:The conclusions are somewhat convoluted and could be presented better.(1) It is not clear from the paper whether the observed error has the same nature as sequence scrambling.

We thank the Reviewer for pointing out that this important point was not clearly explained. The sequence errors observed in our model are indeed of the same nature as sequence scrambling previously identified by Chamanian and Higgs (Chamanian and Higgs, PLoS Comp Biol 2022). The core issue is the ligation of two oligomers representing non-adjacent segments of the genome sequence, leading to the formation of ”chimeric” products that are not part of the desired genome.

Our analysis identifies the ligation of VCG oligomers (V+V reactions) as the primary mechanism driving sequence scrambling. This allowed us to propose two strategies to mitigate sequence scrambling: (i) tuning the length and concentration of the VCG oligomers, and (ii) considering scenarios where only feedstock monomers contribute to elongation (non-reactive VCG oligomers). We modified the Introduction and Results section of our manuscript to convey this connection more clearly.

(2) The authors introduce two important lengths LS1 and LS2 only in the conclusions and do not explain enough which each of them is important. It would make sense to discuss this early in the manuscript.

We agree with the Reviewer and have followed the suggestion to introduce the two important length scales earlier in the manuscript (in the Model section of the main text). In the updated version, we refer to these length scales as the *exhaustive coverage length LE* (formerly LS1) and the *unique subsequence length LU* (formerly LS2). The exhaustive coverage length *LE* is defined as the maximum motif length for which all possible sequences of that length appear somewhere in the genome. In contrast, the unique subsequence length *LU* is the minimum motif length such that each subsequence of that length occurs only once in the genome, thus giving each motif a unique ”address”.

Generally, a genome of length *LG* contains at most 2 *LG* distinct subsequences, implying that *LE* can be at most \begin{document}$L_{E}^{\max }=\left\lfloor\frac{\ln \left(2 L_{G}\right)}{\ln 4}\right\rfloor$\end{document}, and *LU* must be at least \begin{document}$L_{U}^{\min }=\left\lceil\frac{\ln \left(2 L_{G}\right)}{\ln 4}\right\rceil$\end{document}, where ⌊*...*⌋ and ⌈*...*⌉ denote the next lower and higher integer, respectively. While the previous version of the manuscript focused exclusively on the limiting case *LE* = *LEmax* and *LU* = *L**Umin* , we have extended our analysis to genomes with a broader range of *LE* and *LU* values the revised manuscript.

This extended analysis reveals that, for accurate and efficient replication, the VCG oligomer length must always exceed *LU*, regardless of the choice of *LE*. The required margin beyond *LU* depends on the distribution of intermediate-length motifs (i.e., with *LE* < *L* < *LU*), but is typically only a few nucleotides.

(3) It is not entirely clear why specific length distribution for VCG oligomers has to be assumed rather than emerged from simulations.

We have integrated these new findings into the Results section of the main text and expanded the discussion of their implications for the prebiotic relevance of the VCG scenario in the Discussion section. Full methodological details are provided in the Supplementary Material (Sections S1 and S8).

We thank the Reviewer for this insightful question. Our choice to assume specific length distributions for VCG oligomers is motivated by both conceptual and practical considerations. We explain our reasoning more clearly in the revised manuscript, in the beginning of the Model section of the main text.

Conceptually, our study focuses on the *propagation* of sequence information by an already-formed VCG, rather than its *emergence* from a random pool. As discussed by Chamanian and Higgs, the spontaneous formation of a VCG from randomly interacting oligomers is a rare event. Our aim is to understand whether, once formed, such a structure can robustly replicate under prebiotic conditions. This question is best addressed when the genome and the oligomer pool (including their lengths and concentrations) can be systematically controlled.

From a practical standpoint, working with a controllable pool of oligomers facilitates direct comparison to recent experimental studies that use predefined and well-characterized oligomer pools (Ding et al. JACS 2023). With our current methods and realistic rate constants, simulating the emergence of such pools from simple building blocks (e.g., monomers and dimers) would be computationally prohibitive, due to the low ligation rate. For example, in a system containing monomers (concentration 0.1mM) and octamers (concentration 1µM) in a volume of *V* = 3.3µm^3^, simulating the time between two ligation events takes over 300 hours of compute time (see SI Fig. S2). This renders dynamic pool generation unfeasible for the scope of our study.

(4) Furthermore, the problem has another important length, L0 that is never introduced or discussed: a minimal hybridization length with a lifetime longer than the ligation time. From the parameters given, it appears that L0 is sufficiently long (∼ 10 bases). In other words, it appears that the study is done is a somewhat suboptimal regime: most hybridization events do not lead to a ligation. Am I right in this assessment? If that is the case, the authors might want to explore another regime, L_0< LS_1, by considering a higher ligation rate.

Indeed, we assume that the ligation rate is smaller than both the hybridization and dehybridization rates for any oligomer typically included in the pool (up to length 10). In terms of effective length scales, this corresponds to *L*_0_ ≈ 10nt, with *L*_0_ defined as stated by the Reviewer, i.e., the hybridization length corresponding to a lifetime comparable to the ligation time. Most of our analysis actually exploits the small ligation rate, by employing an adiabatic approximation in which ligation is assumed to be slower than any hybridization or dehybridization process in the pool irrespective of oligomer length. As the Reviewer states, in this regime most hybridization events are transient, and will not result in ligation, since the complexes typically dissociate before ligation can occur.

While we agree that this assumption limits the overall yield of replication, it has a beneficial effect on replication fidelity. Oligomers that hybridize with mismatches tend to unbind more quickly due to the destabilizing effect of mismatches. In the slow-ligation regime, such complexes are likely to dissociate before a ligation can occur, preventing the formation of incorrect products. In contrast, if the ligation rate was comparable to the unbinding rate of mismatched hybrids, these incorrect associations could undergo ligation, thereby lowering the fidelity of replication. We thus view the regime *L*_0_ > *LV* as more favorable for studying the error-suppressing potential of the VCG mechanism, though we acknowledge that exploring the effects of faster ligation rates is an interesting question for future work.

**Reviewer #2 (Public review):**
Summary:This important theoretical and computational study by Burger and Gerland attempts to set environmental, compositional, kinetic, and thermodynamic constraints on the proposed virtual circular genome (VCG) model for the early non-enzymatic replication of RNA. The authors create a solid kinetic model using published kinetic and thermodynamic parameters for non-enzymatic RNA ligation and (de)hybridization, which allows them to test a variety of hypotheses about the VCG. Prominently, the authors find that the length (longer is better) and concentration (intermediate is better) of the VCG oligos have an outsized impact on the fidelity and yield of VCG production with important implications for future VCG design. They also identify that activation of only RNA monomers, which can be achieved using environmental separation of the activation and replication, can relax the constraints on the concentration of long VCG component oligos by avoiding the error-prone oligo-oligo ligation. Finally, in a complex scenario with multiple VCG oligo lengths, the authors demonstrate a clear bias for the extension of shorter oligos compared to the longer ones. This effect has been observed experimentally (Ding et al., JACS 2023) but was unexplained rigorously until now. Overall, this manuscript will be of interest to scientists studying the origin of life and the behavior of complex nucleic acid systems.Strengths:• The kinetic model is carefully and realistically created, enabling the authors to probe the VCG thoroughly.• Fig. 6 outlines important constraints for scientists studying the origin of life. It supports the claim that the separation of activation and replication chemistry is required for efficient non-enzymatic replication. One could easily imagine a scenario where activation of molecules occurs, followed by their diffusion into another environment containing protocells that encapsulate a VCG. The selective diffusion of activated monomers across protocell membranes would then result in only activated monomers being available to the VCG, which is the constraint outlined in this work. The proposed exclusive replication by monomers also mirrors the modern biological systems, which nearly exclusively replicate by monomer extension.• Another strength of the work is that it explains why shorter oligos extend better compared to the long ones in complex VCG mixtures. This point is independent of the activation chemistry used (it simply depends on the kinetics and thermodynamics of RNA base-pairing) so it should be very generalizable.

We thank the Reviewer for the careful assessment of our work and this concise summary of our main points.

Weaknesses:• Most of the experimental work on the VCG has been performed with the bridged 2aminoimidazolium dinucleotides, which are not featured in the kinetic model of this work. Oher studies by Szostak and colleagues have demonstrated that non-enzymatic RNA extension with bridged dinucleotides have superior kinetics (Walton et al. JACS 2016, Li et al. JACS 2017), fidelity (Duzdevich et al. NAR 2021), and regioselectivity (Giurgiu et al. JACS 2017) compared to activated monomers, establishing the bridged dinucleotides as important for non-enzymatic RNA replication. Therefore, the omission of these species in the kinetic model presented here can be perceived as problematic. The major claim that avoidance of oligo ligations is beneficial for VCGs may be irrelevant if bridged dinucleotides are used as the extending species, because oligo ligations (V + V in this work) are kinetically orders of magnitude slower than monomer extensions (F + V in this work) (Ding et al. NAR 2022). Formally adding the bridged dinucleotides to the kinetic model is likely outside of the scope of this work, but perhaps the authors could test if this should be done in the future by simply increasing the rate of monomer extension (F + V) to match the bridged dinucleotide rate without changing rate of V + V ligation?

We thank the Reviewer for this insightful comment. Indeed, we did not design our model to specifically describe the use of bridged 2-aminoimidazolium dinucleotides as feedstock for the VCG scenario. Adding the bridged dinucleotides to our model would require allowing for feedstock that effectively changes its length during the ligation reaction. As anticipated already by the Reviewer, this is outside the scope of our current modeling framework, which was chosen to explore the generic issue of sequence scrambling in the VCG scenario without distinguishing between different types of activation chemistries.

Along the lines of the Reviewer’s suggestion, we clarified in the revised manuscript that we consider two limiting cases out of a family of models with two different ligation rate constants, *k*_lig,1_ for ligations involving a monomer and *k*_lig,>1_ for ligations involving no monomer, allowing for kinetic discrimination between these processes. We consider the two limiting cases where either *k*_lig,1_ = *k*_lig,>1_ or *k*_lig,1_*/k*_lig,1_ → 0. The latter case, captures the behavior expected from an activation chemistry that enables fast primer extension but slow ligation, thereby suppressing sequence scrambling via V+V ligation events. The corresponding results, presented in Figure 6 and 7, indeed show that the VCG replication efficiency approaches 100% for pools that are rich in VCG oligomers.

Our coarse-grained model, which does not explicitly describe the activation chemistry, was sufficient to capture important kinetic and thermodynamic constraints of the VCG scenario, and to qualitatively explain the experimental observation of a preferential extension of short over long VCG oligomers (Fig. 7B). For future work, we plan to extend our model to account for the activation chemistry in detail, to allow for a more quantitative comparison between theory and experiment.

• The kinetic and thermodynamic parameters for oligo binding appear to be missing two potentially important components. First, base-paired RNA strands that contain gaps where an activated monomer or oligo can bind have been shown to display significantly different kinetics of ligation and binding/unbinding than complexes that do not contain such gaps (see Prywes et al. eLife 2016, Banerjee et al. Nature Nanotechnology 2023, and Todisco et al. JACS 2024). Would inclusion of such parameters alter the overall kinetic model?

We thank the Reviewer for highlighting these recent studies. Todisco et al. (JACS 2024) report that complexes with gaps are well described by standard nearest-neighbor models, while stacking interactions at nick sites confer additional stability beyond these predictions. Our model is therefore expected to capture the thermodynamics of complexes with gaps accurately, but likely underestimates the stability of complexes containing nicks. In the VCG pool, all productive ligation complexes (F+F, F+V, V+V) inherently contain a nick and thus benefit from this stabilization, whereas unproductive complexes typically do not. The added stability is expected to increase the residence time of oligomers in productive complexes, thereby enhancing overall extension rates. However, since this stabilization applies uniformly across all productive complexes, it does not shift the relative contributions of different ligation pathways (in particular, correct vs. incorrect).

This reasoning assumes that hybridization and dehybridization occur on timescales faster than ligation or primer extension. It is conceivable that this separation of timescales does not hold, particularly for oligomers binding to templates with gaps, where association is slower due to steric hindrance, while dissociation is further slowed by stabilizing nicks. As a result, the residence time of such complexes can become comparable to (or longer than) the ligation timescale. We now discuss this aspect more thoroughly in the revised Results and Discussion sections. Capturing the resulting effects in our analytical framework would require relaxing the adiabatic assumption, which is beyond the scope of this work. We recognize the relevance of the non-adiabatic regime of the dynamics, and hope to explore this regime in follow-up work.

• Second, it has been shown that long base-paired RNA can tolerate mismatches to an extent that can result in monomer ligation to such mismatched duplexes (see Todisco et al. NAR 2024). Would inclusion of the parameters published in Todisco et al. NAR 2024 alter the kinetic model significantly?

In contrast to complexes with nicks and gaps, mismatched complexes (Todisco et al. NAR 2024) will decrease replication fidelity relative to the results presented in our manuscript. Our current model assumes perfect base pairing, such that replication errors arise only from binding events involving regions too short to reliably identify the correct genomic position (sequence scrambling). Allowing mismatches will indeed introduce an additional error mechanism via imperfect yet sufficiently stable duplexes, thereby increasing the rate of incorrect extensions. However, we expect this effect to be limited. Due to the thermodynamic cost of internal loops, mismatched duplexes most often have their mismatches near the ends of the hybridized region, where their destabilizing effect is weakest (Todisco et al. NAR 2024). Terminal mismatches at the 3’end of the primer have been shown to reduce the primer-extension rate significantly via a stalling effect (Rajamani et al. JACS 2010, Leu et al. JACS 2013). Hence, we would expect errors due to mismatched duplexes to primarily occur for mismatches at the 5′ end. Such errors could be mitigated by a VCG pool consisting only of oligomers that are sufficiently long relative to the unique motif length of the virtual genome.

We have extended the Discussion section to address this interesting issue.

**Recommendations for the authors:**

**Reviewer #2 (Recommendations for the authors):**

• ’(apostrophes) should be prime symbols instead of apostrophes

We thank the Reviewer for spotting this mistake, which we have now corrected.

• In the Introduction, the section that discusses the fidelity of enzyme-free copying should include a reference to Duzdevich et al. NAR 2021, as that work measured the fidelity experimentally.

We have included this reference together with other references on the kinetics of hybridization/dehybridization to nicks and gaps in the main text.

• The term feedstock oligomers may be problematic, because these also include monomers. In the ”Templated ligation” section of the Model, the statement ”We consider pools in which all oligomers are activated, as well as pools in which only monomers are activated” is imprecise. ”All oligomers, including monomers,...” would be better so as to avoid confusion in readers accustomed to standard RNA language.

We thank the Reviewer for this helpful suggestion. In the revised manuscript, we now use the term feedstock (rather than feedstock oligomers) to avoid confusion. We have also revised the sentence in the ”Templated ligation” section to read ”all oligomers, including monomers, ...” as recommended.

• The ”Experimentally determined association rate constants” reference 24-26, which measured the rate constants for DNA. Considering that the authors are modeling RNA, I wonder if Ashwood et al. Biophysical Journal 2023 contains any relevant RNA data that could help refine the model?

We thank the Reviewer for pointing us to the study by Ashwood et al. We have added this reference to the corresponding paragraph in the revised manuscript. Their RNA association rate constant (∼ 5 × 10^7^ M^−1^ s^−1^) is larger than the one we used (∼ 1×10^6^ M^−1^ s^−1^), however a larger association rate is in fact beneficial for the validity of our adiabatic approximation, and thus would not affect our results, as long as the thermodynamic stability remains the same. This is because faster association then also implies faster dissociation, and the ratio of the ligation timescale to the timescales of (de)hybridization then becomes even smaller, which is the regime where the adiabatic approximation made in our analysis is justified.

• In ”Triplexes of type 1—8 and 1—9...”, the word triplexes will confuse readers with RNA expertise as triplexe simply a triple-strandedRNA.

We thank the Reviewer for pointing out the potentially ambiguous nomenclature. To avoid confusion with triplestranded RNA structures, we now refer to binary (ternary, ...) complexes instead of duplexes (triplexes, ...) throughout the revised manuscript.